# Validating a microphysical prognostic stratospheric aerosol implementation in E3SMv2 using observations after the Mount Pinatubo eruption

Hunter Brown[1], Benjamin Wagman[1], Diana Bull[1], Kara Peterson[1], Benjamin Hillman[1], Xiaohong Liu[2], Ziming Ke[3], Lin Lin[2]

[1]Sandia National Laboratories, Albuquerque, NM, USA
[2]Department of Atmospheric Sciences, Texas A&M University, College Station, TX, USA
[3]Lawrence Livermore National Laboratories, 7000 East Avenue, Livermore, CA 94550

*Correspondence to*: Hunter Brown (hybrown@sandia.gov) and Benjamin Wagman (bmwagma@sandia.gov)

**Abstract.** This paper describes the addition of a stratospheric prognostic aerosol (SPA) capability – developed with the goal of accurately simulating sulfate aerosol formation and evolution in the stratosphere – in the Department of Energy (DOE) Earth Energy Exascale Model, version 2 (E3SMv2). The implementation includes changes to the 4-mode Modal Aerosol Module microphysics in the stratosphere to allow for larger particle growth and more accurate stratospheric aerosol lifetime following the Pinatubo eruption. E3SMv2-SPA reasonably reproduces stratospheric aerosol lifetime, burden, aerosol optical depth, and top-of-atmosphere flux when compared to remote sensing observations. E3SMv2-SPA also has close agreement with the interactive chemistry-climate model CESM2-WACCM - which has a more complete chemical treatment - and the observationally-constrained, prescribed volcanic aerosol treatment in E3SMv2. Global stratospheric aerosol size distributions identify the nucleation and growth of sulfate aerosol from volcanically injected $SO_2$ from both major and minor volcanic eruptions from 1991 to 1993. Modeled aerosol effective radius is consistently lower than satellite and in-situ measurements (max differences of ~30%). Comparisons with in-situ size distribution samples indicate that this simulated underestimation in both E3SMv2-SPA and CESM2-WACCM is due to overly small accumulation and coarse mode aerosols 6-18 months post-eruption, with E3SMv2-SPA simulating ~50% the coarse mode geometric mean diameters of observations 11 months post-eruption. Effective radii from the models and observations are used to calculate offline scattering and absorption efficiencies to explore the implications of smaller simulated aerosol size on the Pinatubo climate impacts. Scattering efficiencies at wavelengths of peak solar irradiance (~0.5 μm) are 10-80% higher for daily samples in models relative to observations through 1993, suggesting higher diffuse radiation at the surface and a larger cooling effect in the models due to the smaller simulated aerosol; absorption efficiencies at the peak wavelengths of outgoing terrestrial radiation (~10 μm) are 15-40% lower for daily samples in models relative to observations suggesting an underestimation in stratospheric heating in the models due to the smaller simulated aerosol. These potential biases are based on aerosol size alone and do not take into account differences in aerosol number. The overall agreement of E3SMv2-SPA compared to observations and its similar performance to the well-validated CESM2-WACCM makes E3SMv2-SPA a viable alternative to simulating climate impacts from stratospheric sulfate aerosols.

## 1 Introduction

Explosive volcanic eruptions are a significant source of aerosol forcing given their propensity to inject gas and particulate matter into the stratosphere. These gases form long-lived aerosol that can be spread around the globe (Robock, 2000). The impacts of the stratospheric aerosol loading from these eruptions are wide ranging, as exemplified in observational and modeling findings following the Pinatubo eruption in 1991: ozone depletion (Hofmann et al., 1994; Solomon et al., 1993; Portmann et al., 1996), surface temperature decreases (Parker et al., 1996; Soden et al., 2002), lower stratosphere temperature increases (Labitzke and McCormick, 1992), reduction in global precipitation (Gillett et al., 2004), lowering of global sea-level (Church et al., 2005), increases in cirrus cloud cover (Liu and Penner, 2002; Wylie et al., 1994), as well as increased diffusivity

of incoming radiation (Robock, 2000) with resultant impacts on net primary productivity of plants (Gu et al., 2003; Proctor et al., 2018; Greenwald et al., 2006). The extent of these impacts is dependent upon characteristics of the eruption (e.g., magnitude (Marshall et al., 2019)) and climate state (Zanchettin et al., 2022).  The foundation of these physical impacts are the stratospheric aerosol microphysical properties and chemical interactions that occur after an explosive volcanic eruption. Being able to accurately simulate these aerosol microphysical and chemical reactions in Earth system models is important for improving the fidelity of simulations. It also enables climate attribution work that clarifies the role of climatic state versus characteristics of the eruption on downstream impacts. As a first step towards this goal, this paper presents a validation of a prognostic volcanic aerosol implementation within the Department of Energy (DOE) Earth Energy Exascale Model version 2 (E3SMv2) (Golaz et al., 2022; Wang et al., 2020) against observational data from the Pinatubo eruption. Furthermore, E3SMv2 is compared with version 2 of the Community Earth System Model (CESM2) (Danabasoglu et al., 2020) with the Whole Atmosphere Community Climate Model version 6 (WACCM6; Gettelman et al., 2019), which shares many similarities between E3SM in its aerosol microphysical parameterizations but has more advanced atmospheric chemistry. This is to help identify any performance issues associated with a simpler chemical treatment in E3SMv2 and to serve as further validation of our implementation.

The stratosphere contains a persistent background layer of aerosol between the tropopause and ~10 hPa that consists primarily (i.e., ~95% by mass (SPARC, 2006)) of sulfuric acid droplets, or sulfate aerosol (Junge et al., 1961). These aerosol are sustained by tropospheric influx of  gas-phase sulfate precursors carbonyl sulfide (OCS) and sulfur dioxide ($SO_2$) - along with smaller amounts of sulfate aerosol - from anthropogenic emissions and small-to-moderate volcanic eruptions (Hamill et al., 1997; Vernier et al., 2011). Intermodel comparisons of background sulfur (S) mass attribute $0.32 \pm 0.050$ Tg to OCS, $0.012 \pm 0.007$ Tg to $SO_2$, and $0.156 \pm 0.051$ Tg to sulfate aerosol with large intermodal differences due to differing model chemistry, removal processes, and dynamically driven transport and stratosphere-troposphere exchange (Brodowsky et al., 2023). Large magnitude volcanic eruptions of the last century have led to huge injections of sulfur-containing species - mainly in the form of $SO_2$ (Guo et al., 2004a) - into the stratosphere (~2-20 Tg S; McCormick et al., 1995) causing the formation of more numerous, larger particles compared to background conditions (Deshler, 2008). These volcanic particles, which have an e-folding decay time of ~1 year (Barnes and Hofmann, 1997), scatter incoming solar radiation back to space and act to cool the earth's surface by a few tenths of a degree (Robock and Mao, 1995). Recent interest in the intentional stratospheric injection of sulfate precursors to recreate this phenomenon as a means to counter anthropogenic climate change (i.e., geoengineering or stratospheric aerosol injection (SAI) (Caldeira et al., 2013)) has driven a variety of earth system modeling studies that examine the various climate and stratospheric chemistry implications of SAI scenarios (Tilmes et al., 2009; Kravitz et al., 2012, 2015; Kleinschmitt et al., 2018; Visioni et al., 2018; Weisenstein et al., 2022; Visioni et al., 2022). Multi-model comparisons of past volcanic eruptions (Zanchettin et al., 2016; Marshall et al., 2018; Timmreck et al., 2018; Clyne et al., 2021; Zanchettin et al., 2022; Quaglia et al., 2023) further quantify the downstream climate impacts from natural stratospheric injection events, identifying where models differ and where improvements can be made based on the observed climate impacts.

When simulating large-magnitude explosive volcanic eruptions, some climate models use prescribed volcanic forcing datasets as a way reduce computational demand and to avoid uncertainties in prognostic aerosol formation. These datasets can estimate forcing based on satellite data, ground based retrievals, ice core records, and other volcanic evidence (Toohey et al., 2016). One such dataset is the Global Space-based Stratospheric Aerosol Climatology (GloSSAC), which prescribes aerosol properties from a compilation of satellite, airborne, and ground based observations (Kovilakam et al., 2020; Thomason et al., 2018). While GLoSSAC and other prescribed datasets provide an accessible approach for incorporating volcanic forcing and validating model performance, prescribed aerosol products have limitations. Limited dataset availability and/or spatial coverage necessitate data interpolation within the forcing dataset, which may not accurately represent the volcanic forcing in some regions (Kovilakam et al., 2020). Additionally, prescribed aerosols will not respond to the dynamic state in free-running coupled climate simulations as the database has been generated from the observed climatic conditions. This artificially constrains the volcanic forcing across ensembles of simulations and creates a disconnect between volcanic forcing and the actual atmospheric transport patterns, limiting the usability of these simulations for detection and attribution of an evolving impact to the volcanic source. Another limitation is the lack of aerosol microphysical representation and evolution from a volcanic eruption which ignores aerosol indirect effects on clouds and does not allow for model feedbacks on aerosol size and lifetime.

Prognostically modeling the formation and evolution of sulfate aerosol from sulfur dioxide ($SO_2$) injected into the stratosphere is an alternative, more complete approach for simulating volcanic eruptions, with a variety of methods for representing sulfate aerosol mass, size, and number. This approach can serve to recreate conditions where observations are lacking as well as help elucidate microphysical processes that contribute to aerosol properties. Aerosol forcing is also more dynamic in prognostic simulations given that it is not tied to the spatial pattern of the prescribed forcing. This allows the for simulation of evolving aerosol forcings and feedbacks in fully-coupled model simulations or ensemble sets. The simplest prognostic approach is to use a *bulk* aerosol treatment, which prescribes an aerosol size distribution to a predicted aerosol species mass. This was applied to the earliest multi-year simulations of Pinatubo run with the Hamburg climate model ECHAM4 (Timmreck et al., 1999a, b), and has been used recently to show the large impact that $SO_2$ injection height can have on volcanic mass burden and climate forcing (Gao et al., 2023). The most accurate approach to simulating aerosol properties is the *sectional* (or *bin*) approach, but this can be computationally limiting depending on the number of aerosol size bins used. English et al. (2013) coupled the sectional Community Aerosol and Radiation Model for Atmospheres (CARMA; Toon et al. (1988)) with version 3 of the Whole Atmosphere Community Climate Model (WACCM3; Garcia et al. (2007)), showing the value of a sectional model in simulating the large variation in aerosol mode size and width that occurs after Pinatubo and larger magnitude eruptions. More recently, Tilmes et al. (2023) showed that coupling CARMA to WACCM6 better represents the largest aerosol sizes following Pinatubo than a parallel running modal aerosol model. The *modal* aerosol approach represents aerosol size distributions by multiple, evolving lognormal functions. While this method strikes a balance between bulk simplicity and sectional cost, a downside is its dependence on defined modal widths, which can greatly impact stratospheric aerosol removal rates following Pinatubo if not tuned to match the observed stratospheric conditions (Kokkola et al., 2018). A

modal aerosol approach is used in a stratospheric, prognostic aerosol treatment with interactive ozone chemistry (Mills et al., 2016), developed in the Community Earth System Model (CESM, version 1) (Hurrell et al., 2013) using the version 4 of WACCM (Marsh et al., 2013). This model design has been used to identify the impacts of volcanic eruptions on stratospheric ozone (Ivy et al., 2017; Solomon et al., 2016), the importance of interactive chemistry on the representation of sulfate formation and distribution following Pinatubo (Mills et al., 2017), as well as effective strategies for geoengineering (Kravitz et al., 2017).

The accurate simulation of the stratospheric sulfate size distributions is important for simulating the climate impacts of volcanic eruptions: both the scattering of incoming shortwave energy (i.e., surface cooling) and absorption of outgoing longwave energy (i.e., stratospheric warming) are related to aerosol size and number through Mie theory; and size and number in turn depend on the aerosol modeling approach (e.g., bulk, bin, or modal). The choice of aerosol representation  plays a large role in accurately representing the range and variation in volcanic size distributions (English et al., 2013; Tilmes et al., 2023) and can result in more than 50% difference in effective radius ($R_{eff}$) (Laakso et al., 2022) depending on a bin versus modal configuration. The performance of modal aerosol models is also highly dependent on choice of modal widths (Weisenstein et al., 2007; Kokkola et al., 2018). The aerosol nucleation parameterizations within climate models can also affect aerosol formation, with the binary homogeneous nucleation scheme used in many climate models (Vehkamäki et al., 2002) potentially overpredicting nucleation rates by up to 4 orders of magnitude (Yu et al., 2023), leading to underestimating aerosol $R_{eff}$ and overestimating aerosol number. Furthermore, neglecting van der Waals attractive forces which aid in coagulation of smaller aerosol particles may also contribute to a small bias in $R_{eff}$ in some models (English et al., 2013; McGraw et al., 2024).

Model simulations of Pinatubo use a variety of injection parameters, empirically chosen to match observations. Choice of vertical injection heights (17 km (Stenchikov et al., 2021); 18-21 km (Sheng et al., 2015); 18-20 km (Mills et al., 2016); 21-23 km (Dhomse et al., 2020)) span the 18-25 km range estimated from observations (Guo et al., 2004b), and models that do not include short-lived volcanic ash scale their mass emissions to account for the rapid removal of $SO_2$ that condenses on ash following the pinatubo eruption (Neely III and Schmidt, 2016; Mills et al., 2016; Clyne et al., 2021). The choice of $SO_2$ injection height can have a larger impact on volcanic mass burden and climate forcing than injection mass or particle size (Gao et al., 2023). Lower stratospheric injection heights for Pinatubo (19 km) across model simulations result in too rapid a northward transport of the plume compared to observations, leading to more rapid removal and shorter aerosol lifetimes (coarser model vertical resolution can lead to a similar effect (Brodowsky et al., 2021)) (Quaglia et al., 2023). This is related to tropical aerosol retention, which is correlated with larger $R_{eff}$ and a longer global mean aerosol optical depth (AOD) e-folding time (Clyne et al., 2021). Volcanic ash has a short lifespan on the order of days and many models neglect to include this feature in their Pinatubo simulations. However, inclusion of volcanic ash in model simulations leads to strong absorption of longwave and shortwave radiation shortly after the eruption, leading to local dynamics changes in the cloud vicinity (Niemeier and Timmreck, 2009) and lofting the volcanic plume, the latter of which increases the plume height, aerosol lifetime, AOD, and $R_{eff}$ (Stenchikov et al., 2021; Abdelkader et al., 2023).

Rates of sulfate aerosol formation and growth are driven by the oxidation of $SO_2$ by hydroxyl radical (OH) to form $H_2SO_4$ (sulfuric acid gas). This reaction in the stratosphere is OH limited. Models with more advanced interactive chemistry

represent this OH depletion, resulting in longer aerosol lifetimes than models with simpler chemical treatments and prescribed OH (Bekki, 1995; Mills et al., 2017; Clyne et al., 2021). Prescribed OH models tend to rapidly oxidize available $SO_2$ which has been attributed to peak sulfate burdens occurring 3 months earlier than for interactive chemistry models studying the 1815 Tambora eruption (Clyne et al., 2021). Water vapor is also another important reactant in the formation of sulfate aerosol, controlling the nucleation of sulfuric acid gas into sulfate aerosol and increasing the availability of OH through its interaction of exited oxygen ($O(^1D)$, a product of ozonolysis) to form two OH molecules ($H_2O + O(^1D) \rightarrow 2OH$) (Seinfeld and Pandis, 2006; LeGrande et al., 2016). When it is coinjected with $SO_2$ from volcanic eruptions, water can significantly increase OH concentrations and plume AOD (LeGrande et al., 2016; Stenchikov et al., 2021; Abdelkader et al., 2023).

Here we present a new stratospheric prognostic aerosol capability within E3SMv2 that modifies the microphysical treatment of stratospheric aerosol in the 4-mode Modal Aerosol Module (MAM4; Liu et al. (2012, 2016)) to enable simulation of the evolution of volcanic stratospheric aerosols and their properties. Similar to Mills et al. (2016), we add a stratospheric specific sulfate treatment to compliment the preexisting MAM4 chemistry and physics (default MAM4 includes the oxidation of $SO_2$ to form sulfate aerosol, their further growth through condensation and coagulation into larger aerosol size modes, sedimentation of these aerosols, and removal via wet and dry deposition). This model parallels work by Ke et al. (In Prep.) and Hu et al. (In Prep.) on a 5-mode Modal Aerosol Module (MAM5) that incorporates more complete sulfate chemistry and an additional volcanic sulfate mode in E3SMv2. The validation of our implementation presented here will support forthcoming detection and attribution studies of societally relevant climatic impacts from stratospheric aerosols in free-running coupled climate simulations with varying volcanic source characteristics. By enabling dynamical consistency between transport, aerosol distribution, microphysical properties, and eruption characteristics (e.g., impact magnitude, timing and location), this modeling capability facilitates the development of multivariate and multi-step attribution studies sensitive to spatio-temporal evolution (Hegerl et al., 2010). As future studies with this model capability will be free-running, they also enable better differentiation of the role of the climatic state on the detected and attributed impact.

With this new aerosol capability we detail the particle evolution and examine how model representations of the aerosol size distributions are related to global and regional radiative impacts at the surface and in the stratosphere. We use both observations from the 1991 eruption of Pinatubo as well as CESM2-WACCM to validate the implementation and demonstrate that we can reasonably simulate the lifetime, burden, AOD, and TOA flux perturbations of stratospheric sulfate without the computationally expensive, whole-atmosphere, comprehensive chemistry in WACCM. These comparisons are enabled by constraining the model atmospheres to reanalysis horizontal winds present at the time of eruption. Additionally, we highlight the importance of simulated aerosol size and number on shortwave and longwave radiative impacts through comparisons of aerosol $R_{eff}$ and number distributions to in-situ observations. Utilizing these effective radii in single particle Mie scattering calculations, we explore how variations in modeled aerosol microphysics affect the volcanic aerosol impacts on diffuse/direct radiation at the surface as well as longwave absorption and heating rates in the stratosphere.

## 2. Models and Simulations

In this work we modify E3SMv2 to include a stratospheric prognostic aerosol capability. We test stratospheric microphysical and chemical implementations in the context of the Pinatubo eruption and compare to version 2 of the Community Earth System Model CESM2 with the Whole Atmosphere Community Climate Model version 6 (WACCM6; Gettelman et al., 2019).

### 2.1 E3SMv2

Originally a branch from CESM1, E3SMv1 diverged with a focus on computational efficiency, scalability, vertical and horizontal resolution, aerosol and cloud parameterizations, as well as more physically based biogeochemistry, river, and cryosphere models (Leung et al., 2020). Simulations in this study are run with E3SMv2 in which clouds are parameterized with an improved version of the Cloud Layers Unified by Binormals (CLUBB) scheme (Larson, 2017), cloud microphysics are simulated by a two-moment bulk microphysics parameterization (MG2; Gettelman and Morrison, 2015), and mixed phase ice nucleation depends on aerosol type and concentration as well as temperature (Wang et al., 2014; Hoose et al., 2010). Aerosols are simulated with MAM4 (Liu et al., 2016). The explosive volcanic eruption treatment prescribes stratospheric volcanic light extinction from version 1 of the GloSSAC reanalysis dataset (Thomason et al., 2018), which is mainly derived from SAGE-II satellite measurements (Sections 3.1.1, 3.5.1) assuming a sulfate refractive index from Palmer and Williams (1975) and aerosol volume and surface area assumptions from Thomason et al. (2008). E3SM the Rapid Radiative Transfer Method for GCMs (RRTMG) (Iacono et al., 2008; Neale et al., 2012), a two-stream approximation for calculating multiple scattering in the atmosphere from gas and condensed phase (i.e., aerosol, liquid cloud droplets, cloud ice, and hydrometeors) optical properties. Atmospheric chemistry is represented with version 2 of the interactive stratospheric ozone ($O_3$) model (O3v2; Tang et al., 2021). This model uses linearized stratospheric chemistry (Linoz v2; Hsu and Prather, 2009), which calculates net $O_3$ production as a function of temperature, local $O_3$ concentration, and overhead column $O_3$.

### 2.1.1 Prognostic aerosol in E3SMv2-PA

In the default prognostic volcanic aerosol simulations (E3SMv2-PA), prescribed volcanic extinction is removed and the sulfate aerosol precursor, $SO_2$, is emitted in the stratosphere. The emitted $SO_2$ undergoes chemical reactions to form sulfate aerosol and condenses onto the surfaces of preexisting aerosols following the prognostic calculations of MAM4.

In E3SMv2-PA, default MAM4 size modes are employed. All aerosol species are represented by three size modes, shown here with 10[th]/90[th] percentile global, annual average number distribution dry diameter ranges from Liu et al. (2012): Aitken (0.015-0.053 µm), Accumulation (0.058-0.27 µm), and Coarse (0.80-3.65 µm). A fourth mode (i.e., primary carbon mode (0.039-0.13 µm)) represents freshly emitted black carbon and organic carbon from combustion, which then ages into the accumulation mode. Aerosol mass and number are used to define a modal geometric mean diameter ($D_g$). This calculated $D_g$, in conjunction with a fixed modal geometric standard deviation ($\sigma_g$), defines the modal number distribution. The modal

distributions then evolve based on nucleation (aerosol formation), evaporation (aerosol size reduction), condensation and

coagulation (aerosol size growth), and dry/wet deposition (aerosol removal). In E3SMv2-PA, the growth of accumulation to coarse mode aerosol is not included because the troposphere seldom has high enough aerosol mass concentrations to generate such large aerosols through condensation and coagulation. Thus, accumulation mode $D_g$ is allowed to increase until they reach the upper modal threshold ($D_{g,high}$) whereby the model increases accumulation mode number to maintain $D_{g,high}$ until $D_g$ begins decreasing.

**2.1.2 Prognostic stratospheric aerosol in E3SMv2-SPA**

Because MAM4 was designed to accurately represent tropospheric aerosol at their respective concentrations and emission fluxes, the sulfate formation from the massive stratospheric influx of $SO_2$ from the eruption of Pinatubo is not accurately represented in E3SMv2-PA. In addition to removing prescribed volcanic extinction – as for E3SMv2-PA - further modifications are made to MAM4 to create a prognostic stratospheric aerosol version of E3SMv2 (E3SMv2-SPA) that has

improved stratospheric aerosol representation following the Pinatubo eruption. These improvements borrow heavily from changes made to version 1 of CESM (CESM1) with WACCM (Appendix B in Mills et al., 2016), which are present in the default MAM4 version in CESM2-WACCM6. The major modifications to MAM4 include (1) the transfer of aerosol mass and number from the accumulation to coarse mode to increase aerosol size and represent the rapid aerosol growth following the Pinatubo eruption and (2) adjustment of the coarse mode and accumulation mode $\sigma_g$ and minimum/maximum geometric mean

diameters to increase aerosol lifetime. We note that these changes make the E3SMv2-SPA modal widths and size cutoffs identical to those in CESM2-WACCM6. Additional steps were taken to tune E3SMv2-SPA following the change in the accumulation and coarse mode size properties in (2) which included tuning of dust and seasalt emissions to account for the increased coarse mode lifetime in the model as well as recalculating modal optical properties in MAM4 to account for the changes in aerosol size limits and distribution widths.

Unlike typical tropospheric conditions, explosive volcanic eruptions into the stratosphere provide ample $SO_2$ mass to drive sulfate aerosol into the coarse mode. To represent this rapid growth and overall larger aerosol diameters in the stratosphere an irreversible accumulation mode number and mass transfer into the coarse mode is added to E3SMv2-SPA. The model calculates the mass and number of particles in the tail of the distribution above a specified size cut off ($D_{g,cut}$) of 0.44 µm, transferring this overshooting number and volume into the coarse mode. The model prohibits transfer from the

accumulation mode if $D_g < 0.166$ µm and allows total transfer of the gridcell mass and number when $D_g > 0.47$ µm. In CESM2-WACCM, this transfer is reversible in the stratosphere, with an aqueous sulfuric acid ($H_2SO_4$) equilibrium pressure that depends on temperature and relative humidity. We left this out of our implementation under the assumption that, at the low relative humidies and low temperatures characteristic of the stratosphere, the effects from this process would be minimal.

To improve stratospheric aerosol lifetime in E3SMv2-SPA, the default coarse mode $\sigma_g$ is reduced from 2.0 to 1.2.

The default accumulation mode $\sigma_g$ is also reduced, from 1.8 to 1.6, which has a small effect on aerosol lifetime. Additional

changes to the aerosol modes allow for overlap between the coarse and accumulation modes and include increasing the accumulation mode $D_{g,high}$ from 0.44 μm to 0.48 μm and decreasing the coarse mode lower threshold ($D_{g,low}$) from 1.0 to 0.4 μm. Lastly, coarse mode $D_{g,high}$ is increased from 4.0 μm to 40 μm. See Supplementary Table S1 for a summary of these changes.

265 Most of the above changes have little effect on the tropospheric aerosols, except for changes to the coarse mode, which leads to longer lived coarse mode aerosol due to a reduction in removal rates. To account for this, emissions of dust and sea salt are tuned such that a simulation with perpetual present-day forcing obtains a global average AOD (0.1617) and global average dust AOD (0.0281) comparable to present day remote sensing observations of ~0.17 (Lee and Chung, 2013) and 0.028-0.03 (Ridley et al., 2016), respectively. The modal aerosol optical parameterizations are also affected by changes to the

270 prescribed mode $\sigma_g$, and the modal optical properties were recalculated with the above modifications using the Ghan and Zaveri (2007) offline code used to generate the original files for CESM/E3SM.

 The tuning of coarse mode aerosol does not appear to significantly affect global measures of the simulated tropospheric climate. Two fully-coupled, 164-year historical simulations (1850-2014) were run with E3SMv2-SPA, initialized from years 50 and 100 of a 100-year pre-industrial spin-up simulation and run with the Model for Prdiction Across Scales-

275 Ocean (MPAS-ocean) (Golaz et al., 2022). These simulations show total AOD (Fig. S1), 2m surface temperatures (T2m ; Fig. S2), and global radiative balance (Fig. S3) that track the five-member E3SMv2 historical simulations with prescribed volcanic forcing from Phase 6 of the Coupled Model Intercomparison Project (CMIP6; (Golaz et al., 2022)). Differences in atmospheric modes of variability (e.g., El Niño Southern Oscillation (ENSO; (Trenberth, 1997))) due to internal variability affect T2m during the Pinatubo period (Fig. S2, S4), but interval variability would average out if a mean were taken over more ensemble

280 members.

## 2.2 CESM2-WACCM6

 The major CESM2 atmosphere model improvements from CESM1 are the inclusion of CLUBB, MG2 cloud microphysics, MAM4, and orographic wave drag parameterizations (Danabasoglu et al., 2020). These are the same in E3SMv2 (Golaz et al., 2022) with the exception of a stratospheric prognostic aerosol treatment in MAM4 in CESM2-WACCM6 (Mills

285 et al., 2016, 2017) (see Section 2.1.2). WACCM6 includes updated atmospheric chemistry, aerosol microphysics, and gravity wave drag parameterizations from previous versions of WACCM. Atmospheric chemistry is treated comprehensively through the whole atmospheric column, representing key chemical species and reactions across the troposphere, stratosphere, mesosphere, and lower thermosphere (Gettelman et al., 2019). Within the stratosphere and mesosphere, WACCM6 explicitly calculates the net production and transport of 97 different chemical species described by nearly 300 reactions (Mills et al.,

290 2017). This comprehensive chemical treatment is a key difference between CESM2-WACCM6 and E3SMv2 (including E3SMv2-PA and E3SMv2-SPA), as E3SMv2 prescribes from observationally derived climatologies of OH and other relevant chemical species in sulfur and ozone chemistry (Hsu and Prather, 2009).

**2.3 Simulations**

E3SMv2 simulations are run from 1990-1993 (1989 discarded for aerosol spinup). The model uses the horizontal and vertical resolution described in (Golaz et al., 2022), with the dynamics run on a ~110 km horizontal grid, the physics run on a coarsened ~165 km horizontal grid, and both dynamics and physics using the same 72 layer vertical grid with a model top at approximately 0.1 hPa. Simulations have prescribed sea ice and sea surface temperatures (Taylor et al., 2000) and nudged column resolved U and V winds to 6-hourly MERRA2 reanalysis data (Gelaro et al., 2017). CESM2-WACCM6 simulations are run over the same time period on a 0.95˚x1.25˚ grid over 88 pressure levels (model top at ~$4.5\times10^{-6}$ hPa). In addition to nudging model U and V winds to 6-hourly MERRA2 reanalysis data, the CESM2-WACCM6 simulations also nudge column resolved model temperature to the reanalysis to constrain temperature-dependent stratospheric chemical reactions. Note that both E3SM and CESM2-WACCM6 have a variety of stratospheric dynamics biases (e.g., Gettelman et al., 2019)) that are avoided here through atmospheric nudging. An upcoming publication on E3SM stratospheric processes details a variety of biases in E3SM that may impact free-running volcanic eruption modeling, including a weak-amplitude tropical Quasi-Biennial Oscillation which oscillates too frequently and a weak Brewer-Dobson circulation (Christiane Jablonowski, personal communication, 2024).

For simulations that prognostically simulate volcanic aerosol formation (E3SMv2-PA, E3SMv2-SPA, CESM2-WACCM6), the $SO_2$ emissions for explosive volcanic eruptions are from VolcanEESMv3.11, a modified version of Neely III and Schmidt (2016). The VolcanEE3SMv3.11 dataset contains estimates of $SO_2$ from volcanic eruptions on a 1.9x2.5-degree latitude by longitude grid, with 1 km altitude spacing from the surface to 30 km. In our period of interest (1991-1993) this includes the Pinatubo, Hudson, Spurr, and Lascar eruptions. $SO_2$ emissions are provided in molecules $cm^{-3}$ $s^{-1}$, and all eruptions occur over a six-hour period. The modifications to Neely III and Schmidt (2016) are described in Mills et al. (2016) and include a reduction in $SO_2$ emissions for eruptions over 15 Tg $SO_2$ by a factor of 0.55 to compensate for missing ice and ash removal processes. In the case of Pinatubo, while 18-19 Tg of $SO_2$ erupted in the atmosphere, only ~10 Tg remained in the stratosphere 7-9 days after the eruption (Guo et al., 2004b). This rapid reduction in $SO_2$ corresponds to >99% removal of volcanic ash mass (Guo et al., 2004a). Therefore, 10 Tg of $SO_2$ is emitted in this dataset for further chemical and microphysical evolution (Mills et al., 2016). The emission takes place between 18-20 km, at a single lat-lon grid cell (i.e., no spreading). For all simulations, the VolcanEESMv3.11 file was merged with the monthly CMIP6 SO2 emissions for non-explosive volcanic sources and then remapped to 1x1 degree.

A prescribed volcanic forcing simulation (E3SMv2-presc) is run in addition to the prognostic volcanic aerosol simulations. This simulation uses the default prescribed forcing dataset in E3SMv2 (GLoSSAC V1) and allows for an additional validation of prognostic aerosol model performance where observational data are lacking.

Table 1 provides a summary of some of the key model characteristics for the different sensitivity studies used in this work. These studies include: E3SMv2 with prescribed volcanic forcing and no emission of volcanic $SO_2$ (E3SMv-presc); E3SMv2 with the default MAM4 prognostic aerosol treatment (i.e., no stratospheric aerosol modifications) and emission of

volcanic $SO_2$ (E3SMv2-PA); E3SMv2 with the prognostic stratospheric aerosol modifications and emission of volcanic $SO_2$ (E3SMv2-SPA); and CESM2-WACCM6 with emission of volcanic $SO_2$ (from herein referred to as CESM2-WACCM).

**Table 1: Model details for the simulations used within this study. All simulations are run for 5 years (1989-1993) with 1989 discarded for aerosol spinup. All E3SMv2 simulations are run with U + V winds nudged to MERRA2 reanalysis data; CESM2-WACCM has U + V winds and temperature nudged to MERRA2 reanalysis.**

| Model Version | Horizontal, Vertical resolution | Chemistry | Stratospheric Volcanic Aerosol |
|---|---|---|---|
| E3SMv2-presc | ne30pg2<br><br>72 vertical levels (0.01 − 1000 hPa) | Linear ozone chemistry | • Prescribed volcanic forcing (aerosol extinction; no physical aerosol equivalent) derived from satellite, airborne, balloon, and ground based observations (GLoSSAC V1; Thomason et al., 2018) |
| E3SMv2-PA | ne30pg2<br><br>72 vertical levels (0.01 − 1000 hPa) | Linear ozone chemistry | • Injection of volcanic $SO_2$<br>• Default MAM4 (i.e., no stratosphere-specific modifications) |
| E3SMv2-SPA | ne30pg2<br><br>72 vertical levels (0.01 − 1000 hPa) | Linear ozone chemistry | • Injection of volcanic $SO_2$<br>• Prognostic stratospheric aerosol in MAM4<br>• Stratosphere-specific accumulation to coarse mode transfer |
| CESM2-WACCM | 0.95˚x1.25˚<br><br>88 vertical levels ($4.5 \times 10^{-6}$ − 1000 hPa) | Interactive ozone chemistry | • Injection of volcanic $SO_2$<br>• Prognostic stratospheric aerosol in MAM4<br>• Stratosphere-specific, reversible accumulation-to-coarse mode transfer |

**2.4 Effective radius, size distributions, and Mie scattering calculations**

Aerosol size distributions in the model provide information about how MAM4 represents volcanic aerosol evolution and can also help explain changes in radiation balance in the earth system. Here we calculate effective radius ($R_{eff}$) and use this in single-particle Mie scattering to understand how changes in size affect diffuse/direct radiation at the surface and absorption of longwave radiation in the stratosphere. We also plot stratospheric size distributions for our model simulations to visualize the evolution of volcanic injection of $SO_2$ into sulfate. See Appendices A and B for more details on these calculations.

**3. Observational data sets**

In addition to comparing to CESM-WACCM, we also employ observational datasets of sulfate burden, AOD, top-of-atmosphere (TOA) flux, atmospheric temperatures, and microphysical properties ($R_{eff}$ and size distributions) to substantiate the performance of the prognostic aerosol capability implemented within E3SMv2. Details of the observational datasets are presented below.

**3.1 Sulfate burden**

**3.1.1 HIRS**

The High Altitude Infrared Radiation Sounder (HIRS; Baran and Foot, 1994) is an infrared radiometer measuring surface reflectance at 19 different infrared channels (3.7 – 15 μm) and one solar channel (0.69 μm) from a variety of polar orbiting NOAA platforms since 1978 (Borbas and Menzel, 2021). This study uses HIRS-derived aerosol mass loading over

350 the period of May 1991 through October 1993 from Baran and Foot (1994), who used the difference between 8.3 μm (aerosol sensitive) and 12.5 μm (aerosol insensitive) channels to isolate the transmission through the volcanic plume. To back out aerosol mass loading, they assumed an average stratospheric sulfate aerosol composition of 75% $H_2SO_4$ + 25% $H_2O$ by mass, particle size and concentration from dustsonde measurements in July 1991 (Deshler et al., 1992), and a single scattering albedo calculated from Mie theory by integrating over scattering and extinction coefficients from an assumed lognormal distribution

of radius 0.35 μm and standard deviation 1.6. This data covers 80°N – 80°S at 5° resolution with a systematic error of ~10% (±1.4 Tg aerosol) due to assumptions in processing and uncertainty in background concentration. Additional minimum and maximum aerosol composition bounds are included in this range, namely 59%–77% $H_2SO_4$.

**3.1.1 SAGE-3λ**

This work also uses stratospheric sulfate burden taken from the SAGE-3 λ dataset compiled for CMIP6

(ftp://iacftp.ethz.ch/pub_read/luo/CMIP6/, last access: 12 January 2023) as reported in Quaglia et al. (2023). The SAGE-3λ dataset uses the Stratospheric Aerosol and Gas Experiment II (SAGE-II; Section 3.5.1) wavelengths of 0.454, 0.525, 1.024 μm, fitting the measured extinction at these wavelengths to a lognormal size distribution and estimating sulfate mass burden from the number density, mode radius, and width of the distribution (Revell et al., 2017)

### 3.2 Aerosol Optical Depth

### 3.2.1 AVHRR

The Advanced Very High Resolution Radiometer (AVHRR; Zhao et al., 2013; Heidinger et al., 2014) is a radiometer that measures surface reflectance in six spectral bands (0.63, 0.86, 1.6, 3.75, 11, and 12 µm) and has served as a meteorological imaging sensor on the NOAA polar orbiting platforms since 1978. It has a 1.1 km spatial resolution and, during the Pinatubo eruption period of interest (1990-1994), had 2-4 global views per day (Heidinger et al., 2014). An offline radiative transfer model is used to determine lookup tables for AOD retrievals. The radiative transfer model assumes fine and coarse mode aerosol properties based on validation of AVHRR with the surface radiometer measurements from the Aerosol Robotic Network (AERONET; Zhao et al., 2002), and uncertainty in the AVHRR AOD is estimated at 11.3 % based on surface AERONET validation (Zhao, Xuepeng, 2022). This work uses monthly, clear-sky AOD retrieved over oceans and regridded to 1˚ resolution. Retrievals are made from channel 1 (0.63 µm) and related to the radiatively equivalent AOD at 0.6 µm through a radiative transfer and surface/atmosphere model (Rao et al., 1989). Detection limits on AOD from the AVHRR range from a minimum of 0.01 and a maximum of 2 (Russell et al., 1996).

### 3.3 Top-of-atmosphere radiative flux

### 3.3.1 ERBS

The TOA global radiative flux at a 1˚x1˚ resolution is used from version 2 of the Diagnosing Earth's Energy Pathways in the Climate project (DEEP-C) merged data product drawing from the Earth Radiation Budget Satellite (ERBS) near-global (60˚S-60˚N) non-scanning instrument and other reanalysis and observational datasets (Allan et al., 2014). The ERBS instrument measures reflected shortwave radiation and total outgoing radiation, allowing for the separation of longwave radiative flux by subtraction (Minnis et al., 1993).

### 3.4 Atmospheric temperature profiles

### 3.4.1 MERRA-2

The Modern-Era Retrospective Analysis for Research and Applications, version 2 (MERRA-2) is a reanalysis product that assimilates satellite, radiosonde, radar, ship, buoy, and aircraft observations into version 5.12.4 of the Goddard Earth Observing System (GEOS) atmospheric general circulation model (Rienecker et al., 2011; Gelaro et al., 2017). This data is produced on a 0.5˚x0.625˚ grid with 72 vertical levels from the surface to 0.01 hPa. MERRA-2 observations include atmospheric state (temperature, pressure, humidity), dynamics, precipitation, radiation, and ozone, with updated aerosol observations from AVHRR over the period 1979-2002 (Gelaro et al., 2017).

### 3.4.2 RICH-obs

Version 1.5.1 of the Radiosonde Innovation Composite Homogenization (RICH-obs) software package is a compiled global radiosonde dataset that is merged with the help of reanalysis climatologies and neighboring data temperature records dating back to 1958 (Haimberger et al., 2012, 2008). The data gaps in station data are identified by divergence from 40-year climatology in the European Center for Medium-Range Weather Forecasts Reanalysis (ERA-40) and the interpolation of these gaps are estimated from time series of neighboring radiosonde measurements, making RICH-obs less affected by satellite observations in the reanalysis but potentially biased in remote regions due to interpolation errors. This dataset also consists of 32-ensemble members that span a variety of sensitivity parameters and thresholds for interpolating to nearby radiosonde time series (Haimberger et al., 2012).

## 3.5 Effective radius and size distributions

### 3.5.1 SAGE-II

The SAGE-II (Mauldin et al., 1985) instrument flew from October 1984 to August 2005 on the Earth Radiation Budget Satellite (ERBS), measuring light extinction through the atmospheric limb at seven channels from 0.385 to 1.02 µm. The global coverage is 80˚S-80˚N and 1 km vertical resolution. This work uses aerosol $R_{eff}$ from SAGE-II version 7 (Damadeo et al., 2013) over the tropics (20˚S-20˚N), limited to 21-27 km (50-20 hPa) due to sparse data at lower altitudes (Quaglia et al., 2023). The $R_{eff}$ is derived from a combination of extinction inversion algorithms that make use of the extinction ratios between 0.525 and 1.02 µm and assume that the aerosols are spheres with a sulfate composition of 75% $H_2SO_4$ + 25% $H_2O$ by mass (Damadeo et al., 2013).

### 3.5.2 UARS/SAGE-II (Stenchikov et al., 1998)

This work uses column average $R_{eff}$ derived from two Upper Atmosphere Research Satellite (UARS; Grainger et al., 1995; Lambert et al., 1997) instruments (the Improved Stratospheric Mesospheric Sounder (ISAMS) and Cryogenic Limb Array Etalon Spectrometer (CLAES)), and SAGE-II extinction data (Stenchikov et al., 1998). The UARS instruments ISAMS and CLAES are limb sounders, reporting aerosol extinction –12.11 µm and 12.66 µm wavelengths, respectively, across altitude profiles above 100 hPa (Lambert et al., 1997). Vertical resolution is approximately 2.5 km and horizontal resolution is about 4˚ (Taylor et al., 1994). These extinction values are then used to derive aerosol $R_{eff}$ assuming volcanic aerosol size distribution parameters from balloon borne measurements (Deshler et al., 1992, 1993) and a sulfate refractive index corresponding to 75% $H_2SO_4$ aerosol mass composition (Grainger et al., 1995; Lambert et al., 1997). Another limb sounder, the SAGE-II (McCormick et al., 1995), reports extinction at the 1.02 µm wavelengths which is then used to derive aerosol number density in the atmosphere. The height resolved number density and $R_{eff}$ are then used to calculate column average $R_{eff}$ zonal means at 40˚N and 7˚S from 200 hPa – 10 hPa, reported in Fig. 4 of Stenchikov et al. (1998).

### 3.5.3 WOPC

The balloon-borne University of Wyoming optical particle counter (WOPC; Deshler et al., 1993; Deshler, 1994, 2003) uses particle scattering of white light to measure particle counts across 8-12 channels that range in size from $0.15 – 2$ µm (Kalnajs and Deshler, 2022). These particle counts are then fit to unimodal or bimodal size distributions such that they minimize the root mean square error in number concentration between the measured cumulative count and the integral of the size distribution (Deshler, 2003). This instrument was launched from 1989-2013 from a variety of locations across the globe, with the most continuous sampling in Laramie, Wyoming. Instrument uncertainties include: ±10% in concentration and ±10% in aerosol radius, and ±40% in the distribution moments (e.g., surface area, volume, and extinction) (Deshler, 2003; Deshler et al., 2019). In this work we reproduce WOPC aerosol size distributions and derived $R_{eff}$ over Laramie, WY.

### 4. Results

This section traces model performance across scales (global to microphysical), tying the latter to the former through single particle optical properties. The global performance of E3SMv2-SPA stratospheric mass burden (Section 4.1) and AOD (Section 4.2) is determined through comparisons with remote sensing data. Climate impacts of Pinatubo are explored via TOA radiative flux (Section 4.3) and atmospheric temperature (Section 4.4) perturbations. Regional comparisons to remote and in-situ observations of stratospheric $R_{eff}$ (Section 4.5) identify a model bias toward smaller sizes, which is explored in more detail through analysis of aerosol size distributions (Section 4.6). Lastly, model data is compared to individual balloon-borne measurements of aerosol size distributions and $R_{eff}$, relating the effective single particle scattering (Section 4.7) and absorption (Section 4.8) efficiencies to changes in direct/diffuse radiation at the surface (Section 4.7.1) and longwave heating in the stratosphere (Section 4.8.1).

### 4.1 Sulfur burden

Figure 1 shows the stratospheric mass burden of the sulfur component of sulfate aerosol in the different model sensitivity tests, the HIRS observational dataset (Baran and Foot, 1994), and the SAGE-$3\lambda$ dataset (Revell et al., 2017). When compared to HIRS and SAGE-$3\lambda$, E3SMv2-SPA improves the modeled stratospheric aerosol burden over E3SMv2-PA, especially in the years following the Pinatubo eruption. The increased aerosol burden – and thus, aerosol lifetime – in the stratosphere is mainly due to our modifications to the coarse mode $\sigma_g$ in E3SMv2-SPA. While E3SMv2-PA reaches a similar peak in sulfate burden, the underestimated aerosol burden following Pinatubo in E3SMv2-PA is mainly caused by too wide an aerosol number distribution, causing fast sedimentation of the larger coarse mode particles in the upper tail of the distribution. The E3SMv2-SPA tends to overestimate aerosol burden compared to HIRS and SAGE-$3\lambda$ in the 6 months after Pinatubo but agrees well with the slow decay reported in observations during 1992. In the four months following Pinatubo, models agree best with HIRS over SAGE-$3\lambda$, likely due to saturation issues identified in SAGE-II limb-occulation data (Russell et al., 1996; Sukhodolov et al., 2018; Quaglia et al., 2023). From 1992 onward, stratospheric mass burden in E3SMv2-SPA agrees the best

with SAGE-3$\lambda$, which reports higher burdens in 1993 than HIRS. E3SMv2-SPA and WACCM are similar in atmosphere and aerosol treatments, but have very different atmospheric chemistry, which seems to impact lifetime.

In CESM2-WACCM, the interactive hydroxyl radical (OH) treatment causes OH depletion in the vicinity of the plume as the oxidation of SO$_2$ to form sulfate aerosol depletes available OH and therefore limits the reaction rate (Mills et al., 2017). This is in contrast to E3SMv2 which assumes an OH climatology unaffected by the oxidation of SO$_2$. The result is faster depletion of SO$_2$ and higher initial sulfate concentrations in the stratosphere in E3SMv2 (Fig. S5). The difference in OH treatment can be seen in Fig. 1, marked by the faster increase in stratospheric sulfate burden in E3SMv2-PA and E3SMv2-SPA. Another significant difference between the two chemical treatments is the presence of carbonyl sulfide (OCS) in CESM2-WACCM, which is a largely inert tropospheric chemical species that is oxidized and photolyzed when it enters the stratosphere, forming sulfate. This is a major contributor to non-volcanic stratospheric sulfate and will lead to a higher pre-Pinatubo sulfate concentrations in the stratosphere in CESM2-WACCM than in E3SMv2 (Mills et al., 2016). In Fig. 1, the effect of OCS is shown in larger pre-Pinatubo stratospheric burdens in CESM2-WACCM.

Figure 1 shows that E3SMv2-SPA performs reasonably well when forming sulfate from SO$_2$ and simulating increased aerosol lifetimes. The following sections will address how well the model parameterizes the aerosol microphysical properties and their impacts on the global radiative balance.

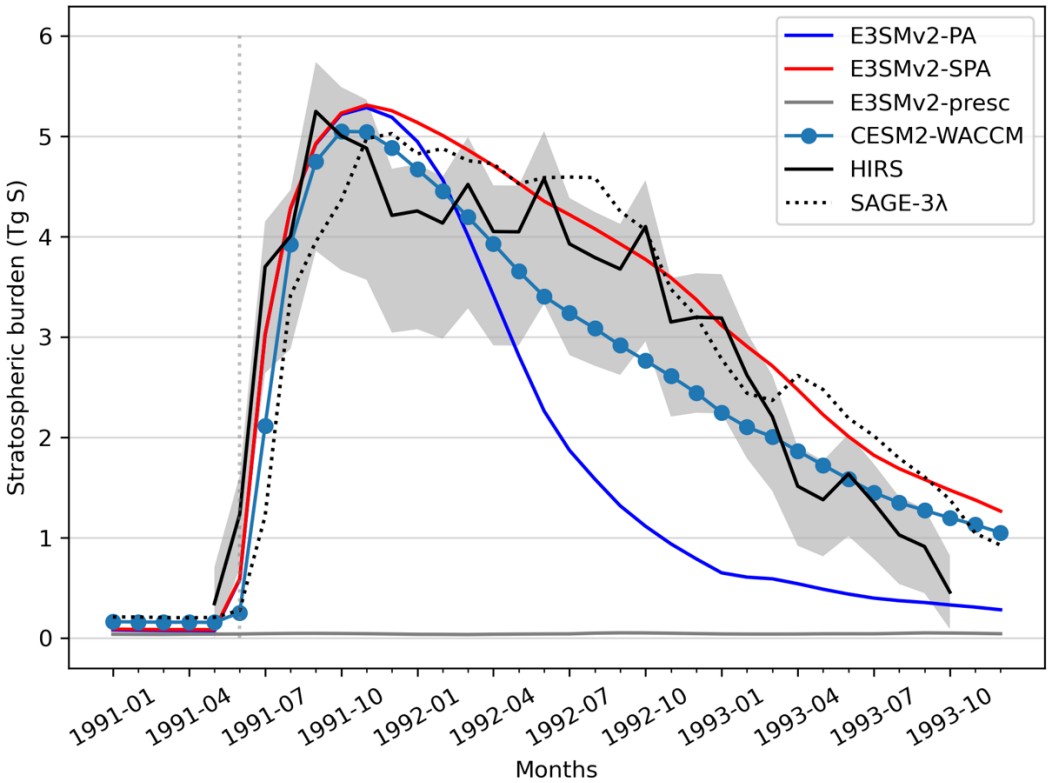

**Figure 1:** Stratospheric sulfate burden – reported in Tg of the sulfur mass contribution – for model simulations, as well as HIRS and SAGE-3λ remote sensing observations. The model data is processed to match the HIRS and SAGE-3λ data coverage of 80˚N – 80˚S above the model lapse rate tropopause height. The sulfur component is determined by scaling modeled sulfate mass by the ratio of sulfur and sulfate molecular weights (MW) such that $Tg\,S = Tg\,SO_4 * \frac{32.066\,g\,mole^{-1}}{MW\,Sulfate}$. In both E3SMv2 and CESM2-WACCM, sulfate density and MW are assumed to be ammonium bisulfate ($(NH_4)HSO_4$; density = 1.7 kg m$^{-3}$; MW= $115.11$ g mole$^{-1}$) (Liu et al., 2012, 2016; Mills et al., 2016). Gray shading around the HIRS data represents systematic error of ~10% (±1.4 Tg aerosol) and the minimum and maximum aerosol composition bounds (59%–77% $H_2SO_4$).

## 4.2 Aerosol optical depth

In Figure 2, the stratospheric contribution is isolated by subtracting the monthly mean AOD from pre-Pinatubo years, relying on the assumption that non-volcanic background aerosol in the atmosphere is similar in the near-term. Model data is normalized to 1990 monthly means, while AVHRR is normalized to the monthly means for the period June 1989 to May 1991, with the exception of missing data from October-December 1990. Normalized model data is masked to reflect the same temporal and spatial sampling as AVHRR data from 1991-1993 over the oceans only and between 60˚N – 60˚S. Here, AOD from the models is reported at 0.55 μm, while the AVHRR AOD is 0.6 μm.

The AVHRR AOD peaks two months after Pinatubo, linearly decreasing except for periods of flattening in the months March – July 1992 and January – July 1993. The flattening in 1992 is attributed to the continual growth of aerosols due to coagulation, which increases sulfate aerosol scattering efficiency (Russell et al., 1996; Stenchikov et al., 1998) even as aerosol burden continues to decay (Fig. 1). This is supported by a simulated increasing accumulation mode – and to a lesser extent, coarse mode – $D_g$ (Section 4.6; Fig. 7) and a slight increase in global average stratospheric aerosol $R_{eff}$ (Section 4.5; Fig. 6). The flattening in 1993 for both observations and models may be due in part to the influence of the smaller Chilean volcanic eruption Lascar (1993-01-30; 23.36˚S), which has a discernable impact on modeled global accumulation mode aerosol mean diameters >0.2 μm (Section 4.6; Fig. 7).

In the first year following Pinatubo, AVHRR and the volcanically parameterized models (i.e., E3SMv2-SPA, E3SMv2-presc, and CESM2-WACCM) follow a similar trend, decreasing and leveling off near the beginning of 1992. As for aerosol mass burden, the overly short stratospheric aerosol lifetime in E3SMv2 leads to a rapid decay in AOD. All models consistently underpredict AOD in the first year after the eruption but tend to overpredict AOD in the third year following the eruption. Underprediction may be due in part to a lack of volcanic ash and incorrect number/size representation in the models. Overprediction is likely a factor of aerosol removal assumptions. Of the models, CESM2-WACCM has the closest agreement with AVHRR in 1993, due to a faster decline in AOD than E3SMv2-SPA. This indicates that the comprehensive chemistry in CESM2-WACCM may better represent aerosol size distributions than E3SMv2-SPA (see Section 4.4).

A surprising finding is the close agreement between E3SMv2-SPA and E3SMv2-presc. These models utilize very different approaches: prognostic aerosol microphysics from emitted $SO_2$ in MAM4 (E3SMv2-SPA) and prescribed stratospheric aerosol extinction from a range of observations in GloSSAC (E3SMv2-presc). As noted in Quaglia et al. (2023), the dependence of GLoSSAC on SAGE-II data - which saturates at AOD ~0.15 - means that E3SMv2-presc may be underestimated in the months shortly after the eruption, suggesting that the close agreement between these datasets may be

partly coincidental. Regardless, their similarities indicate that the changes to aerosol microphysics in MAM4 reasonably recreate the reanalysis data product, especially from 1992 onward when instrument saturation is less of a concern.

Figure 3 shows a zonal average of the stratospheric AOD values described in Fig. 2. As for the 60˚S-60˚N average, AVHRR exhibits much higher maximum AOD values in the stratosphere (0.428) than all other models (E3SMv2-PA: 0.175, E3SMv2-SPA: 0.204, E3SMv2-presc: 0.152, CESM2-WACCM: 0.25). The stratospheric parameterized models (E3SMv2-SPA (Fig. 3b), CESM2-WACCM (Fig. 3d)) have the closest agreement with AVHRR in the tropical plume shortly after the eruption. E3SMv2-PA, E3SMv2-SPA, and CESM2-WACCM fail to simulate the tropical confinement of aerosol present in

AVHRR and E3SMv2-presc, simulating peak AOD that occurs further north and a weaker southern hemisphere AOD signal up to 1993 than observations and prescribed forcing datasets. Quaglia et al. (2023) identified this behavior in a range of models simulating the Pinatubo eruption, attributing this feature to model resolution, vertical wind structure, and the vertical distribution of the model volcanic cloud shortly after the eruption. Specifically, during the atmospheric conditions under which Pinatubo occurred, aerosols at levels $<\sim$20 km are transported north while aerosols at levels $>\sim$20 km are more effectively

confined to the tropics (McCormick and Veiga, 1992). While E3SMv2-PA, E3SMv2-SPA, and CESM2-WACCM have the same injection parameters, E3SMv2-SPA has slightly more southern transport than CESM2-WACCM, which may be related to higher concentrations of sulfate above 20 km than CESM2-WACCM (Fig. S6). We speculate that this disagreement may arise from interactive chemistry in CESM2-WACCM and its effect on sulfate nucleation, growth, and $SO_2$ lifetime given the similarities in aerosol transport (i.e, nudged atmospheric dynamics), vertical resolution aerosol microphysics, and injection

between the two models. E3SMv2-presc (Fig. 3c) has a noticeable low bias in AOD over the northern hemisphere over 1992, and we believe this is related to our use of an older version of the GLoSSAC dataset (Thomason et al., 2018) in which higher latitudes have a low bias in AOD attributed to linear interpolation of the SAGE-II data (Kovilakam et al., 2020). This may contribute to the close agreement between E3SMv2-SPA and E3SMv2-presc pointed out earlier, likely indicating a higher E3SMv2-presc signal with an updated GLoSSAC dataset. Lastly, when compared to other model simulations in Quaglia et al.

(2023), the volcanic Pinatubo signals in E3SMv2-SPA and CESM2-WACCM without land and AVHRR-missing-data masks (Fig. S7) show qualitatively similar patterns and magnitudes to ECHAM6-SALSA and ECHAM6-HAM with a comparable injection treatment (18-20 km; 7 Tg $SO_2$).

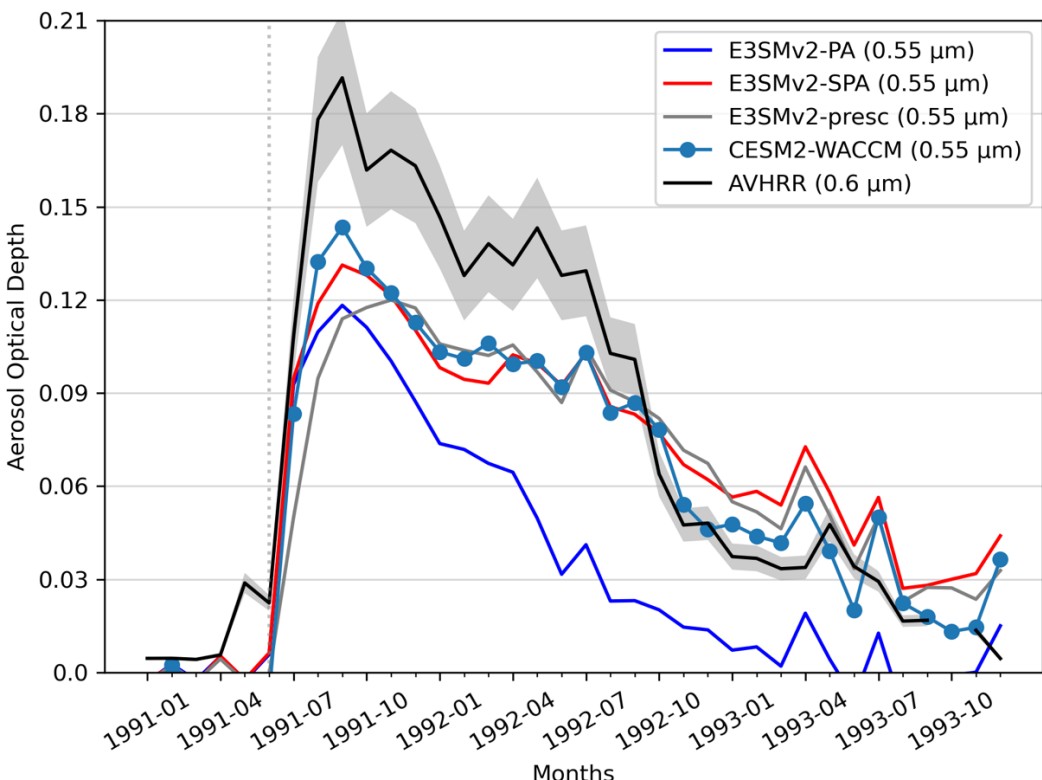

**Figure 2: Stratospheric aerosol optical depth (AOD) over the ocean and across latitudes 60˚S–60˚N from the model simulations and AVHRR. Models calculate AOD at 0.55 µm and AVHRR channel-1 AOD (0.63 µm) is processed to 0.6 µm. Both datasets are normalized to volcanically quiescent periods as described in the text. The Pinatubo eruption is marked with the gray dotted line at 1991-06. The gray shading indicates ±11.3% uncertainty in AVHRR AOD.**

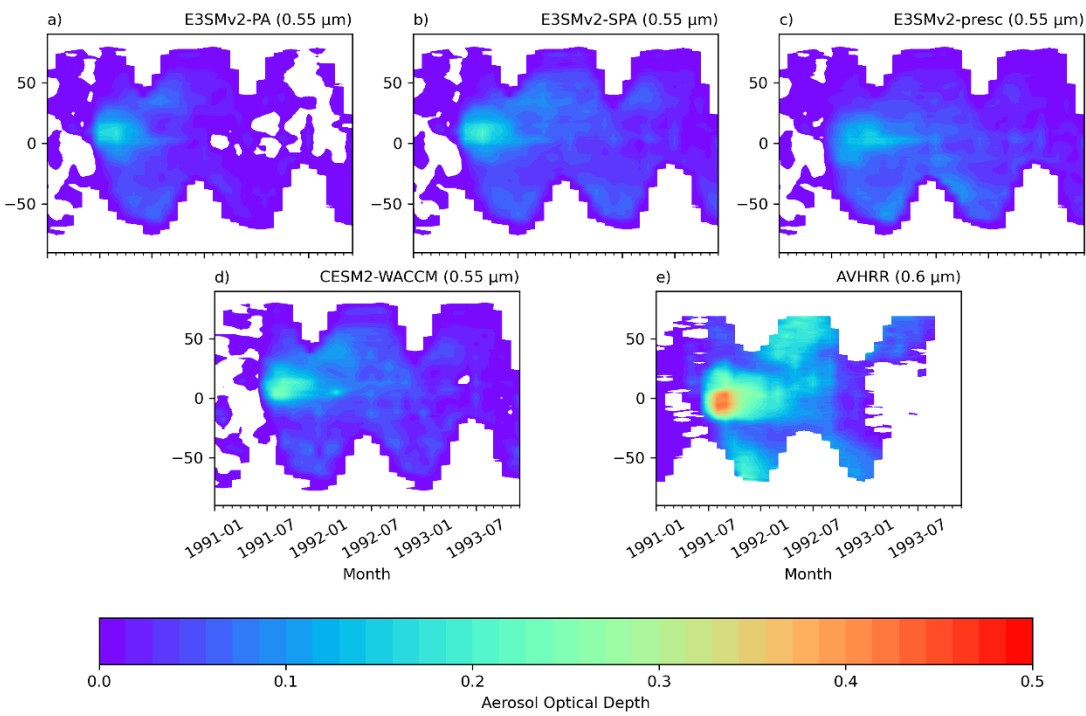

**540**

**Figure 3: Stratospheric aerosol optical depth (AOD) zonally averaged over the ocean and across latitudes 60°S–60°N from the model simulations and AVHRR. Models calculate AOD at 0.55 µm and AVHRR channel-1 AOD (0.63 µm) is processed to 0.6 µm. Both datasets are normalized to volcanically quiescent periods as described in the text.**

**545    4.3 TOA radiation flux**

Figure 4 compares the global TOA radiative flux from model simulations to the all-sky ERBS observations over 1991-1993, subtracting out corresponding monthly means from the pre-Pinatubo year 1990 (Note: this is a different non-volcanic period than used in previous publications (2001-2005 (Allan et al., 2014; Liu et al., 2015); 1999 (Mills et al., 2017)) which will result in differing magnitues for ERBS over 1991-1993). Model TOA flux is shown for all-sky (solid lines), clear-

sky (faint dashed lines), and aerosol impact only (faint dotted line) conditions. The radiative flux is reported as absorbed shortwave radiation (ASR, positive downward flux; Fig. 4a), outgoing longwave radiation (OLR, positive upward flux; Fig. 4b), and net radiative flux (NET, positive downward flux; Fig. 4c). In Fig. 4a, ASR shows the clearest model separation 3-4 months after Pinatubo corresponding with peak AOD (Fig. 2). There is close agreement between E3SMv2-SPA, E3SMv2-presc, and CESM2-WACCM during the year 1992 which corresponds to the largest sulfate particles during the Pinatubo plume

evolution (see Sections 4.5 and 4.6). The all-sky signal exhibits noise due to differences in atmospheric conditions (i.e., cloud cover, tropospheric aerosol) and surface albedo between the period of interest and our control year (1990). There is a clear seasonal increase in ASR in 1991/1992 and 1992/1993 Northern Hemisphere winters relative to Northern Hemisphere summer. When clear-sky (no influence from clouds) is compared to all-sky conditions in the models the seasonality disappears, implying

that the seasonality is cloud-related and cloud albedo was greater in Northern Hemisphere winter 1990 than Northern

Hemisphere winter 1991/1992 and 1992/1993. Even with noise introduced by non-Pinatubo factors, there is a distinct all-sky ASR signal in E3SMv2-SPA, CESM2-WACCM, and E3SMv2-presc that is improved compared to ERBS.

The all-sky OLR (Fig. 4b), which is affected both by aerosol absorption of infrared emissions from the earth's surface and the cooling of the troposphere and surface by the scattering of solar radiation, has a weaker response across these models than ASR. This is due in part to a less efficient absorption of outgoing longwave radiation than scattering of incoming solar

radiation, leading to a lower sensitivity of OLR to aerosol growth and evolution (see Section 4.8). The largest spread in model simulations occurs during 1992 when aerosols are at their largest (i.e., highest absorption efficiency of longwave radiation; Section 4.8) and the highest reduction in surface temperatures were observed (Parker et al., 1996). All-sky E3SMv2-SPA has the greatest reduction in OLR from April 1992 to the end of 1993, and overestimates the longwave flux reduction compared to ERBS. This corresponds with E3SMv2-SPA overestimation of global AOD values compared to AVHRR over this period

(Fig. 2). During this same period, CESM2-WACCM has slightly better agreement with ERBS, which may be related to the temperature nudging in this simulation which will modulate CESM2-WACCM surface temperature reduction and stratospheric temperature. When clear-sky OLR fluxes are compared, there is a weaker reduction in OLR for E3SMv2-PA, E3SMv2-SPA, and CESM2-WACCM, and nearly no change in E3SMv2-presc during 1992. Due to the lack of stratospheric aerosol in E3SMv2-presc, this appears to be evidence of volcanic influence on high altitude clouds which act to reduce OLR further

supporting conclusions from Liu and Penner (2002) and Wylie et al. (1994). Lastly, the aerosol-only model simulations remove the 1991/1992 and 1992/1993 wintertime peaks in the OLR signal, indicating similar or smaller OLR in 1990 than our period of interest due to cooler surface conditions.

The improvements in all-sky NET (Fig. 4c; solid lines) with volcanic parameterizations are less apparent across the models than in ASR (Fig. 4a), but do show improvement during the first 6 months after the eruption and during 1992.

Differences in cloud cover and surface conditions between our period of interest and 1990 introduce substantial noise to this comparison, but the removal of clouds (clear-sky) and the isolation of aerosol TOA forcing (aerosol only) show a clear separation of volcanic parameterizing models and E3SMv2-PA.

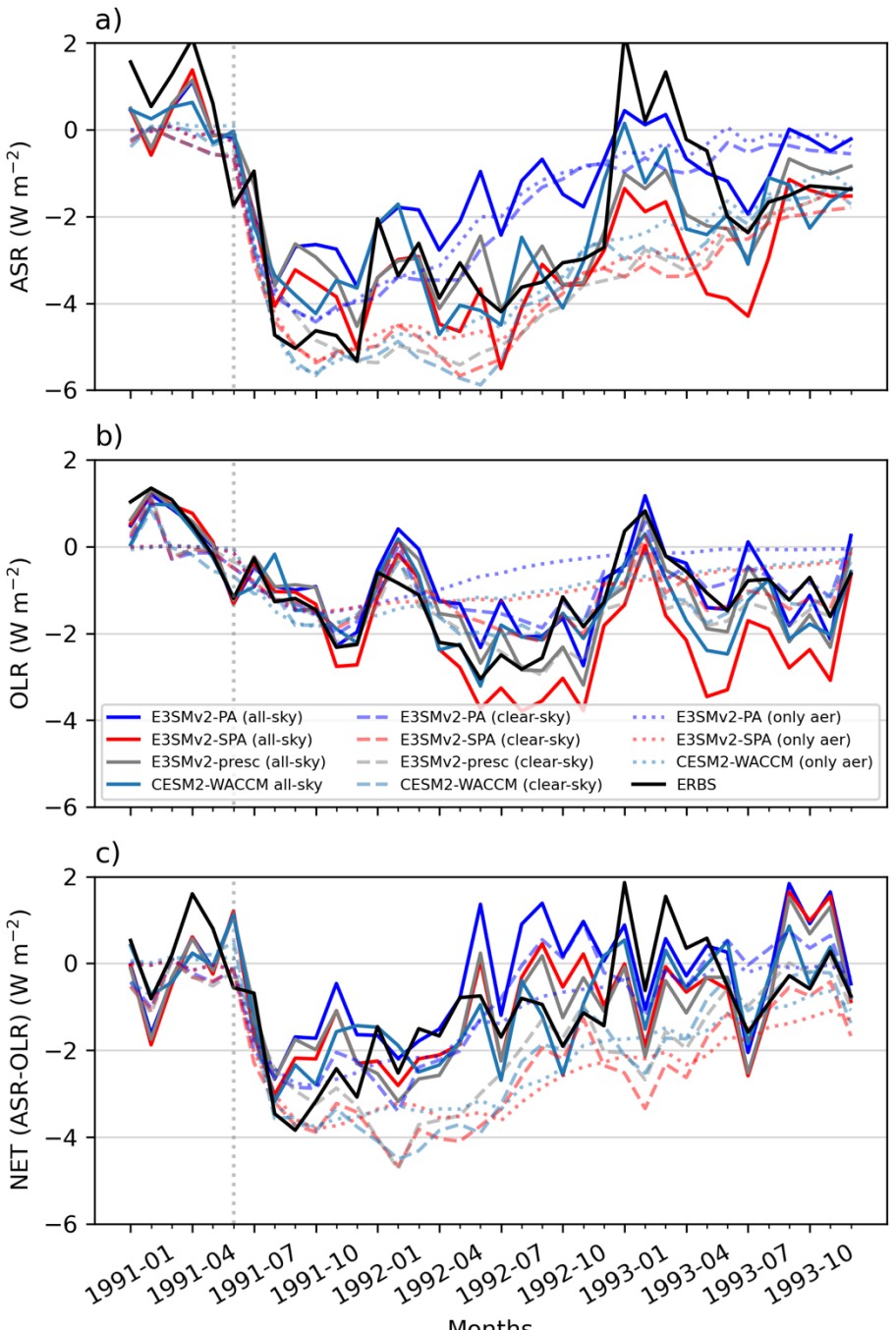

Figure 4: Top-of-atmosphere, radiative flux from model simulations and ERBS observations (Allan et al., 2014; Liu et al., 2015).
The panels describe: (a) absorbed solar radiation (ASR; positive downward flux); (b) outgoing longwave radiation (OLR; positive upward flux); and (c) net radiative flux (NET=ASR-OLR; positive downward flux). ERBS TOA flux is under all-sky conditions, while model TOA flux is shown under all-sky (solid line), clear-sky conditions (faint dashed line), and aerosol only (faint dotted line)

conditions. Monthly mean data is normalized to the pre-Pinatubo conditions by subtracting respective monthly means from the year 1990 and both datasets are averaged over the entire globe.


## 4.4 Atmospheric temperature profiles

The radiation interactions described in Section 4.3 will lead to changes in atmospheric temperature. Namely, a warming of the stratosphere due to aerosol absorption of outgoing longwave radiation and a cooling of the surface due to reflection and scattering of incoming solar radiation by the aerosol plume. Figure 5 shows the 1992 annual mean atmospheric temperature anomalies (subtracting the 1990 annual mean) in the models (Fig. 5a-d), MERRA-2 reanalysis data (Fig. 5e), and the RICH-obs radiosonde product (Fig. 5f). The year 1992 was chosen given the highest model spread in TOA flux (Fig. 4), peak modeled reduction in ASR (Fig. 4a) and reduction in OLR (Fig. 4b), and peak surface cooling (Parker et al., 1996) over this period. Models and observations share similar anomaly spatial patterns, with the exception of RICH-obs in the 60S-90S upper troposphere and near the tropical tropopause. Differences in RICH-obs may be related to temperature interpolation errors introduced in these remote regions due to fewer radiosonde datasets (Haimberger et al., 2012; Free and Lanzante, 2009). There is greater stratospheric warming in E3SMv2-SPA (Fig. 5b), E3SMv2-presc (Fig. 5c), and CESM2-WACCM (Fig. 5d) compared to E3SMv2-PA (Fig. 5a). Furthermore, there is an improvement in midlatitude warming at higher altitudes (i.e., 50 hPa and above) over E3SMv2-PA when comparing Fig. 5a-d to observations (Fig. 5f), reflecting the higher plume heights in these models (Fig. S6). CESM2-WACCM and MERRA-2 have very similar temperature magnitude and distribution, which is due to temperature nudging of CESM2-WACCM to the latter reanalysis dataset. There is not as obvious a surface cooling difference between E3SMv2-PA and other models and observations. All datasets show a large cooling signal in the northern troposphere that roughly corresponds with early-1992 max AOD between 30˚N and 50˚N (Fig. 3), but this cooling signal could be influenced by internal variability in the normalization year of 1990 (Section 4.3).

Table 2 shows 50 hPa and 850 hPa pressure level averages from Fig. 5. These comparisons represent stratospheric and near-surface changes in temperature, with the 850 hPa level chosen to accommodate the lowest pressure level in the RICH-obs data. These latitude-weighted averages range from 65˚S-65˚N to avoid missing data in the upper atmosphere and surface RICH-obs data (Fig. 5f). This comparison shows stratospheric warming that is overestimated E3SMv2-SPA (1.57˚ K) and underestimated in CESM2-WACCM (0.9 ˚K) compared to MERRA-2 reanalysis (0.89 ˚K) and previously reported estimates of ~1˚ K (Ramachandran et al., 2000). RICH-obs struggles to represent lower stratospheric warming either due to aforementioned sparcity of data and/or its low horizontal resolution (5˚) compared to models (1˚) and MERRA-2 (0.5˚). E3SMv2-presc shows a more than three times the stratospheric warming of MERRA-2, which is likely due to a known error converting CLAES infrared extinction to the SAGE-II and GloSSAC V1 reported 1020 nm extinction coefficient, resulting in an exaggeration of peak aerosol extinction (Kovilakam et al., 2020). The 850 hPa cooling in CESM2-WACCM (-0.33 ˚K) agrees best with MERRA-2 (-0.36 ˚K) and RICH-obs (-0.29±0.007 ˚K) anomalies, due in part to nudging of CESM2-WACCM

temperatures to MERRA-2. There is small improvement in E3SMv2-SPA (-0.23 ˚K) and E3SMv2-presc (-0.26 ˚K) compared to E3SMv2-PA (-0.22 ˚K), but it is unclear how much internal variability is influencing these values.

This comparison gives an all-sky snapshot of surface and stratospheric 1992 temperature anomalies due to Pinatubo. The 50 hPa show a clearer improvement in simulated temperature anomaly in E3SMv2-SPA and CESM2-WACCM than 850 hPa height due to the influence of interannual differences in internal variability (Section 4.3) and internal modes of variability 625  (e.g., ENSO; Santer et al., 2014) in the troposphere. The model trends in stratospheric and near-surface temperature changes are consistent with changes in OLR and ASR (Fig. 4), respectively. Temperature trends also tend to agree better with observations and reanalysis with stratospheric volcanic parameterizations (E3SMv2-SPA, CESM2-WACCM) and prescribed volcanic aerosol (E3SMv2-presc). The next sections explore the microphysical representation within the models and how this influences lifetime, AOD, TOA flux, and temperature.

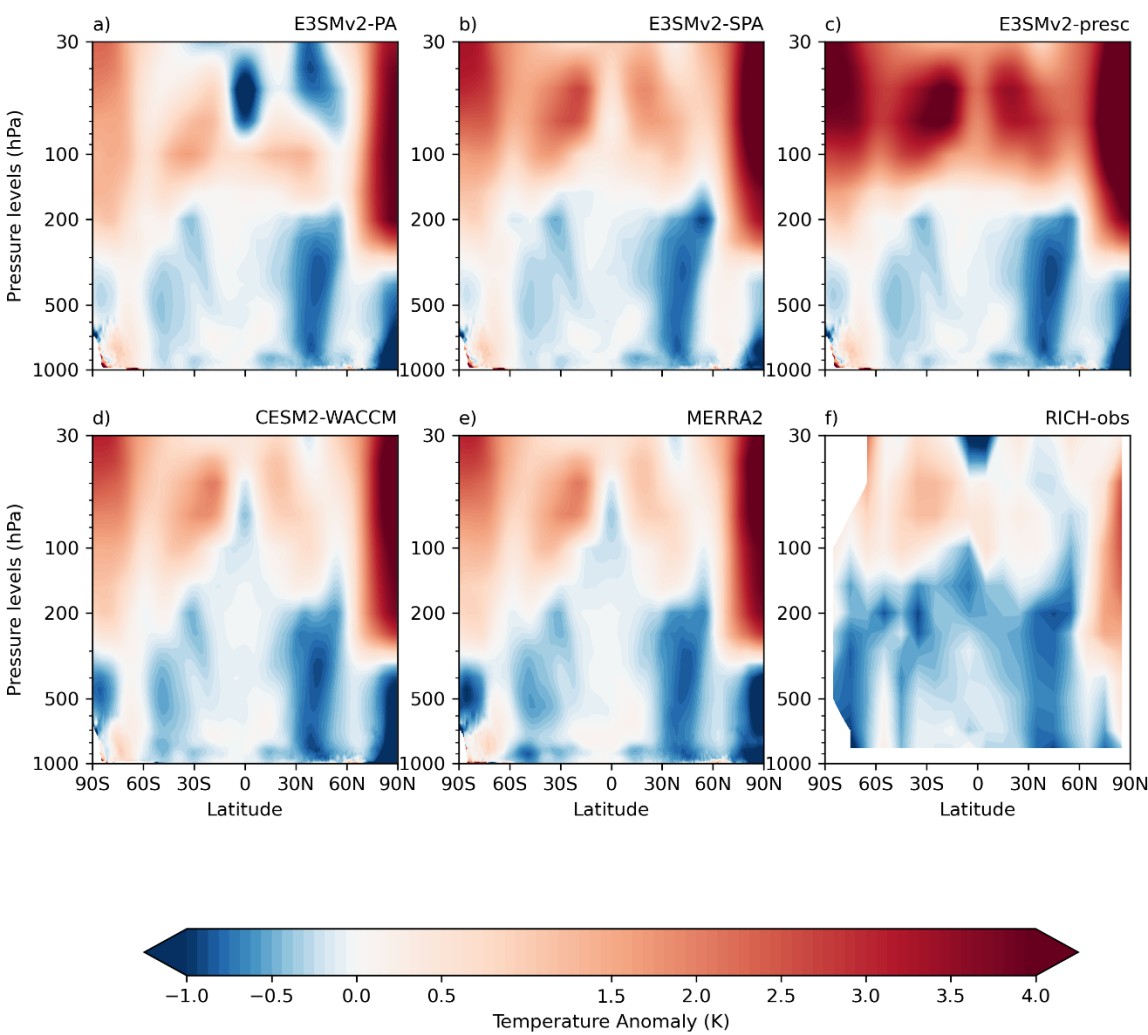

**Figure 5: Annual mean change in atmospheric temperatures (˚K) for the year 1992, shown for (a-d) model simulations,€) MERRA-2 reanalysis, and (f) RICH-obs radiosonde dataproduct. Anomalies are calculated by taking the difference between year 1992 and 1990 annual means. Model data is remapped from hybrid vertical coordinate to MERRA-2 pressure levels.**

**Table 2: Annual mean temperature anomalies at 50 hPa and 850 hPa levels, shown from model simulations, MERRA-2 reanalysis, and RICH-obs radiosonde data product. As for Fig. 5, anomalies are calculated as the difference between the year 1992 and 1990 annual means. Data is averaged over 65S-65N to avoid missing data in RICH-obs in the Antarctic. Included in the RICH-obs is one standard deviation about the 32-member ensemble spread.**

| | Pressure level | E3SMv2-PA | E3SMv2-SPA | E3SMv2-presc | CESM2-WACCM | MERRA-2 | RICH-obs |
|---|---|---|---|---|---|---|---|
| Temperature anomaly (˚K) | 50 hPa | -0.17 | 1.57 | 3.03 | 0.9 | 0.89 | 0.4±0.015 |
| | 850 hPa | -0.22 | -0.23 | -0.26 | -0.33 | -0.36 | -0.29±0.007 |

## 4.5 Stratospheric Effective Radius

$R_{eff}$ has frequently been used to characterize stratospheric aerosol properties, with stratospheric $R_{eff}$ of less than ~2 μm leading to a net solar radiation scattering effect and surface cooling (Lacis et al., 1992; McGraw et al., 2024). Based on a range of in-situ and remote sensing datasets, background stratospheric $R_{eff}$ is estimated at 0.17-0.19 μm, and following Pinatubo reaches average values of around 0.5 μm with observed values as large as 0.8 or 1.0 μm (Russell et al., 1996, and references therein). In the month following Pinatubo, there is little change in $R_{eff}$. This is due to a rapid increase in very small (i.e., Aitken mode sulfate) and very large (i.e., ash) aerosol particles following the eruption. Enhanced coagulation and condensation, coupled with low sedimentation rates, lead to a steady increase in $R_{eff}$ over the next 3-6 months. Aerosol growth continues until approximately mid-1992 when $R_{eff}$ peaks, lagging peak values in other metrics such as mass burden and AOD. The smaller magnitude eruptions of Hudson, Spurr, and Lascar also contribute to an increased $R_{eff}$ over this period, with more of an impact in near-source regions.

Figure 6a shows global $R_{eff}$, in addition to $R_{eff}$ from three different regional zones specific to different observational datasets: comparisons at 40˚N and 7˚S latitude bands and less than 100hPa with UARS (Fig. 6b), 41˚N 105˚W between 130-10hPa with WOPC (Fig. 6c), and 20˚S-20˚N between 50-20hPa with SAGE-II (Fig. 6d). Observations tend to measure a minimum size that falls in the middle of the Aitken mode in the model. To account for this characteristic of the data, the model $R_{eff}$ is an average of effective radii calculated with and without the Aitken mode, with the range between the two $R_{eff}$ calculations represented by shading about the line. Maximum differences tend to occur before Pinatubo, shortly after Pinatubo, and with other volcanic eruptions (e.g., Hudson (1991-08-08; 45.9˚S)). There is much larger spread between these two approaches in CESM2-WACCM in the Northern Hemisphere winter-spring, which may be due to enhanced stratosphere-tropopause exchange leading to higher concentrations of lower stratospheric aitken mode (Section 4.6 and Fig. 7). In Fig. 6a the models reproduce the expected background $R_{eff}$ of 0.17-0.19 μm, and the improvements to aerosol lifetime in E3SMv2-SPA and CESM2-WACCM can be seen in the slower decrease in $R_{eff}$ compared to E3SMv2-PA. There is also a nearly identical pattern in E3SMv2-SPA and CESM2-WACCM data, but with slightly higher $R_{eff}$ in CESM2-WACCM.

All of these models underestimate $R_{eff}$ compared to observations (Fig. 6b-d). In Fig. 6b, the finer temporal responses to the Pinatubo eruption in the UARS data are less apparent in the model, namely the large peak in $R_{eff}$ at 7˚S associated with short-lived volcanic ash in the stratosphere and the delayed peak at 40˚N and 7˚S due to particle aggregation (Stenchikov et al., 1998). The models neglect volcanic ash contributions, explaining the more gradual particle growth at 7˚S. While the models

don't show the same sensitivity to increases in $R_{eff}$ at these latitude bands – possibly due to higher vertical and horizontal spatial scale in the number concentrations from SAGE-II observations – E3SMv2-SPA does show a flattening of the 7˚S $R_{eff}$ akin to the UARS estimate. This corresponds to the eruption of Hudson in Chile, and the resulting high influx of smaller, Aitken mode particles into the southern stratosphere which drives down $R_{eff}$. The sensitivity to this eruption in E3SMv2-SPA may be due to higher Aitken mode production in this model than in CESM2-WACCM (see Section 4.7).

The tropical regions tend to have a larger $R_{eff}$ than the midlatitudes in the simulations. This is true of UARS regions 6-12 months after the eruption and also the SAGE-II (tropics; Fig. 6d) data. These larger $R_{eff}$ persist in the tropics a year or more after the eruption. While all models exhibit similar modal aerosol diameters (Fig. S8), the higher $R_{eff}$ is correlated with higher number concentrations in all aerosol modes and a slower decrease in aerosol number (i.e., reduced sedimentation) (Fig. S9). The reduced removal is likely due to higher initial concentrations of $SO_2$ in the volcanic plume over the tropics (Fig. S10)

contributing to more rapid local aerosol growth and a net positive aerosol production. Furthermore, the presence of the upwelling branch of the Brewer-Dobson circulation in this latitude band may help suspend larger aerosol species, slowing aerosol sedimentation rates and increasing their lifetime. The eruption of the Lascar volcano in northern Chile around February of 1993 also contributes to a bump in $SO_2$, as well as Aitken and accumulation mode aerosol number at the 7˚S band (Fig. S9, S10).

When comparing the models to WOPC (Fig. 6c) and SAGE-II (Fig. 6d), E3SMv2-PA has the closest agreement to these datasets in its initial aerosol growth. This growth is more rapid than the other models and leads to a peak in $R_{eff}$ that, while being closer to observed values, drops off precipitously. The $R_{eff}$ in E3SMv2-SPA and CESM2-WACCM have the best agreement with observational values and decay rate a year or more after Pinatubo. Differences across the models are due to the different microphysical assumptions, which can be explored by looking at aerosol size distributions.

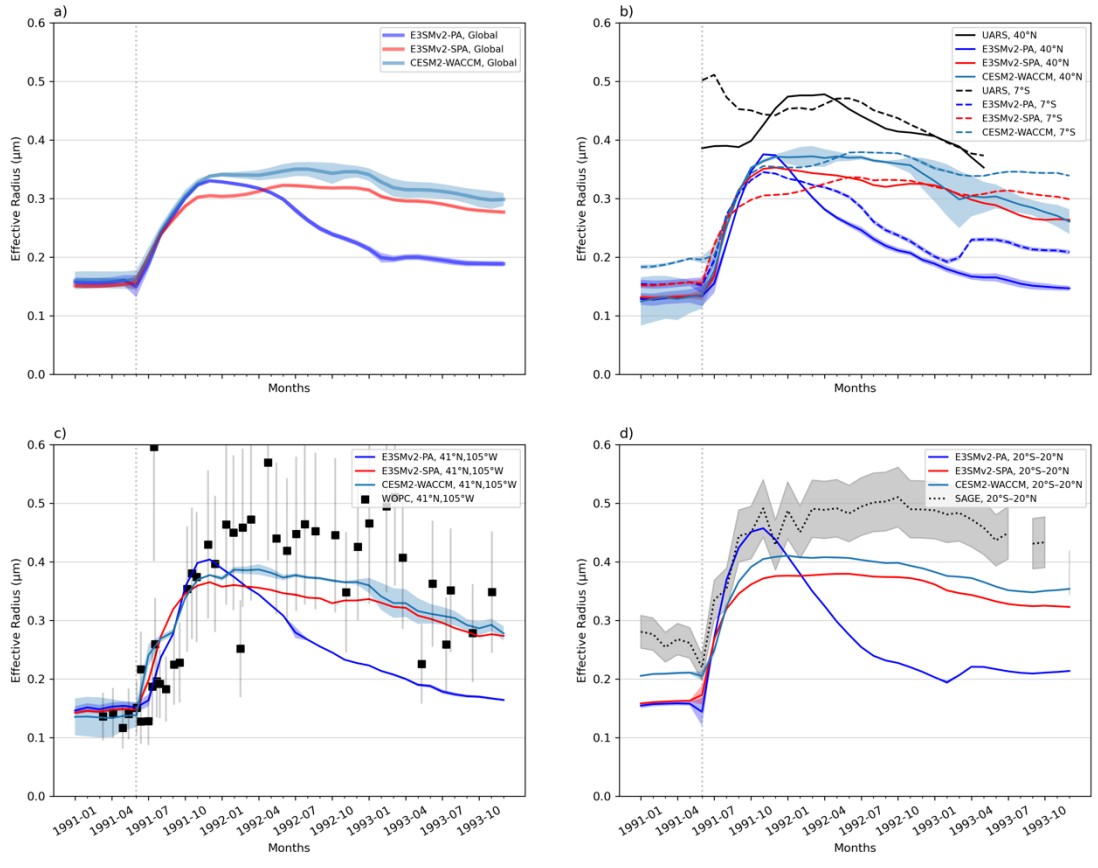

**Figure 6: Stratospheric aerosol effective radius (R$_{eff}$) averaged (a) globally above model tropopause, (b) at 40°N and 7°S and <100 hPa to compare with UARS data (Stenchikov et al., 1998), (c) over Laramie, Wyoming (41°N, 105°W) and 130-10 hPa to match WOPC data (Quaglia et al., 2022), and (d) over 20°S-20°N and 50-20 hPa to match SAGE-II observations (Quaglia et al., 2022). The shaded range in models represents R$_{eff}$ with and without the Aitken mode, and the model line is an average of the two. Error bars in WOPC data assume a 40% uncertainty and correlation coefficients of 0.5 between aerosol moments and levels (see Appendix A2 in Quaglia et al. (2023)). The Pinatubo eruption is marked with the gray dotted line at 1991-06.**

### 4.6 Aerosol size distributions

While R$_{eff}$ is a good representation of aerosol size in the context of optical properties, it is not always a good indicator for behavior of aerosol microphysical processes. For example, an increase in accumulation mode particle number and a decrease in coarse mode particle number could manifest as an unchanging R$_{eff}$. An examination of the aerosol size distribution can be more informative when understanding how aerosol chemistry and the MAM4 microphysics contribute to model performance.

Figure 7 shows the globally averaged, stratospheric, aerosol distributions from the three models used in this study and their evolution from 1991 to the end of 1993. The contour fill represents aerosol number (dN/dlogD; cm$^{-3}$) with dashed and dotted lines indicating modal D$_g$ (μm). In all three models, there is growth in Aitken mode due to new sulfate particle formation following major and minor volcanic eruptions during this period: Pinatubo (1991-06; vertical dotted line), Hudson

(1991-08), Spurr in the Aleutian Islands (1992-06-27; 61.3°N), and Lascar (1993-02). In E3SM (Fig. 7a-b), the prescribed concentrations of OH lead to a rapid oxidation of available $SO_2$ and higher concentrations of Aitken mode aerosol compared to CESM2-WACCM (Fig. 7c), resulting in higher aerosol number concentrations in E3SM. It also appears that CESM2-WACCM has higher tropospheric aerosol transport into the stratosphere based on a higher concentration of Aitken mode aerosol across the period of interest. CESM-WACCM has seasonal peak concentrations occurring asynchronous with volcanic eruptions and corresponding to northern hemisphere winter (e.g., 1991 and 1992). The increased Aitken mode number concentration is also seen at 40°N (Fig. S9). These peaks are attributed to a lower tropopause in Northern Hemisphere winter (i.e., enhanced troposphere-stratosphere exchange) and may also be due to the inclusion of OCS in CESM-WACCM.

The larger volcanic eruptions, Pinatubo and Hudson, inject enough $SO_2$ into the stratosphere that Aitken mode aerosols grow through condensation and coagulation into the accumulation mode, which then grow through coagulation into the coarse mode. The exception to this is in E3SMv2-PA which lacks the ability to transfer sulfate mass into the coarse mode and so retains a roughly constant coarse mode $D_g$ derived from trace mass ($10^{-22}$-$10^{-21}$ kg cm$^{-3}$; global average) and number ($10^{-5}$-$10^{-4}$ cm$^{-3}$; global average) concentrations of dust, seasalt, and sulfate aerosol advected from the troposphere (Note: mass and number are related to aerosol size through Equations S1 and S2 in the Supp. Material). Aerosol growth through the modes can be seen in increased number and an increasing trend in aerosol size. Dips in modal $D_g$ correspond to sudden increases in aerosol number (e.g., Aitken mode nucleation from freshly injected $SO_2$, transfer of a large aerosol number from Aitken to accumulation mode) while mass remains relatively unchanged. This leads to a division of mass across a larger number of aerosols and a subsequent decrease in $D_g$.

Overall, the aerosol modal diameters are similar across the three models, but slight differences exist. There are slightly larger accumulation mode and coarse mode $D_g$ in CESM2-WACCM (0.262 μm; 0.843 μm) than E3SMv2-SPA (0.259 μm; 0.749 μm). This could be a factor of the interactive chemistry in CESM2-WACCM, where lower nucleation rates lead to fewer Aitken mode aerosol that initially grow faster through condensation as the longer lived $SO_2$ in the stratosphere condenses on preexisting particles. Contrast this with E3SM where high nucleation rates lead to more numerous, smaller aerosol that consume the available $SO_2$ more quickly. The E3SMv2-PA model accumulation mode $D_g$ reaches a higher maximum $D_g$ (0.285 μm) than E3SMv2-SPA or CESM2-WACCM (~0.26 μm). This is due to the missing coarse mode treatment in this model coupled with a larger $D_{g,high}$ (Section 2.1.2). The result is an accumulation mode that grows to $D_{g,high}$ following Pinatubo, whereby number is increased to maintain this size instead of transferring mass and number to a larger mode. This also explains the better initial agreement in $R_{eff}$ between E3SMv2-PA and observations in Fig. 6b-d.

Figure 8 compares modeled aerosol size distributions to in-situ measurements from WOPC, tracking the evolution of a slice of the plume from pre-Pinatubo (1991-04-19) to the end of 1993 (1993-11-16). Daily samples taken with the WOPC over Laramie, WY at a single level (18 km; roughly corresponding to the peak plume $R_{eff}$ across the period (Fig. S11)) are used to validate daily data from E3SMv2-PA and E3SMv2-SPA. Modal $D_g$ (vertical lines) and dN/dlogD are denoted by dotted and dashed lines, as in Fig. 7. Effective diameter ($D_{eff}$) for models and WOPC are included to relate the changing distributions to aerosol-light interactions.

What follows is a breakdown of Fig. 8 into a rough timeline evolution in aerosol size distributions and $D_{eff}$:

- Pre-Pinatubo conditions (Fig. 8a) are very similar between E3SMv2-PA and E3SMv2-SPA, and the $D_{eff}$ are nearly identical between model and observations. CESM2-WACCM has higher background number concentrations than E3SM.

- One month after Pinatubo (Fig. 8b), all datasets exhibit rapid growth in the smaller diameter aerosols, with some growth in the modeled accumulation mode but little sign of the coarse mode in E3SMv2-SPA. CESM2-WACCM has a massive increase in Aitken and accumulation mode number, possibly due to transport of $SO_2$ and Aitken mode aerosol into the region from the tropics. This Aitken mode peak is also seen at 17 and 19 km levels (Fig. S12-S13) resulting in a decrease in $R_{eff}$ over this height range (Fig. S11), and may be related to lower altitude sulfate aerosol in CESM2-WACCM (<20 km; Fig. S6) being more effectively transported into northern latitudes (McCormick and Veiga, 1992). The $D_{eff}$ are still similar across the different datasets, with the exception of CESM2-WACCM which is smaller due to the Aitken mode influence.

- Six months after the eruption (Fig. 8c), a clear coarse mode signal emerges in WOPC, E3SMv2-SPA, and CESM2-WACCM along with a sharp increase in modeled accumulation mode number. Models and WOPC are all comparable in their bimodal $D_g$ and accumulation mode number, while WOPC coarse mode has lower number than E3SMv2-SPA and CESM2-WACCM. E3SMv2-PA has a larger $D_{eff}$ than other datasets (though it is still within the uncertainty of WOPC $D_{eff}$). This peak in $D_{eff}$ – which is also noted in $R_{eff}$ in Fig. 6 – is attributed to a wider accumulation mode and larger modal $D_g$ in E3SMv2-PA. CESM2-WACCM has larger modal number than all other datasets.

- 11 months after the eruption (Fig. 8d), the WOPC data continues to grow in accumulation and coarse mode. Accumulation mode number is decreasing while coarse mode is increasing in E3SMv2-SPA, suggesting conversion of accumulation mass into the longer-lived coarse mode. The WOPC coarse mode $D_g$ (1.92±0.19 µm) is more than twice that of E3SMv2-SPA (0.828 µm), while the WOPC accumulation mode best-fit ($D_g$=0.909±0.09 µm, $s_g$=1.23) is close to E3SMv2-SPA coarse mode ($D_g$=0.828 µm, $s_g$=1.2) and is nearly identical to the CESM2-WACCM coarse mode ($D_g$= 0.918 µm, $s_g$=1.2). . The overall larger aerosols in the WOPC distributions lead to largest difference in $D_{eff}$ between model and WOPC.

- For one year and longer after the eruption, accumulation mode in E3SMv2-PA is on a steady and rapid decay that can be tracked in size and number (Fig. 8d-i). Over this period, E3SMv2-SPA and CESM2-WACCM generate very similar size distributions, with slightly lower coarse mode number and slightly larger coarse $D_g$ in CESM2-WACCM. Overall, WOPC and E3SMv2-SPA/CESM2-WACCM have similar distributions and similar $D_{eff}$, with E3SMv2-SPA showing slightly smaller $D_{eff}$ but still within the uncertainty of WOPC data. A peak in Aitken mode aerosol in the models in Fig. 8h may be due to the Lascar eruption in 1993-02.

The previous timeline shows an improved aerosol evolution in E3SMv2-SPA compared to WOPC and exhibits the similarities between CESM2-WACCM and E3SMv2-SPA. It also indicates that E3SMv2-SPA doesn't simulate large enough

aerosol compared to the observations. Similar behavior in $D_{eff}$ and relative size bias is noted at 17 km, 19 km, and 20 km levels as well (Fig. S12-S13). Given that the model is nudged to meteorology and that the aerosol in the stratosphere tend to evolve relatively slowly over time, it is assumed that the resolution artifacts of comparing a 1˚x1˚ model grid cell to in-situ measurements are minimal. We present a couple of possibilities for this size bias. One is that the aerosol density is too large due to the model assumption of ammonium bisulfate density of 1.7 kg m$^{-3}$ instead of the sulfuric acid density of ~1.6 kg m$^{-3}$ (Seinfeld and Pandis, 2006). This may bias the aerosol small while also contributing to more rapid removal of coarse mode aerosol. Another is the lack of van der Waals forces in both E3SMv2 and CESM2-WACCM. This intermolecular attraction has been shown to aid in the coagulation of smaller particles, enhancing $R_{eff}$ (English et al., 2013). The impact of this size bias on aerosol climate impact is now explored through effective single particle scattering.

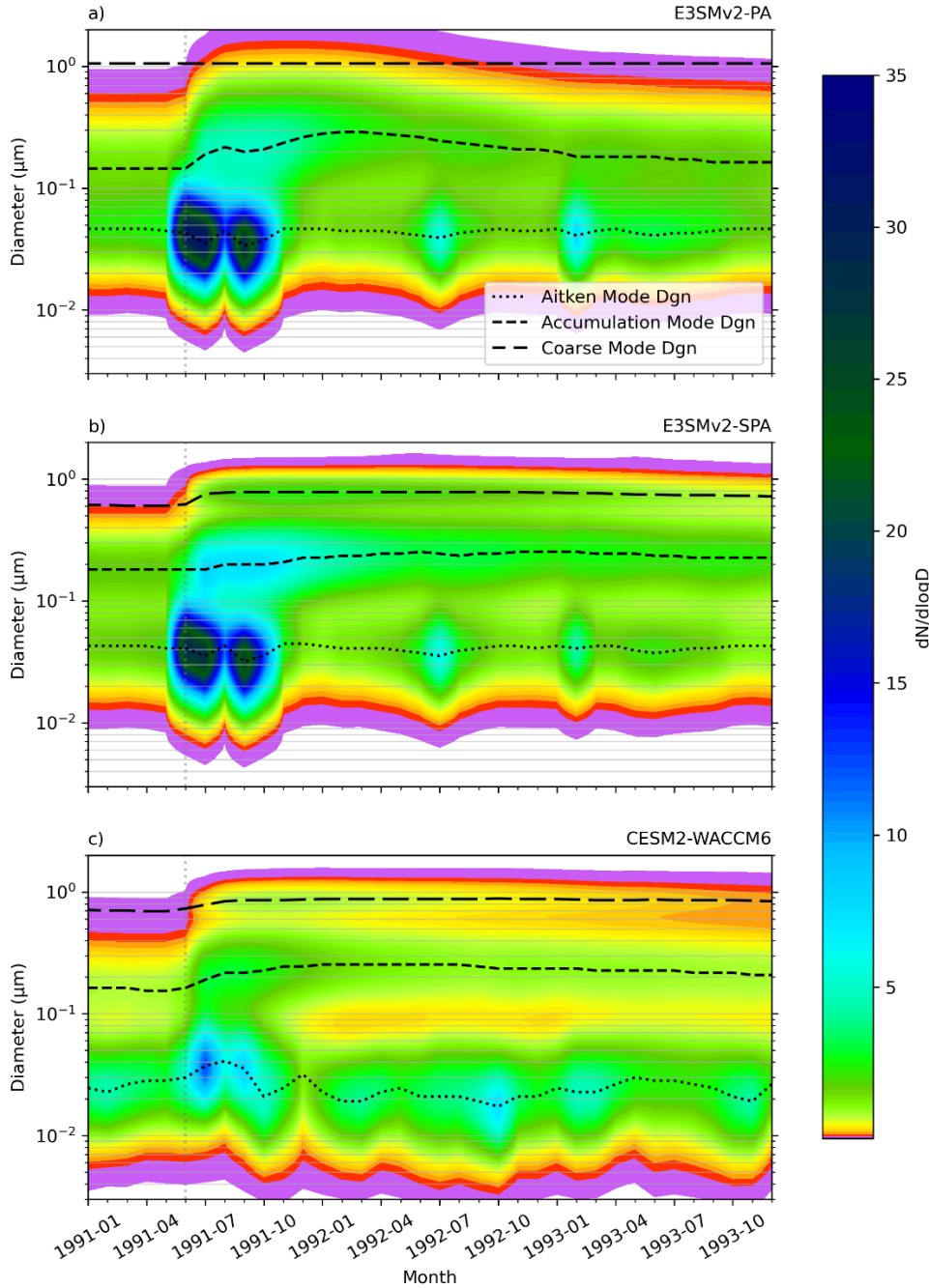

**Figure 7: Globally averaged, stratospheric lognormal aerosol size distributions from 1991-1993 for E3SMv2-PA, E3SMv2-SPA, and CESM2-WACCM. The dotted, dashed, and long-dashed lines indicate geometric mean diameters of the Aitken, Accumulation, and Coarse aerosol modes, respectively. The color contour is lognormally scaled. The Pinatubo eruption is marked with the gray dotted line at 1991-06.**

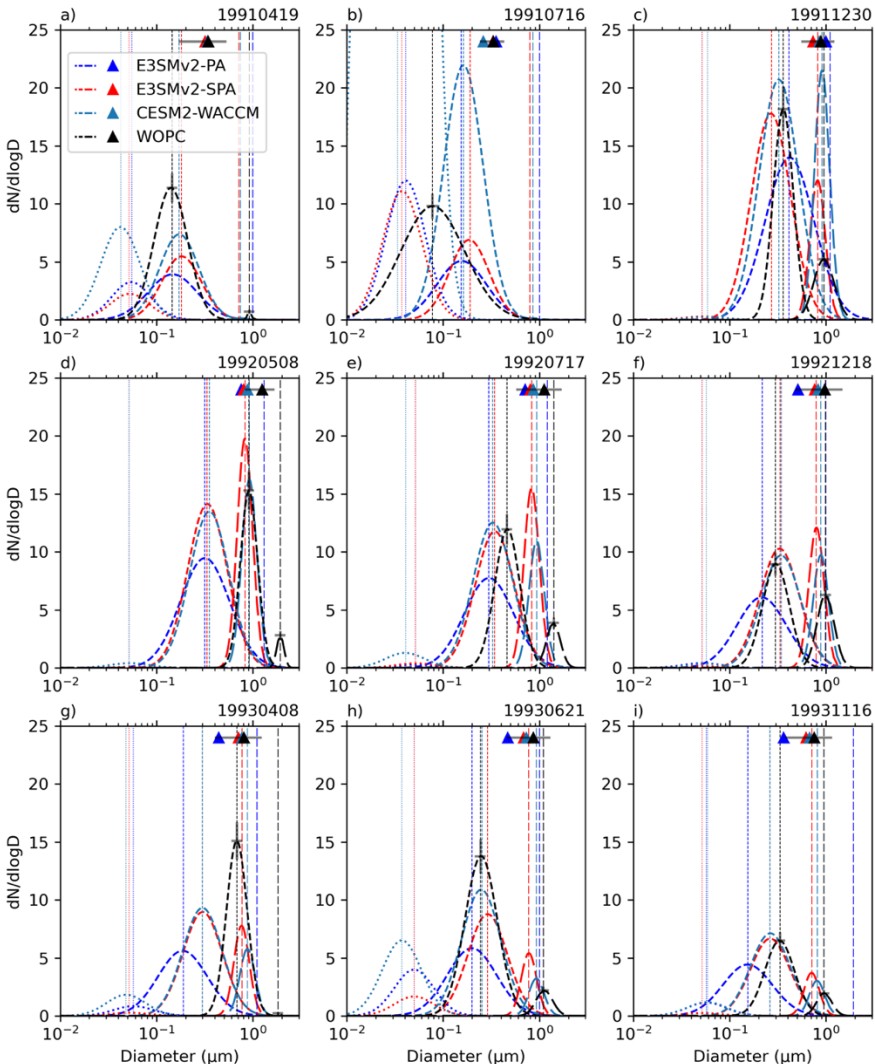

**Figure 8: Stratospheric aerosol size distributions from E3SMv2-PA (blue), E3SMv2-SPA (red), and WOPC (black) from 1991-1993. WOPC launches chosen to correspond to spring, summer, and winter measurements for each year, and samples are taken from the 18 km measurements and matched to the nearest model height and grid cell over Laramie, WY (41.3°N, 105°W). The dotted, dashed, and long-dashed lines indicate geometric mean diameters (vertical) and dN/dlogD size modes (curves) of the Aitken, Accumulation,**

 and Coarse aerosol modes, respectively. Triangles denote effective diameters ($D_{eff}$) derived from the size distributions. Uncertainties in WOPC diameter, number, and $D_{eff}$ are denoted by grey bars at the peak of the distributions and on the markers. Note: anomalously large max number in CESM2-WACCM Aitken mode (376.5 cm$^{-1}$) is out of the plotting bounds.

### 4.7 Single particle scattering efficiency

The $R_{eff}$ can be used to characterize the evolving aerosol size distribution, which conveniently can be used in the offline calculation of single particle optical properties using Mie theory (see Appendix B). We choose to use offline Mie code for both model and observations to allow for a direct comparison between optical properties derived from modelled and observed $R_{eff}$. Figure 9 shows the scattering efficiency ($Q_s$) of sulfate particles of size $R_{eff}$ for the same time and height samples as Fig. 8. Additionally, this plot marks the approximate wavelength of peak solar black-body irradiance ($\lambda_{solar}$=0.5 µm) to identify when aerosol scattering of solar radiation has the largest impact on radiative balance. $Q_s$ is proportional to the effective size parameter ($x_{eff} = 2\pi R_{eff}/\lambda$) to the fourth power ($x_{eff}^4$) (i.e., $\lambda^{-4}$) (Petty, 2006). This scattering feature can be seen in Fig. 9 where, as wavelengths increase, you will get a rapid decrease in $Q_s$.

The peak in $Q_s$ shifts from $\lambda$ of ~0.2 µm before and shortly after Pinatubo (Fig. 9a-b), to 0.5-0.7 µm 6 months after the eruption (Fig. 9c) with the onset of rapid particle growth. For the remainder of the time (Fig. 9d-i), E3SMv2-SPA and CESM2-WACCM $R_{eff}$ result in a $Q_s$ peak right around $\lambda_{solar}$, indicating a very efficient scattering of sunlight by these simulated aerosols. The WOPC reports larger $R_{eff}$ than the models, which have a $Q_s$ that is 10-80% higher than observations at $\lambda_{solar}$ during 1992 (Fig. 9d-f), becoming more similar during 1993 with the decay of stratospheric $R_{eff}$ (Fig. 9g-i). E3SMv2-PA has a similar $Q_s$ to E3SMv2-SPA/CESM2-WACCM until the end of 1992 (Fig. 9f) when there is a quick progressive drop in $R_{eff}$ due to rapid stratospheric aerosol deposition.

These results indicate an optimal sulfate scattering $R_{eff}$ of ~0.3-0.4 µm, which is similar to estimates of the optimal sulfate aerosol size of 0.3 µm in geoengineering applications (Dykema et al., 2016). Given that WOPC predicts larger $R_{eff}$ and smaller $Q_s$ at $\lambda_{solar}$ 6-18 months after Pinatubo, our results suggest that E3SMv2-SPA/CESM2-WACCM overestimates aerosol scattering and the global cooling effect of the Pinatubo aerosol at this level. This is supported by Fig. 4a, where both E3SMv2-SPA and CESM2-WACCM scatter more strongly than ERBS and E3SMv2-presc from April-August 1992 (though differences in clouds may also be contributing to this result). Temperature comparisons at 850 hPa over 1992 do not reflect a cooler surface in E3SMv2-SPA and CESM2-WACCM compared to observations (Table 2), but this comparison isn't ideal for making a connection between perturbations in scattering and surface temperature due to the unknown role of internal variability on temperature. Figure 9 relates size distribution differences in models and WOPC to climate impact at a single level. For a better validation of the stratospheric plume and to see if the model disagreement changes when looking across multiple levels, the next step is to calculate $Q_s$ from the stratospheric $R_{eff}$.

Figure 10 shows monthly $Q_s$ generated from averaged stratospheric $R_{eff}$ at a midlatitude site (solid line) (Fig. 6c) and the tropics (dotted line) (Fig. 6d). As in Fig. 9, a similar pattern is shown where smaller modeled $R_{eff}$ lead to higher modeled $Q_s$ at $\lambda_{solar}$ than observed $Q_s$. However, the differences between modeled and observed $Q_s$ at $\lambda_{solar}$ are not as stark for the

monthly, stratospheric average values, with max differences on the order of 5-14% as opposed to the 10-80% seen in the daily, 18 km level data. The larger $R_{eff}$ in the tropics leads to a higher $Q_s$ at $\lambda_{solar}$ one month after Pinatubo (Fig. 10b), and lower $Q_s$ at $\lambda_{solar}$ 18 months after Pinatubo (Fig. 10f), than midlatitude measurements in the stratosphere. The E3SMv2-SPA and CESM-WACCM maximum $Q_s$ from both regions hover around $\lambda_{solar}$, with higher modeled $Q_s$ at $\lambda_{solar}$ in the tropics 2 years after the eruption (Fig. 10h) due to a consistently larger $R_{eff}$ in this region.

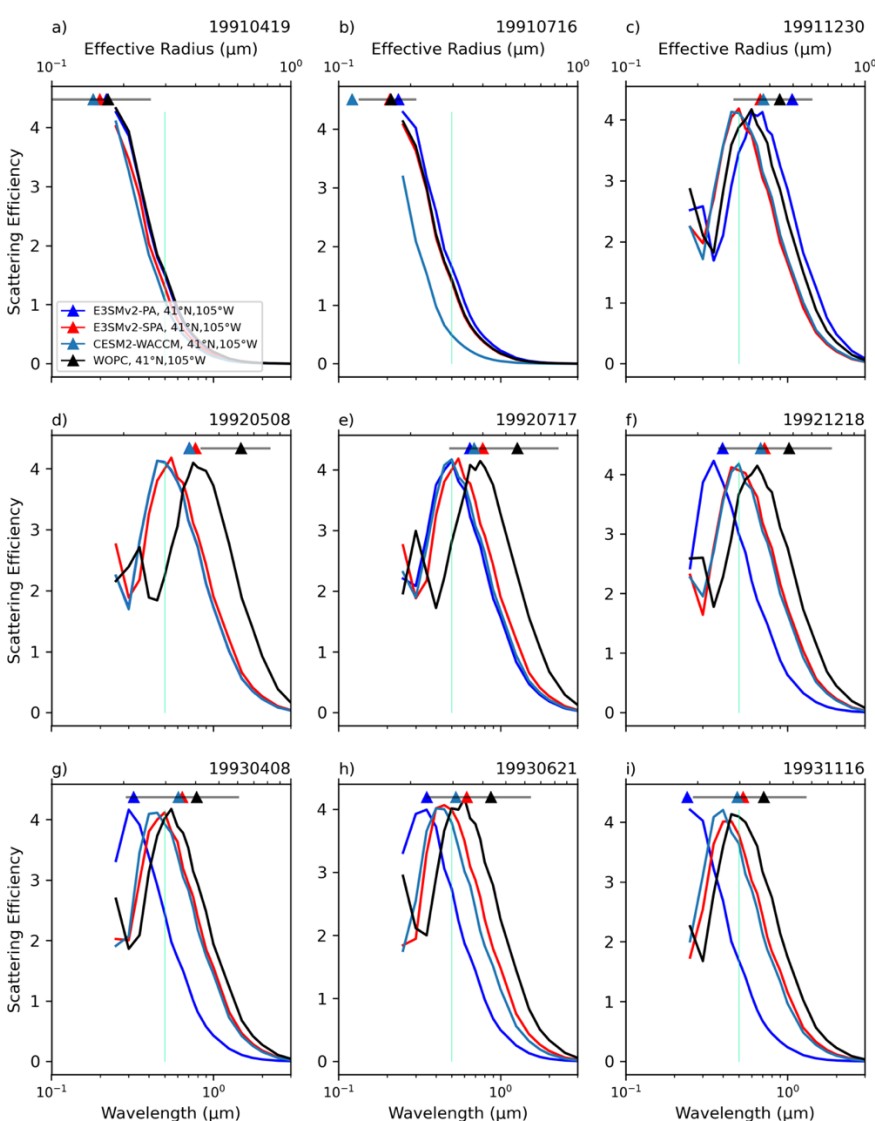

**Figure 9: Daily 18 km scattering efficiency ($Q_s$) using effective size parameter ($x_{eff} = 2\pi R_{eff}/\lambda$) and sulfate refractive index at 0% relative humidity (Hess et al., 1998). Effective radius ($R_{eff}$) from E3SMv2-PA, E3SMv2-SPA, and WOPC are calculated at 18 km, and are marked by colored triangles at the top of the plot. The bottom x-axis is wavelength ($\lambda$; 0.1–3 µm). The top axis is $R_{eff}$ (0.1-1 µm) used in the calculation of $Q_s$ via the $x_{eff}$. The turquoise vertical line marks the solar black body wavelength of maximum irradiance (0.5 µm).**

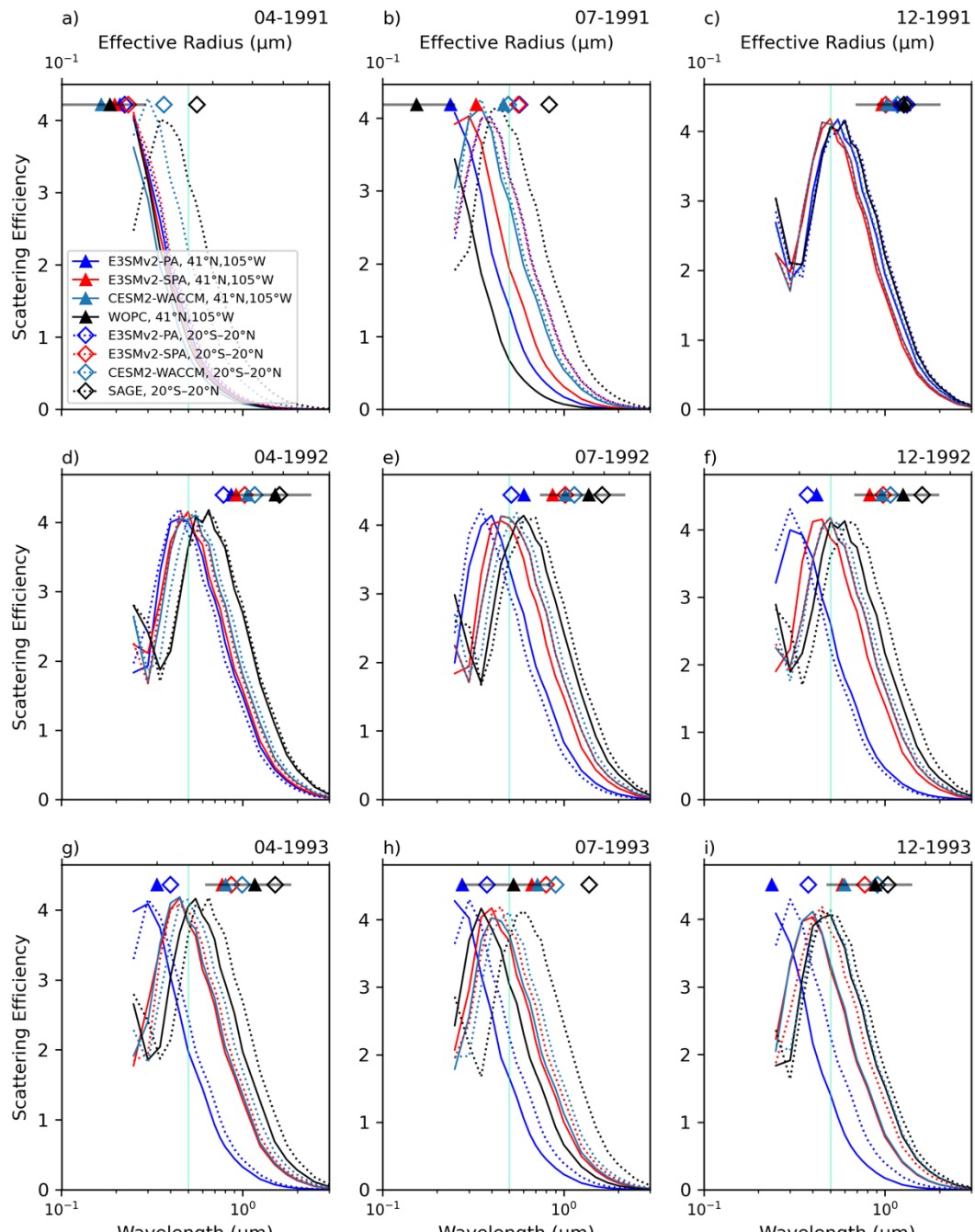

**Figure 10: Monthly stratosphere scattering efficiency (Q$_s$) using effective size parameter (x$_{eff}$ = 2πR$_{eff}$/λ) and sulfate refractive index at 0% relative humidity (Hess et al., 1998). Solid lines are scattering efficiencies over 41˚N 105˚W and dotted lines are scattering efficiencies over 20˚S–20˚N. Monthly R$_{eff}$ are from E3SMv2-PA, E3SMv2-SPA, CESM2-WAACCM, WOPC, and SAGE-II from Fig. 6c-d, and are marked by colored triangles (41˚N 105˚W; WOPC) and hollow diamonds (20˚S–20˚N; SAGE-II) at the top of the**


### 4.7.1 Modeled direct / diffuse radiation

The differences in $R_{eff}$ between the presented model and observational datasets ultimately affect diffuse and direct radiation at the surface by changing AOD and other aerosol optical properties. In scattering incoming shortwave, a small fraction of incident light is scattered back to space by the volcanic aerosol reducing the amount of energy incident to Earth.

More substantially, the forward scattered SW radiation increases the diffusivity of incident radiation. Crop yields are differentially impacted by the increase in diffuse radiation as certain canopy structures benefit from diffuse radiation penetrating to deeper leaves. Thus, while a decrease in direct radiation has an overall negative influence on crop yield, the increase in diffuse radiation allows some crop types to be less influenced (e.g., rice, soybean, and wheat) then others (e.g., maize) (Proctor et al., 2018). The Mauna Loa observatory documented a lessening of the direct beam and increase in the

diffuse radiation after the eruption of Pinatubo (Robock, 2000), and globally there was a 21% reduction in direct and a 20% increase in diffuse radiation at the surface (Proctor et al., 2018).

We diverge from the idealized $Q_s$ calculations in section 4.7 to analyze modeled radiation diagnostics in the atmosphere, calculated via the online model Mie code and a two-stream approximation for calculating multiple scattering in the atmosphere (Iacono et al., 2008; Neale et al., 2012). The latter calculates diffuse and direct radiation at the surface from

incident radiation, AOD (i.e., the column integrated $b_s$ (Eq. B2)), single scattering albedo (SSA; Fig. S15), and aerosol asymmetry parameter (g; Fig. S16). In a general sense, the input aerosol optical properties determine what fraction of direct radiation makes it through the aerosol layer (i.e., AOD), how much of the incident radiation is absorbed in the aerosol layer (i.e., $1 - SSA$), and the degree to which the radiation is scattered by the aerosol layer (i.e., g).

Figure 11 shows the perturbation in diffuse and direct radiation at the surface due to the Pinatubo aerosol layer in the

visible to ultraviolet wavelengths (with similar behavior in the near-infrared (Fig. S17)). The values are averaged over the whole earth (Fig. 11a), the 41˚N latitude band (Fig. 11b), and 20˚S-20˚N tropical region (Fig. 11c). There is an increase in diffuse radiation across all of the models and a mirrored decrease in direct radiation. These changes track the increasing $R_{eff}$ (Fig. 6) and the enhanced $Q_s(\lambda)$ (Fig. 10). The magnitude of the direct radiation loss tends to be larger than the increase in diffuse radiation at the surface because some of the scattered incoming solar radiation is lost to space. In all regions, E3SMv2-

SPA and CESM2-WACCM have more diffuse and less direct radiation than E3SMv2-PA, the latter of which rapidly approaches zero as the aerosol are removed from the atmosphere. The peak direct and diffuse influence occur in the tropics (Fig. 11c) prior to the 41˚N band (Fig. 11b) as aerosol transport to northern latitudes relies on stratospheric circulation. In the 41˚N band (Fig. 11b) CESM2-WACCM has a higher proportion of diffuse radiation at the surface than all of the other models.

Diffuse radiation is not only attributed to aerosol size and material properties – as is $Q_s$ in Fig. 10 – but is also related

to the aerosol number concentration. This is also true of AOD, which is a cloud free, aerosol-specific input into multiple

scattering radiation calculations in the models. AOD, which accounts for both number and $Q_s$, gives a cleaner signal of the aerosol influence. Figure 12 shows AOD over the same spatial regions as Fig. 11. While peaks in AOD and diffuse radiation differ, the intermodal relationships are very similar between the two comparisons. As in Fig. 11b, Fig. 12b shows CESM2-WACCM with higher AOD over the 41°N band. This is attributed to both the larger aerosol $R_{eff}$ (Fig. 6c) and higher

accumulation and coarse mode number concentrations (Fig. S9a) in CESM2-WACCM. The higher number concentrations are correlated with more $SO_2$ transport into the midlatitudes in CESM2-WACCM (Fig. S10a), which is likely due to slower oxidation of $SO_2$ in CESM2-WACCM due to the aforementioned OH depletion in this model.

The impact of clouds on diffuse/direct radiation in the midlatitudes shows up in Fig. 11b, with peaks in normalized total cloud cover (right axis) corresponding to diffuse (direct) radiation peaks (troughs), especially in the late spring/early

summer of 1992 and 1993. The approximate co-location of peaks in cloud cover (Fig. 11b) and AOD (Fig. 12b) in the midlatitudes in June of 1992 and 1993 may indicate the presence of aerosol indirect effects. This is consistent with findings by Liu and Penner (2002) which show high rates of homogeneous ice nucleation persisting in the northern midlatitudes through July 1992.

Lastly, E3SMv2-presc has good agreement with E3SMv2-SPA and CESM2-WACCM when compared globally or in

the tropics, but reports ~60% the modeled magnitudes for diffuse/direct radiation and ~40-60% modeled AOD in the midlatitudes (Fig. 11 and 12, respectively). This may be due to a low bias in AOD in GLoSSAC V1 (Thomason et al., 2018) used here where higher latitudes have a low bias in AOD attributed to linear interpolation of the SAGE-II data (Kovilakam et al., 2020).

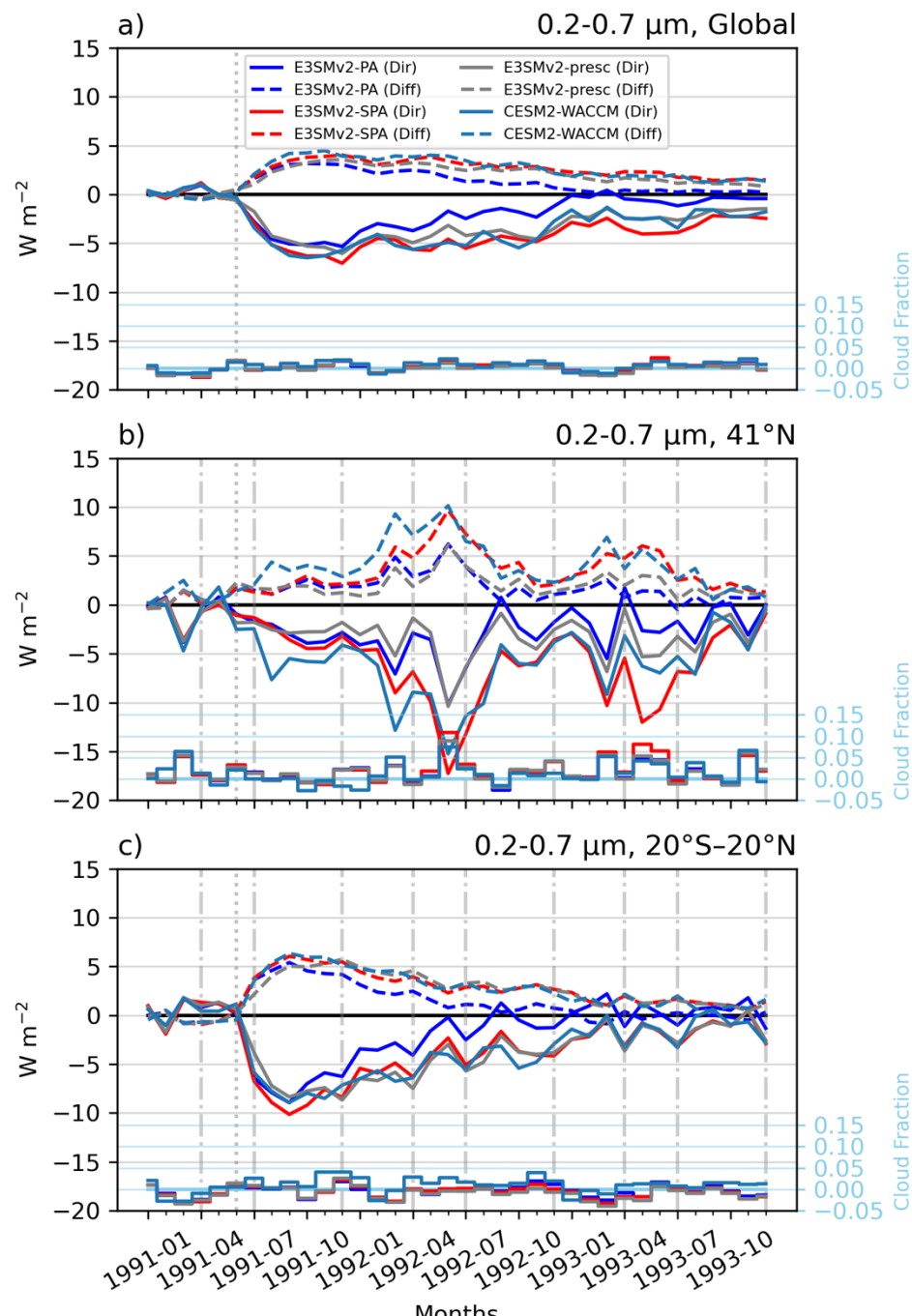

Figure 11: Pinatubo induced changes in diffuse and direct solar radiation at the surface in the 0.2-0.7 μm band (left axis), and total cloud cover fraction (right axis; blue). Regions include global (a), midlatitude band (b; 41˚N), and tropical band (c; 20˚S-20˚N) averages. The Pinatubo eruption is marked with the gray dotted line at 1991-06, and the sample months for Fig. 10 are marked with the gray dash-dot lines. The data is normalized to 1990 monthly means.

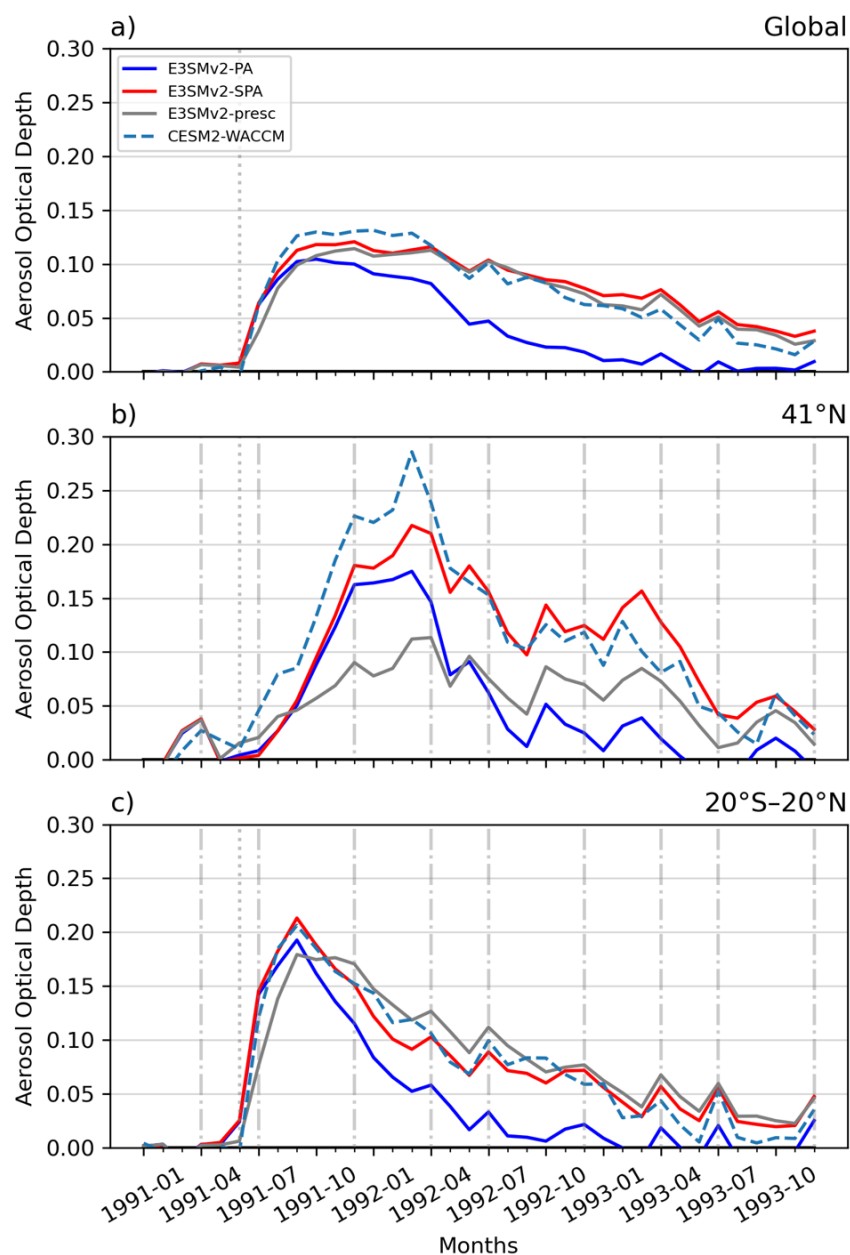

Figure 12: Pinatubo induced changes in aerosol optical depth (AOD) at 0.55 µm. Regions include global (a), midlatitude band (b; 41°N), and tropical band (c; 20°S-20°N) averages. The Pinatubo eruption is marked with the gray dotted line at 1991-06, and the sample months for Fig. 10 are marked with the gray dash-dot lines. The data is normalized to 1990 monthly means.

## 4.8 Single particle absorption efficiency

While the Pinatubo eruption resulted in a net climate cooling effect due to increased scattering, it also contributed to stratospheric warming through the absorption of outgoing longwave radiation within the aerosol layer (Kinne et al., 1992). Fig. 13 shows the $Q_a$ calculated offline from model and in-situ $R_{eff}$, with a dark red line indicating the wavelength of peak terrestrial black-body irradiance (10 μm; $\lambda_{earth}$). While sulfate scatters strongly at visible wavelengths ($\lambda_{solar}$ in Fig. 10 and 9), it is not an effective absorber at visible wavelengths ($Q_a = 0$ at $\lambda_{solar}$) and the determining factor in stratospheric heating is the

magnitude of $Q_a$ at $\lambda_{earth}$. The wavelength dependence of $Q_a$ is weaker than $Q_s$ (proportional to $x_{eff}$ (i.e, $\lambda^{-1}$)) due to a weaker longwave absorption sensitivity to $R_{eff}$ (Lacis, 2015), and $Q_a$ is strongly tied to the imaginary part of sulfate refractive index at 0% relative humidity ($n_{Hess}$; Appendix B) (Hess et al., 1998). The proportionality to $x_{eff}$ is denoted in Fig. 13 by the linear increase in $Q_a(\lambda)$ with increasing $R_{eff}$; the dependence on the imaginary part of $n_{Hess}$ is reflected in the unchanging pattern in absorption magnitudes for all datasets and times. Generally speaking, $Q_s$ is characterized by changes in the wavelength and is

more sensitive to aerosol size fluctuation (Fig. 10 and 9), while $Q_a$ shows changes in the absorption efficiency and is less sensitive to aerosol size fluctuation.

    In Fig. 13, the linear dependence of $Q_a$ on $R_{eff}$ means that an individual particle in the tropics has more absorption than the midlatitudes across the whole period, and the observations will have more absorption than the models. This is explained by the $R_{eff}$ in these two regions at the top of each panel. For example, the smaller modeled $R_{eff}$ at 41˚N leads to a

modeled $Q_a$ that is 15-30% lower than WOPC through 1992 (Fig. 13c-f).

    Because the same refractive index is used across all models and observational datasets, the differences in $Q_a$ are due only to differences in aerosol size. However, this is a simplifying assumption for the modeled $Q_a$ due to their explicit calculation of the soluble aerosol refractive index ($n_s$) in the stratosphere based on aerosol water and sulfate content (Eq. B4). E3SM simulates a stratosphere with unrealistically low stratospheric water vapor (Keeble et al., 2021; Christiane Jablonowski,

personal communication, 2024) which results in aerosol that are mostly sulfate (Fig. S18) with a refractive index similar to $n_{Hess}$ (Appendix B). Higher water vapor in CESM2-WACCM means that the assumption of $n_{Hess}$ in calculation of $Q_a$ doesn't account for the volume-weighting of the hydrated aerosol refractive index (Appendix B). When interstitial stratospheric water and sulfate are used to calculate a volume-weighted $n_s$ using $n_{Hess}$ and $n_{wat}$, $Q_{abs}$ at $\lambda_{earth}$ is smaller than that reported with only $n_{Hess}$ by ~14-20% over the 41˚N band and ~13-18% over 20˚S–20˚N (Table S3). This is due to a smaller magnitude imaginary

refractive index for water at this wavelength.

    It is unclear whether injecting volcanic $H_2O$ would help offset the stratospheric dry bias in E3SM. Abdelkader et al. (2023) showed that the injection of $H_2O$ for a 20 km Pinatubo-like injection increases sulfate mass and stratospheric AOD by ~5%, but at the colder temperatures in the lower stratosphere almost all of this water vapor freezes and sediments out. Retention of water vapor in the stratosphere depends on injections at higher altitudes where temperatures are warmer. Given that

E3SMv2-SPA already tends to produce higher sulfate burdens compared to other simulations (Fig. 1), it is possible that the enhanced sulfate aerosol mass attributed to the addition of water vapor in the plume would bias the aerosol sulfate content high compared to observations, even as it would improve upon the aerosol size. Furthermore, the lower injection height of our simulations (20 km) and a cold-point tropopause that is too cold in E3SMv2 (Christiane Jablonowski, personal communication, 2024) would aid in the rapid removal of most of the injected water, reducing the effectiveness of the injection on aerosol properties.

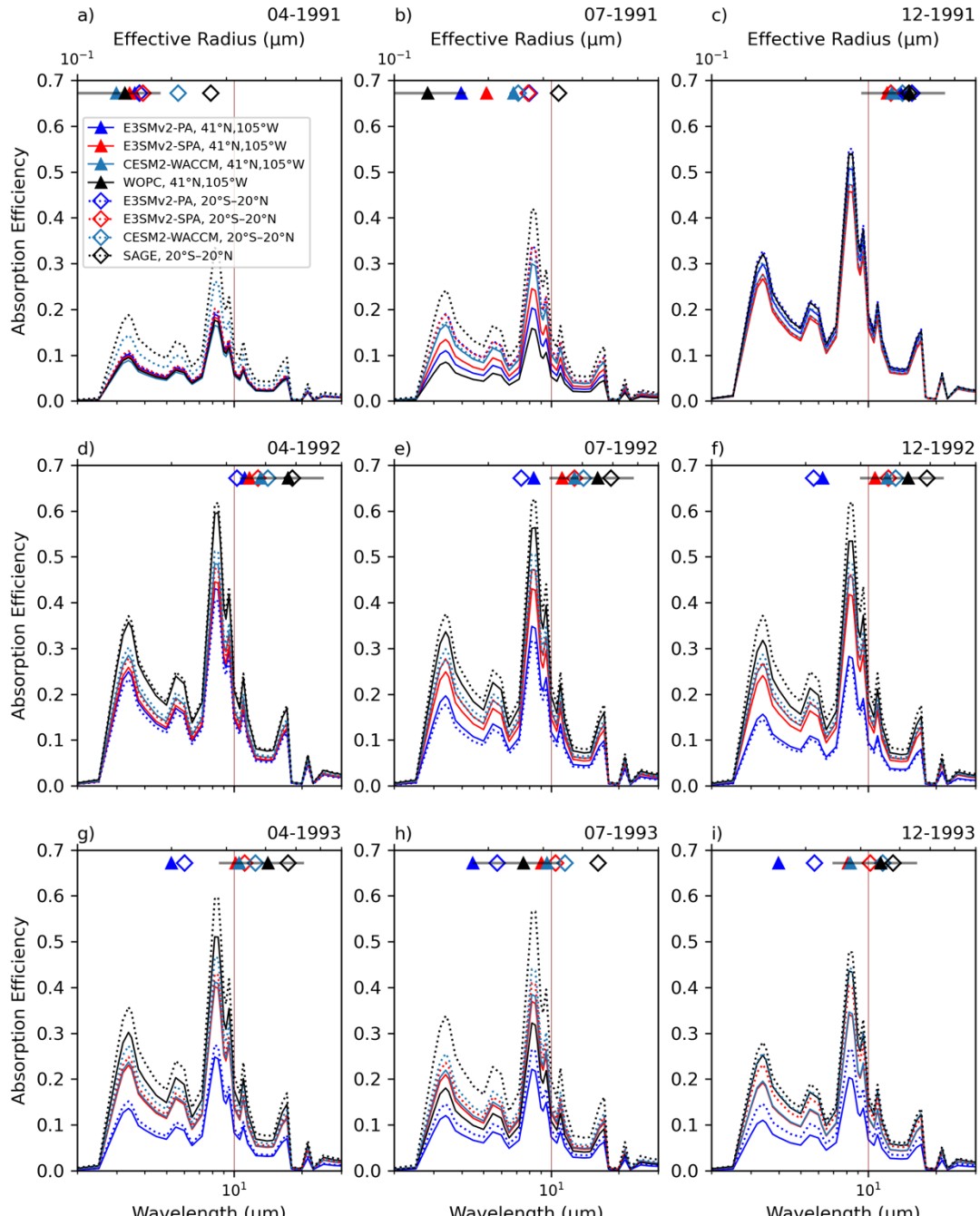

**Figure 13: Monthly stratosphere absorption efficiency ($Q_a$) using effective size parameter ($x_{eff} = 2pR_{eff}/\lambda$) and sulfate refractive index at 0% relative humidity (Hess et al., 1998). Solid lines are scattering efficiencies over 41°N 105°W and dotted lines are scattering efficiencies over 20°S–20°N. Monthly $R_{eff}$ are from E3SMv2-PA, E3SMv2-SPA, CESM2-WAACCM, WOPC, and SAGE-II from Fig. 6c-d, and are marked by colored triangles (41°N 105°W; WOPC) and hollow diamonds (20°S–20°N; SAGE-II) at the top of the**

plot. The bottom x-axis is wavelength ($\lambda$; 3–40 $\mu$m). The top axis is $R_{eff}$ (0.1-0.7 $\mu$m) used in the calculation of $Q_a$ via the $x_{eff}$. The dark red line marks the terrestrial black body wavelength of maximum irradiance (10 $\mu$m).

### 4.8.1 Modeled longwave heating rates

The absorption of outgoing longwave radiation in the stratosphere results in local heating of the aerosol layer. Figure 14 shows the longwave heating rate (LWH; K day$^{-1}$) due only to Pinatubo aerosols and normalized to 1990 monthly means over the same regions as Fig. 11 and 12. As for Fig. 11 and 12, Fig. 14 is calculated from the online model radiation code. Higher initial global (Fig. 14a) and 41°N (Fig. 14b) LWH in CESM2-WACCM up to March 1992 mirrors model behavior for AOD (Fig. 12a-b). The similarity to AOD indicates a LWH dependence on aerosol absorption optical depth (AAOD; level integrated $b_a$ (Eq. B3)) – and therefore aerosol number concentration and $Q_a$ – in these two regions. As with diffuse/direct radiation (Section 4.7.1), characterizing the stratospheric longwave heating rate based solely on $Q_a$ (i.e., aerosol size) is an oversimplification of the actual stratospheric absorption because number concentration is also acting to scale the $b_a$ (Note: the E3SMv2-presc has no Pinatubo signal in LWH which is related to how the prescribed volcanic forcing is treated in the model and is not indicative of a missing physical mechanism. As a result, E3SMv2-presc is not included in this comparison).

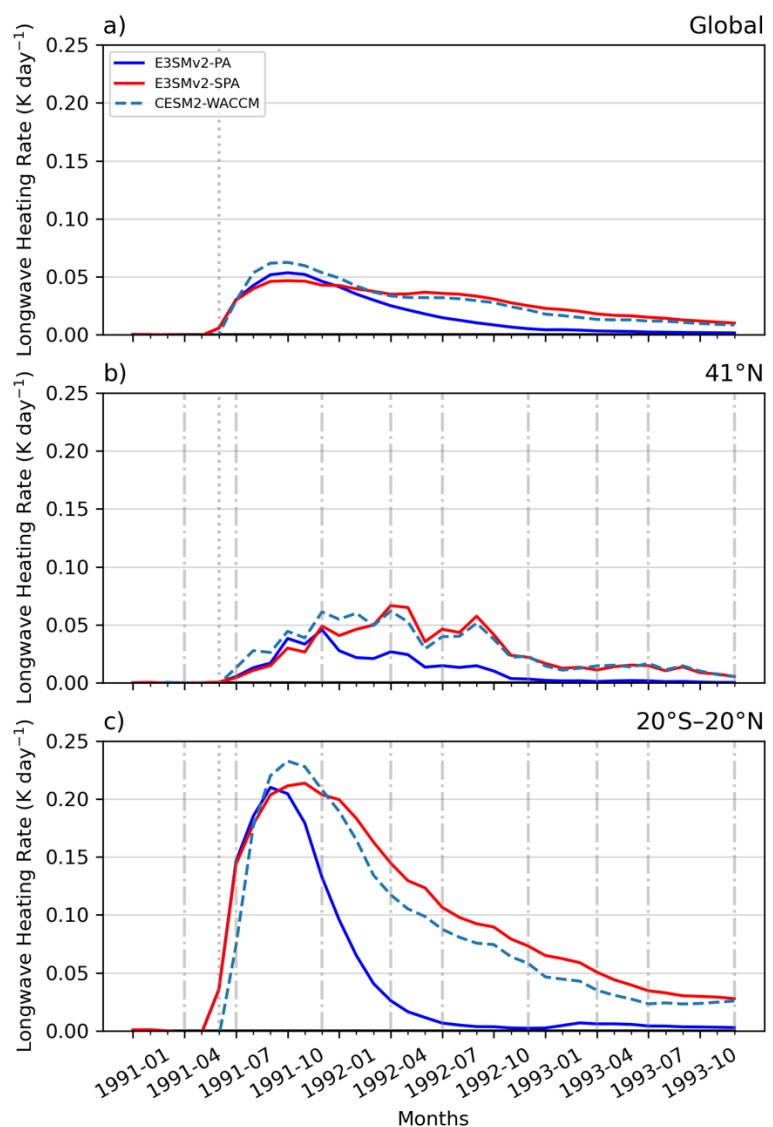

**Figure 14: Pinatubo induced changes in longwave heating rate (K day$^{-1}$) over the modeled longwave wavebands (waveband midpoints of ~3.5-514 μm) . Regions include global stratospheric mean (a), a midlatitude band stratospheric mean at WOPC pressure levels (b; 41°N, 130-10 hPa), and a tropical band stratospheric mean at SAGE-II pressure levels (c; 20°S-20°N, 50-10 hPa). The Pinatubo eruption is marked with the gray dotted line at 1991-06, and the sample months for Fig. 10 are marked with the gray dash-dot lines. The data is normalized to 1990 monthly means and contributions from non-aerosol effects (i.e., clouds and gases) have been removed.**


## 5. Discussion

The preceding sections present encouraging results for E3SMv2-SPA and its ability to model explosive volcanic eruptions. These results begin at the global scale and move into the micro-scale, tying the latter to the former to understand the strengths and weaknesses of the modal aerosol modeling approach. While this stratospheric parameterization is new to E3SM, it is based on a modeling framework developed for CESM2-WACCM which has an additional interactive chemistry element not present in E3SM (Mills et al., 2016). Along with the variety of observational datasets, E3SMv2-SPA is validated against

CESM2-WACCM which elucidates the effects of interactive chemistry on aerosol formation. Ongoing work on a 5-mode version of MAM in E3SM (MAM5) will include interactive stratospheric sulfate chemistry and will have a separate stratospheric mode for volcanic sulfate to avoid impacts on tropospheric coarse mode aerosol (Ke et al., In Prep.; Hu et al., In Prep.).

On the global scale, E3SMv2-SPA performs well compared to observational datasets and has similar behavior to

CESM2-WACCM. The near-globally averaged stratospheric sulfate burden in E3SMv2-SPA agrees well with SAGE-$3\lambda$ and HIRS remote sensing datasets (Fig. 1), indicating that the model does a reasonable job of simulating aerosol lifetime and loading in the stratosphere. The higher mass burdens in E3SMv2-SPA than in CESM2-WACCM point to the OH consumption in CESM2-WACCM's interactive chemistry limiting aerosol formation from $SO_2$. For near-global AOD, the volcanically parameterized E3SMv2-SPA, CESM2-WACCM, and E3SMv2-presc all have similar performance and all exhibit a similar

shape to the retrieved AVHRR AOD (Fig. 2). The models underestimate AOD compared to AVHRR in the 15 months following the eruption, but then agree well with AVHRR to the end of 1993. An aspect of the initial eruption that is left out of the models is the injection of large volcanic ash particles, and this may partly explain the low AOD bias shortly after the eruption. While these large particles are short lived (~3 days (Guo et al., 2004a)), their presence heats the volcanic cloud and can loft the volcanic plume to higher altitudes at rates of 1 km/day in the first week, increasing the lifetime of the stratospheric

sulfate and AOD (Stenchikov et al., 2021). The choice of injection may also be contributing to AOD underestimation, with E3SMv2-SPA and CESM2-WACCM both exhibiting lower tropical confinement than AVHRR and E3SMv2-presc (Fig. 3), leading to $R_{eff}$ and lower AOD (Clyne et al., 2021). This emulates the behavior of other models with similar injection parameters (Quaglia et al., 2023). Slightly better tropical entrainment in E3SMv2-SPA is attributed to a higher plume compared to CESM2-WACCM (Fig. S6), so increasing the injection height or adding a lofting mechanism associated with volcanic ash

may improve this feature in both E3SMv2-SPA and CESM2-WACCM.

The climate impacts of volcanic aerosol are explored through TOA flux (Fig. 4) and atmospheric temperature (Fig. 5) perturbations. In E3SMv2-SPA, the maximum reductions in all-sky ASR (-5.5 W m$^{-2}$), OLR (-3.79 W m$^{-2}$), and NET (-3.0 W m$^{-2}$) flux compare reasonably well with those of ERBS (-5.3 W m$^{-2}$, -3.05 W m$^{-2}$, and -3.8 W m$^{-2}$, respectively). The clearest improvement of E3SMv2-SPA compared to ERBS is in ASR which also has the most distinct Pinatubo signal in the data

especially during 1992 when aerosol are at their largest. There is also a stark difference between the E3SMv2-PA and E3SMv2-SPA, CESM2-WACCM, and E3SMv2-prec clear-sky and aerosol-only diagnostics, with additional reduction of 2.5–3 W m$^{-2}$

in the three volcanic parameterized models. OLR is noisier but does show increased LW absorption during the year 1992. Seasonal cloud albedo and surface cooling effects are introduced in models and observations through our choice of normalization year, and their effect can be removed in the models through comparison of clear-sky and aerosol-only TOA flux diagnostics. Temperature anomalies are calculated for the year 1992, corresponding to the greatest model spread in the TOA flux. Compared to MERRA-2 and RICH-obs, E3SMv2-SPA has a more comparable distribution of temperature anomalies than E3SMv2-PA with notable improvements in the stratosphere over the midlatitudes. CESM2-WACCM is nearly identical to MERRA-2 temperature due to it being nudged to that particular dataset. This nudging puts CESM2-WACCM in very good agreement with MERRA-2, while E3SMv2-SPA overestimates 50 hPa temperatures and underestimates 850 hPa temperatures compared to MERRA-2. Aside from noting the improvement from E3SMv2-PA to E3SMv2-SPA, we caution against directly relating these temperature changes to aerosol size without a more rigorous removal of internal variability in the model (e.g., Santer et al., 2014).

The modelled aerosol size – represented by $R_{eff}$ – is consistently underestimated compared to both in-situ and remote sensing datasets (Fig. 6). E3SMv2-SPA also has slightly smaller $R_{eff}$ than CESM2-WACCM, where the slightly larger aerosols are attributed to the preferential growth of fewer initial Aitken mode aerosol associated with the limited OH oxidation in CESM2-WACCM. Unpacking $R_{eff}$ into its component modal distributions on a global scale illustrates the larger modal $D_g$ and also smaller modal number concentration in CESM2-WACCM (Fig. 7). While useful for visualizing the evolution of a variety of volcanic eruptions over this period, validation with observations requires the finer-scale comparison of model and in-situ data. The comparisons with balloon-borne WOPC data (Fig. 8) identify a key reason for the underestimation in $R_{eff}$. Namely, that modelled coarse mode aerosols are not big enough in the models during the first year after Pinatubo. The aerosol modes reach a similar pattern of change after a year and there is very close agreement in the microphysics of E3SMv2-SPA and CESM2-WACCM.

While choice of bulk, sectional, or modal aerosol representation will impact $R_{eff}$ (English et al., 2013; Laakso et al., 2022; Tilmes et al., 2023), the presence of this $R_{eff}$ underestimation is in modal as well as sectional models with similar injection parameters (e.g., ECHAM6-SALSA, SOCOL-AERv, UM-UKCA (Quaglia et al., 2023; Dhomse et al., 2020)) indicates that it isn't attributed solely to the choice of aerosol microphysical approach. We present a variety of potential reasons and solutions for why the model simulates aerosols that are too small, though testing these is beyond the scope of our study. As with AOD, higher injection levels and plume lofting could increase $R_{eff}$ by increasing tropical confinement (Clyne et al., 2021). Improvement of the stratospheric dry bias in E3SMv2 may improve this disagreement by increasing aerosol size due to water uptake. There are also inherent model assumptions that may be contributing to the aerosol small bias: 1) the lack of interparticle van der Waals forces in both E3SMv2 and CESM-WACCM (though recent iterations of WACCM coupled with the CARMA sectional aerosol model do include these forces (Tilmes et al., 2023)), which have been shown to drive aerosol nucleation and lead increased peak $R_{eff}$ following Pinatubo-sized eruptions (English et al., 2013; Sekiya et al., 2016); 2) the nucleation scheme used in these models may be overestimating nucleation rates by 3-4 orders of magnitude (Yu et al., 2023),

leading to more numerous smaller particles. The inability to create large enough aerosol in the models can affect the simulated climate impacts, and the size-based effects on scattering and absorption are explored through the single particle Mie scattering.

The plotted $Q_s$ (Fig. 9,10) and $Q_a$ (Fig. 13) tell an idealized story about how size affects scattering of incoming solar and absorption of outgoing terrestrial radiation. At the wavelength of peak incoming solar irradiance (0.5 μm; $\lambda_{solar}$), the larger observed aerosols manifest as weaker scatterers of incident radiation than E3SMv2-SPA and CESM2-WACCM. This

suggests that the models will overestimate aerosol scattering, though the differences between model and observation are smaller for monthly as opposed to daily averages. For $Q_a$, the relationship is simpler, with a linear increase in aerosol absorption with increasing aerosol size. Our results show that the models may underestimate absorption of light at the wavelength of peak outgoing terrestrial radiation (10 μm; $\lambda_{earth}$) due to their smaller simulated aerosol size. When the effects of modeled interstitial aerosol water are included in the CESM2-WACCM refractive index, modeled $Q_a$ is decreased further when

compared to observations due to a lower absorption of $\lambda_{earth}$ by water than by sulfate.

Care should be taken in interpreting $Q_s$ and $Q_a$ as there are some caveats to the above comparisons. While $R_{eff}$ is a reasonable representation of aerosol size and is an important factor in aerosol optics, it is not the singular factor in aerosol optical properties. Aerosol number is neglected in these comparisons, which is a secondary determiner to aerosol AOD. The importance of aerosol number can be seen in the diffuse radiation (Fig. 11), which has relative model behaviors that mirror

those of AOD (Fig. 12). Number is also important in longwave absorption, where relative modeled longwave heating rates (Fig. 14) similarly mirror those in AOD. Another assumption in the calculation of scattering and absorption efficiencies is the assumption of a stratospheric sulfate refractive index. This is consistently done for remote sensing datasets where no additional information can be provided, but in models that prognostically calculate the volume-weighted soluble refractive index this may be misleading. Here, the same refractive index is assumed across observations and models to make them more comparable

(Hess et al., 1998; Appendix B). It also avoids differences in CESM2-WACCM and E3SM that arise from the extremely low stratospheric water bias in E3SM. The accurate simulation of number and refractive index are two aspects of this work that require further study.

It is unclear the extent to which biases in aerosol size and the resultant impacts on $Q_a$ and $Q_s$ are impacting the downstream climate impacts of stratospheric sulfate. While quantifying this impact and improving this bias are important next

steps, we note that overall E3SMv2-SPA does well compared to observational datasets and produces Pinatubo sulfate burdens and AOD similar to CESM2-WACCM and other models run with comparable volcanic injection parameters (Quaglia et al., 2023). Furthermore, a free-running historical simulation run with E3SMv2-SPA recreates major historical volcanic AOD peaks compared to a five-member E3SMv2 historical simulations with prescribed volcanic forcing (Fig. S1). While some disagreements exist, the general mirroring of the signals gives confidence in simulating variable injection parameters of varying magnitudes with this parameterization. Lastly, comparison of historical surface temperatures (Fig. S2) and net top-of-

model radiative flux (Fig. S3) from two ensembles of a fully-coupled E3SMv2-SPA historical simulation indicate similar performance to E3SMv2 historical simulations with prescribed volcanic forcing. This indicates a reasonable climate representation in a historical setting.

This study has served to validate the stratospheric prognostic aerosol treatment in E3SMv2-SPA against observations,
while also showing it can have reasonable performance compared to the more comprehensive chemical treatment of CESM2-WACCM. While an interactive chemical treatment does have better overall agreement with observations of mass burden, AOD, and R$_{eff}$, E3SMv2-SPA can still serve as a viable alternative. We have connected the microphysical evolution of aerosols to radiative impacts, which then cascade into further downstream impacts (i.e., changes in temperature, precipitation, water vapor, etc.) that will be explored in follow-on papers. The use of a prescribed forcing dataset (E3SMv2-presc) does not allow the same connections to be drawn as there is no aerosol mass and size distribution that can evolve dynamically and feedback on the radiation balance in the earth system, both directly and indirectly (i.e., aerosol-cloud interactions). Issues in representation and the lack of a dynamically evolving aerosol plume in prescribed forcing datasets could have a significant impact on detection and attribution studies looking at mid-latitiude agricultural productivity impacts, for instance, and illustrates the necessity of prognostically evolved aerosol for forthcoming attribution studies sensitive to spatio-temporal evolution. E3SMv2-SPA will facilitate prognostic aerosol simulations that include naturally produced stratospheric sulfate (i.e., volcanoes) as well as anthropogenic climate interventions to modify incoming solar radiation with stratospheric aerosol injection (SAI). Detection and attribution studies of societally-relevant climate impacts in free-running fully-coupled climate simulations with varied stratospheric sulfate aerosols are essential for casual identification of the source of impact and risk assessments (OSTP, 2023).

## Appendix A: Effective radius and aerosol size distribution calculations

Aerosol effective radius (R$_{eff}$) – or area-weighted aerosol radius – is a good representation of a size distribution's optically relevant size (Hansen and Travis, 1974; Russell et al., 1996). This relies on the fact that intercepted light is proportional to the particle radius in Mie and geometric scattering regimes. These regimes can be designated by an aerosol size parameter, $x = \frac{2\pi r}{wavlength\ (\lambda)}$, with Mie scattering corresponding to $x \approx 0.2 - 2000$ and geometric scattering corresponding to $x > 2000$ (Petty, 2006). R$_{eff}$ is calculated by taking the ratio of the third aerosol moment (volume) and second aerosol moment (cross-sectional area) (M$_3$ and M$_2$, respectively) of the aerosol size distribution (Hansen and Travis, 1974):

$$R_{eff} = \frac{M_3}{M_2}, \qquad\qquad (1)$$

A lognormal size distribution is defined as

$$n_N(\ln r) = \frac{dN(r)}{d\ln r} = \frac{N}{\ln \sigma_g \sqrt{2\pi}} e^{\left[-\frac{1}{2}\left(\frac{\ln r - \ln r_g}{\ln \sigma_g}\right)^2\right]}, \qquad\qquad (2)$$

where N is total aerosol number concentration, $r_g$ is the wet geometric mean radius, r is the radius, and $s_g$ is the geometric standard deviation. The analytic expression for the aerosol moments, $M_n$, of a lognormal distribution are,

$$M_n = N r_g^n e^{\left[\frac{n^2}{2}(\ln \sigma_g)^2\right]},$$

(3)

For a multimodal system, the modal moments ($M_{n,i}$) can be summed, giving the final equation for $R_{eff}$:

$$R_{eff} = \frac{\sum_{i=1}^m M_{3,i}}{\sum_{i=1}^m M_{2,i}} = \frac{\sum_{i=1}^m N_i r_{g,i}^3 e^{\left[\frac{9}{2}\ln \sigma_{g,i}^2\right]}}{\sum_{i=1}^m N_i r_{g,i}^2 e^{\left[2\ln \sigma_{g,i}^2\right]}},$$

(4)

When calculating $R_{eff}$ over a range, z, of model hybrid or pressure levels (e.g., in the stratosphere, $R_{eff,strat}$), we integrate over the vertical levels weighted by the vertical layer thickness (*h*) and the aerosol surface area density (SAD).

$$\boldsymbol{R_{eff,strat}} = \frac{\sum_z (R_{eff} * h * SAD)}{\sum_z (h * SAD)},$$

(5)

The thickness, h, is calculated using the hypsometric equation:

$$\boldsymbol{h = \frac{R_d * T}{g} ln \frac{P_{z+1}}{P_z}},$$

(6)

and SAD ($cm^2$ $cm^{-3}$) is represented by the second moment as

$$SAD = 4\pi M_2,$$

(7)

For stratospheric averages of lognormal aerosol distributions, we calculate the individual modal aerosol distributions (Eq. A2) and sum across size bins (*r*) to create a multi-modal distribution at each grid cell ($n_N(\ln r)_{3mode}$). We then weight this distribution by the grid cell thicknesses (*h*) and integrate above the tropopause

$$n_N(\ln r)_{strat} = \frac{\sum_z (n_N(\ln r)_{3mode} * h)}{\sum_z (h)},$$

(8)

### Appendix B: Single-particle Mie scattering

In order to understand how improvements to stratospheric aerosol size distributions affect simulated climate impact, aerosol scattering and absorption efficiencies ($Q_s$, $Q_a$) are calculated based on Mie theory, the solution of Maxwell's equations

for the interaction of radiation with a sphere (Hansen and Travis, 1974). The $Q_s$ ($Q_a$) is the surface area of the shadow cast by the particle due to scattering (absorption) of intercepted light divided by the geometric cross section ($\pi r^2$). $Q_s$ and $Q_a$ are calculated using the python package, miepython (https://miepython.readthedocs.io/en/latest/index.html#). This package simulates absorption and scattering of incident light by non-absorbing or partially absorbing spheres and requires a complex aerosol refractive index ($n = a + bi$) and aerosol size parameter (aerosol circumference over the incident wavelength of light) as inputs. In the complex refractive index, the real and imaginary parts define scattering and absorption, respectively. Here, an effective size parameter is used as a proxy for the particle size distributions in our calculations:

$$x_{eff} = \frac{2\pi R_{eff}}{\lambda} , \tag{1}$$

We note that because Mie theory calculates these values based on interactions with a single sphere and we are using the number-independent $R_{eff}$ as an input, $Q_s$ and $Q_a$ do not account for effects of different number concentrations. The volume scattering and absorption coefficients ($\beta_s, \beta_a$; units of m$^{-1}$) do include the effect of aerosol number, but are not generated based on our comparisons due to the ambiguity of assuming an effective aerosol number. Their equations are included below for completeness.

$$\beta_s = NQ_s\pi r^2 , \tag{2}$$
$$\beta_a = NQ_a\pi r^2 , \tag{3}$$

The aerosol refractive index in the models is derived from a combination of sulfate aerosol and water due to the prevalence of sulfate in the stratosphere following Pinatubo. Because sulfate is a soluble species, a soluble refractive index ($n_s$) can be calculated with the volume mixing rule (Ghan and Zaveri, 2007):

$$n_s = \frac{1}{V_s}\left(\frac{n_{wat}m_{wat}}{\rho_{wat}} + \frac{n_{sulf}m_{sulf}}{\rho_{sulf}}\right) , \tag{4}$$

where $n_{wat}$, $m_{wat}$, and $r_{wat}$ are the water complex refractive index, aerosol water mass (kg), and water density (kg m$^{-3}$); and $n_{sulf}$, $m_{sulf}$, and $r_{sulf}$ are the sulfate complex refractive index, sulfate mass (kg), and sulfate density (kg m$^{-3}$). The $m_{sulf}$ and $m_{wat}$ are summed across the three aerosol modes. The soluble aerosol volume from sulfate and water across modes ($V_s$) is calculated as:

$$V_s = \frac{m_{sulf}}{\rho_{sulf}} + \frac{m_{wat}}{\rho_{wat}} \tag{5}$$

Sulfate and water refractive indices are taken from the literature (Hess et al., 1998; Hale and Querry, 1973). The sulfate refractive index reported in Hess et al. (1998) ($n_{Hess}$) is at 0% relative humidity and corresponds to aerosol that is approximately 25% water / 75% $H_2SO_4$ by mass (Seinfeld and Pandis, 2006). A similar solution ratio has been used in other remote sensing applications (see Sections 3.1, 3.5), and for our in-situ and remote sensing datasets we assume stratospheric aerosol optical properties are reasonably represented by $n_{Hess}$. The pure sulfate refractive index used in E3SM/CESM ($n_{sulf}$) is calculated from water and $n_{Hess}$ following methods in Ghan and Zaveri (2007) and is nearly identical to $n_{Hess}$, albeit reported at a lower wavelength resolution. Given their similarity and the higher wavelength resolution in $n_{Hess}$, we use $n_{Hess}$ instead of E3SM's pure sulfate refractive index for model single particle scattering calculations.

## 6. Code Availability

The model code base used to generate E3SMv2-SPA and E3SMv2-PA – along with information for how to access the publicly available CESM2-WACCM code base – can be found on Zenodo at https://doi.org/10.5281/zenodo.10602682. Plotting and processing scripts used in the analyses of this paper can be found on Figshare at https://doi.org/10.6084/m9.figshare.24844815.v2.

## 7. Data Availability

The WOPC balloon data are available in the data repository at the University of Wyoming Libraries (Deshler and Kalnajs, 2022). AVHRR data can be found at doi:10.25921/w3zj-4y48 (Xuepeng and NOAA CDR Program). The RICH-obs 32 ensemble dataset can be accessed via https://imgw.univie.ac.at/forschung/klimadiagnose/raobcore/. The ERBS dataset can be found at http://www.met.reading.ac.uk/~sgs02rpa/research/DEEP-C/GRL/. For SAGE, AVHRR, and some of the postprocessed WOPC data, see https://doi.org/10.7298/mm1s-ae98 (Quaglia et al., 2022) .The original 2-degree volcanic emissions file (VolcanEESMv3.11_SO2_850-2016_Mscale_Zreduc_2deg_c180812.nc) can be retrieved from https://svn-ccsm-inputdata.cgd.ucar.edu/trunk/inputdata/atm/cam/chem/stratvolc/. The raw model output data used in this paper has been archived on the DOE NERSC resources and is available upon request.

## 8. Author Contribution

**Conceptualization**: Hunter Brown, Benjamin Wagman, Diana Bull, Kara Peterson, Xiaohong Liu, Ziming Ke; **Data curation:** Hunter Brown, Benjamin Wagman, Benjamin Hillman, Ziming Ke, Lin Lin; **Formal analysis**: Hunter Brown; **Funding acquisition**: Diana Bull, Kara Peterson; **Investigation:** Hunter Brown; **Methodology**: Hunter Brown, Benjamin Wagman, Diana Bull, Xiaohong Liu, Ziming Ke, Lin Lin; **Project administration:** Diana Bull, Kara Peterson; **Resources**: Diana Bull, Kara Peterson; **Software**: Hunter Brown, Ben Hillman, Ziming Ke, Lin Lin; **Supervision**: Diana Bull, Kara Peterson, Benjamin Wagman; **Validation**: Hunter Brown; **Visualization**: Hunter Brown; **Writing – original draft preparation**: Hunter Brown, Diana Bull, Kara Peterson, Benjamin Wagman; **Writing – review & editing**: All authors.

## 9. Competing interests

The authors declare that they have no conflict of interest.

## 10. Acknowledgements

We would like to acknowledge Terry Deshler and Lars Kalnajs for providing access to the WOPC balloon data and helping with questions that arose. We would also like to acknowledge Mike Mills for his correspondence and clarifications on MAM4 stratospheric aerosol treatments in CESM2-WACCM. Po-Lun Ma provided 6-hourly MERRA2 nudging data. Processing the
1195 MERRA-2 reanalysis for nudging E3SM was supported by the Enabling Aerosol-cloud interactions at Global convection-permitting scalES (EAGLES) project (project no 74358), funded by the U.S. Department of Energy, Office of Science, Office of Biological and Environmental Research, Earth System Model Development program area. Thanks to Ilaria Quaglia for access to processed AVHRR, SAGE, WOPC data. Work at LLNL was performed under the auspices of the U.S. DOE by Lawrence Livermore National Laboratory under contract DE-AC52-07NA27344 UCRL: LLNL-JRNL-858807. This research
used resources of the National Energy Research Scientific Computing Center (NERSC), a U.S. Department of Energy Office of Science User Facility located at Lawrence Berkeley National Laboratory, operated under Contract No. DE-AC02-05CH11231 using NERSC award BER-ERCAP0022865. This work was supported by the Laboratory Directed Research and Development program at Sandia National Laboratories, a multimission laboratory managed and operated by National Technology and Engineering Solutions of Sandia LLC, a wholly owned subsidiary of Honeywell International Inc. for the U.S.
Department of Energy's National Nuclear Security Administration under contract DE-NA0003525. SAND2023-XXXX.

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
