# Peer review of "Validating a microphysical prognostic stratospheric aerosol implementation in E3SMv2 using observations after the Mount Pinatubo eruption"

_EGUsphere, 2023_

## Author Comment (AC3)

We thank the reviewer for their thorough and thoughtful review. We have addressed your comments below and have added our own comments in RED:

The authors present the development of a stratospheric prognostic aerosol (SPA) capability for the Energy Exascale Model, version 2 (E3SMv2) to simulate the stratospheric aerosol formation in the aftermath of large explosive volcanic eruptions. Their implementation includes changes to the 4-mode Modal Aerosol Module microphysics to allow for larger particle growth and more accurate stratospheric aerosol lifetime following the Mt. Pinatubo eruption. Hunter et al. tested their model for the Post Pinatubo period with remote sensing and in situ observations and the interactive chemistry-climate model, CESM2-WACCM. On the global scale, E3SMv2-SPA performs well compared to observational datasets and has similar behavior to CESM2-WACCM. They found that the modeled aerosol effective radius for both versions is consistently lower than satellite and in-situ measurements (max differences of ~30%). Compared to observations, the models also show a higher diffuse radiation at the surface and a larger cooling and an underestimation in stratospheric heating in the models.

Although the manuscript type is declared as a development and technical paper, the content should be placed in the general context of global stratospheric aerosol modelling, ***otherwise it should be published as a specific technical institute report. The introduction and discussion sections therefore need some substantial improvements. The motivation of the paper could be more clearly stated, and some of the results could be discussed in a broader context. I therefore recommend publication after major revisions, see below.

General comments

In the introduction important literature is missing. Several global stratospheric aerosol modelling studies have been published in the last year. An overview of the development and current state of stratospheric aerosol modelling can be found for example in Kremser et al. (2016) and in Timmreck et al. (2018). In addition, a number of global comparative aerosol modelling studies have been published in recent years, e.g. for background aerosol (Brodowsky et al., 2024), volcanic events (Marshall et al., 2018; Clyne et al., 2021; Quaglia et al., 2023) and artificial sulphur injections (Franke et al., 2021; Weisenstein et al., 2022). I was more than surprised that these studies were completely ignored by the authors. In particular, the results of the Pinatubo study by Quaglia et al. (2023) should be mentioned and discussed in the paper and not just used as a reference to observational data.

Thank you for pointing this out. We have addressed his comment by adding a paragraph to the introduction specific to stratospheric aerosol, touching on background aerosol, the role of volcanic injections, and stratospheric aerosol injection. We also added three paragraphs that address the significance of microphysical representation, injection height, and sulfate chemistry for modeling volcanic eruptions from the context of the above and related references.

I wonder how model specific your results are?  How valuable are they to other stratospheric aerosol modellers? I am missing in the discussion section a dedicated paragraph on the strengths

and weaknesses of the applied global aerosol models with respect to other global stratospheric aerosol models. Recent intercomparison studies of global aerosol models reveal several difficulties that the current generation of global aerosol models has to deal with. For example, the study by Qualia et al. (2023) comparing the different model results with satellite observations after the eruption of Mt. Pinatubo shows a stronger transport towards the NH extra-tropics, suggesting a much weaker subtropical barrier in all models. How does the spatial aerosol distribution in your model look like? It should be much better as you nudge the winds, so discrepancies can be traced back to other sources. This could be more elaborated with respect to free running models. Nevertheless, it would be nice to see a global distribution of your sulfate burden/AOD also in the paper or in the supplements.

We agree that the larger context of these results was lacking. To address this we have added a zonally averaged AOD plot to the paper to help clarify global transport in these models and allow for comparison to models in Quaglia et al. (2023). We have also added additional discussion of where the AOD and $R_{eff}$ compare to other modeling work. We also include discussion on results from fully coupled historical simulations (1850-2014) run with E3SMv2-SPA to support the use of this model for historical eruptions.

The motivation of the paper could be stress out a bit more. It is also not really clear to me how different your SPA version is from the MAM4 version in WACCAM, except the model reversal and simplified precursor chemistry.

We have added more to the introduction and discussion regarding the purpose of this model. The key difference is the interactive chemistry versus simplified precursor chemistry. There are also slightly different assumptions for sulfate in the stratosphere (i.e., inclusion of a reversible coarse to accumulation mode transfer in the stratosphere). We present arguments for these choices in our paper and in some of your comments below.

The applied methodology is not sufficiently explained in the manuscript. I miss for example a detailed description how you calculate a spatially averaged aerosol size distribution or effective radii which is not straightforward. A subsection "Methodology" for section 2 would be helpful with more details in the appendix.

We have added more details regarding calculation of stratospheric Reff and size distributions to the end of Appendix A, now titled, "Effective radius and aerosol size distribution calculations"

Specific comments

- Line 2: "…using *observations* after the MT. Pinatubo eruption"
  - Fixed title to reflect this change
- Line 45: "Mt. Pinatubo" sometimes you use "Mt. Pinatubo" sometimes "Pinatubo" only, please be consistent
  - We have removed occurrences of 'Mt.' as well as 'Cerro'
- Lines 49-51 The fact does a model produce similar results like another model does not make it per se to a viable tool

- o We acknowledge that saying this minimizes some of the uncertainties between the two models. We also recognize that this neglects the most important comparisons, which are to the observational data. Still, we think the comparison with CESM2-WACCM is important given that it is a well-validated and widely used model for these comparisons. We have changed this sentence to read: "The overall agreement of E3SMv2-SPA compared to observations and its similar performance to the well-validated CESM2-WACCM makes E3SMv2-SPA a viable alternative to simulating climate impacts from stratospheric sulfate aerosols."
- Lines 95 ff: Concerning the advantages of sectional aerosol models there is a recent paper by Tilmes et al. (2023) in GMD where they are describing a sectional aerosol microphysical model in CESM2 and compare it with the CESM2 standard version with the Modal Aerosol Model MAM4 for the Pinatubo episode. This paper should be cited and briefly discussed here as well.
  - o We have included a reference to this paper. The sentence now reads: "…Pinatubo and larger magnitude eruptions. **More recently, Tilmes et al. (2023) showed that coupling CARMA to WACCM6 better represents the largest aerosol sizes following Pinatubo than a parallel running modal aerosol model.** The modal aerosol approach – representing aerosol size distributions by multiple, evolving lognormal functions – strikes a balance between bulk simplicity and sectional cost. …"
- Line 148: What about sedimentation?
  - o We included a mention of sedimentation.
- Line s172-174: Any reason why you did not take this process into account in your model.
  - o Including this process would certainly be a more complete representation, but our goal with this work was to have a simpler model that performed well in representing stratospheric sulfate formation. We left this reaction out under the assumption that the effects on coarse mode aerosol in the stratosphere would be minimal given the temperature and relative humidity dependent nature of the reversible process. We have added a statement to this effect: "In CESM2-WACCM, this transfer is reversible in the stratosphere, with an aqueous sulfuric acid ($H_2SO_4$) equilibrium pressure that depends on temperature and relative humidity. **We left this out of our implementation under the assumption that, at the low relative humidities and low temperatures characteristic of the stratosphere, the effect from this process would be minimal**."
- Lines 213 -214: This is not what Kremser et al (2016) wrote *"Recent modeling studies support lower stratospheric sulfur levels than those inferred from the TOMS and TOVS observations [Dhomse et al.,2014; Mills et al., 2016]. The difference between the initial and the persistent sulfur levels is important and generally supports a more complex development process following a major eruption than has been considered in the past. (Kremser et al., 2016, page 12),* Please cite them properly
  - o I am also confused as to why the Kremser reference was used here, and thank you for pointing this out. We have changed the lines to the following: "In the case of Pinatubo, while 18-19 Tg of SO2 erupted in the atmosphere, only ~10 Tg remained in the stratosphere 7-9 days after the eruption (Guo et al., 2004b). This rapid reduction in SO2 corresponds to >99% removal of volcanic ash mass (Guo

et al., 2004a). Therefore, 10 Tg of SO2 is emitted in this dataset for further
chemical and microphysical evolution (Mills et al., 2016)."

- Line 451 ff: Solar flux changes: you can also compare your model results to observations
  by the Earth Radiation Budget Experiment (ERBE) (Barkstrom, 1984; Barkstrom and
  Smith, 1986). This would an additional approach

We have added an additional TOA flux figure and new section (4.3 TOA radiative flux) to
address your concern. We also include clear-sky and aerosol-only model diagnostics to
identify biases arising from our subtraction of year 1990 conditions:

**3.3 Top-of-atmosphere radiative flux**

**3.3.1 ERBS**

The TOA global radiative flux at a 1̊x1̊ resolution is used from version 2 of the Diagnosing
Earth's Energy Pathways in the Climate project (DEEP-C) merged data product drawing from
the Earth Radiation Budget Satellite (ERBS) near-global (60Ŝ-60Ň) non-scanning instrument
and other reanalysis and observational datasets (Allan et al., 2014). The ERBS instrument
measures reflected shortwave radiation and total outgoing radiation, allowing for the
separation of longwave radiative flux by subtraction (Minnis et al., 1993).

**4.3 TOA radiation flux**

Figure 4 compares the TOA radiative flux from model simulations to the all-sky ERBS
observations over the 1991-1993 period, subtracting out corresponding monthly means
from the pre-Pinatubo year 1990. Model TOA flux is shown for all-sky (solid lines), clear-sky
(faint dashed lines), and aerosol impact only (faint dotted line) conditions. The radiative flux
is reported as absorbed shortwave radiation (ASR, positive downward flux; Fig. 4a), outgoing
longwave radiation (OLR, positive upward flux; Fig. 4b), and net radiative flux (NET, positive
downward flux; Fig. 4c). In Fig. 4a, ASR shows the clearest model separation 3-4 months
after Pinatubo corresponding with peak AOD (Fig. 2). There is close agreement between
E3SMv2-SPA, E3SMv2-presc, and CESM2-WACCM during the year 1992 which corresponds
to the largest sulfate particles during the Pinatubo plume evolution (see Sections 4.5 and
4.6). The all-sky signal exhibits noise due to differences in atmospheric conditions (i.e., cloud
cover, tropospheric aerosol) and surface albedo between the period of interest and our
control year (1990). There is a clear seasonal increase in ASR in 1991/1992 and 1992/1993
Northern Hemisphere winters relative to Northern Hemisphere summer. When clear-sky (no
influence from clouds) is compared to all-sky conditions in the models the seasonality
disappears, implying that the seasonality is cloud-related and cloud albedo was greater in
Northern Hemisphere winter 1990 than Northern Hemisphere winter 1991/1992 and
1992/1993. Even with noise introduced by non-Pinatubo factors, there is a distinct all-sky
ASR signal in E3SMv2-SPA, CESM2-WACCM, and E3SMv2-presc that is improved compared
to ERBS.
The all-sky OLR (Fig. 4b), which is affected both by aerosol absorption of infrared emissions
from the earth's surface and the cooling of the troposphere and surface by the scattering of
solar radiation, has a weaker response across these models than ASR. This is due in part to a

less efficient absorption of outgoing longwave radiation than scattering of incoming solar radiation, leading to a lower sensitivity of OLR to aerosol growth and evolution (see Section 4.8). The largest spread in model simulations occurs during 1992 when aerosols are at their largest (i.e., highest absorption efficiency of longwave radiation; Section 4.8) and the highest reduction in surface temperatures were observed (Parker et al., 1996). All-sky E3SMv2-SPA has the greatest reduction in OLR from April 1992 to the end of 1993, and overestimates the longwave flux reduction compared to ERBS. This corresponds with E3SMv2-SPA overestimation of global AOD values compared to AVHRR over this period (Fig. 2). During this same period, CESM2-WACCM has slightly better agreement with ERBS, which may be related to the temperature nudging in this simulation which will modulate CESM2-WACCM surface temperature reduction and stratospheric temperature. When clear-sky OLR fluxes are compared, there is a weaker reduction in OLR for E3SMv2-PA, E3SMv2-SPA, and CESM2-WACCM, and nearly no change in E3SMv2-presc during 1992. Due to the lack of stratospheric aerosol in E3SMv2-presc, this appears to be evidence of volcanic influence on high altitude clouds which act to reduce OLR further supporting conclusions from Liu and Penner (2002) and Wylie et al. (1994). Lastly, the aerosol-only model simulations remove the 1991/1992 and 1992/1993 wintertime peaks in the OLR signal, indicating similar or smaller OLR in 1990 than our period of interest due to cooler surface conditions.

The improvements in all-sky NET (Fig. 4c; solid lines) with volcanic parameterizations are less apparent across the models than in ASR (Fig. 4a), but do show improvement during the first 6 months after the eruption and during 1992. Differences in cloud cover and surface conditions between our period of interest and 1990 introduce substantial noise to this comparison, but the removal of clouds (clear-sky) and the isolation of aerosol TOA forcing (aerosol only) show a clear separation of volcanic parameterizing models and E3SMv2-PA.

[Figure]

**Figure 4: Top-of-atmosphere, radiative flux from model simulations and ERBS observations (Allan et al., 2014; Liu et al., 2015). The panels describe: (a) absorbed solar radiation (ASR; positive downward flux); (b) outgoing longwave radiation (OLR; positive upward flux); and (c) net radiative flux (NET=ASR-OLR; positive downward flux). Monthly mean data is normalized to the pre-Pinatubo conditions by subtracting respective monthly means from the year 1990. ERBS TOA flux is under all-sky conditions, while model TOA flux is shown under all-sky (solid line), clear-sky conditions (faint dashed line), and aerosol only (faint dotted line) conditions.**

- Line 494: Date of the Lascar eruption is not correct
  - Fixed.

- Line 571: I am wondering why you choose the following latitudes band and not (also) the location of Mauna Loa where you have some data for a direct comparison.
    - We didn't include a direct comparison between this dataset and the models because it was not clear how to create an equivalent comparison between the coarser temporally resolved model data and the Mauna Loa dataset, the latter of which was reported for clear-sky morning conditions at a fixed solar zenith angle.
- Lines 639-647: Does an integrated longwave heating rate really make sense here. Would not it be more useful to compare stratospheric temperature profiles here where are at least some observations are available, e.g. Free and Angell (2002) and Free and Lazante (2009).
    - We chose longwave heating rate due to its clearer signal (at least given our aerosol specific diagnostic methods) and its physical relationship to AOD and absorption efficiency. We agree that temperature profiles would give a better idea of model performance in the stratosphere. We have included a temperature comparison (Section 4.4) earlier in the paper following the TOA flux plot.

[revised manuscript text omitted]

- Line 667-668: Here you can also refer to work of Clyne et al (2021) and Quaglia et al 2023
  - We have added references to AOD results from these publications.

Figures:

- Figure 1: The figure caption is very short, lacks information and is difficult to understand, e.g. What does the grey shaded region indicate? What does for "mode sensitivity tests" mean? Are you referring to the global sulfate burden?

  Thank you for pointing this out. We have included more information in the figure caption, including some information that was previously included in the beginning of section 4.1. It now reads: "**Figure 1:** Stratospheric sulfate burden – reported in Tg of the sulfur mass contribution – for model simulations, as well as HIRS and SAGE-3$\lambda$ remote sensing observations. The model data is processed to match the HIRS and SAGE-3$\lambda$ data coverage of 80°N – 80°S above the model lapse rate tropopause height. The sulfur component is determined by scaling modeled sulfate mass by the ratio of sulfur and sulfate mass weights (MW) such that $Tg\ S = Tg\ SO_4 * \frac{32.066\ g\ mole^{-1}}{MW\ Sulfate}$. In MAM4 of E3SM, sulfate is assumed to be ammonium bisulfate (($NH_4$)$HSO_4$; MW= $115.11$ g mole$^{-1}$) (Liu et al., 2012, 2016). In MAM4 of CESM2-WACCM, sulfate is assumed to be sulfuric acid ($H_2SO_4$; MW= $98.08$ g mole$^{-1}$) (Mills et al., 2016). Gray shading around the HIRS data represents systematic error of ~10% (±1.4 Tg aerosol) and the minimum and maximum aerosol composition bounds (59%–77% $H_2SO_4$)."

- Figure 2: Again, the gray shading?

  We have added the following sentence to the end of the Figure 2 caption: "The gray shading indicates ±11.3% uncertainty in AVHRR AOD."

- Figure 3: Citation of Quaglia et (2023) is misleading here, as it is a model intercomparison paper which uses the observational data for comparison and validation.

  We have changed this to the citation to the source datasets from the Quaglia et al., 2023 paper: Quaglia, I., Niemeier, U., Visioni, D., Pitari, G., Brühl, C., Dhomse, S. S., Franke, H., Laakso, A., Mann, G. W., Rozanov, E., and Sukhodolov, T.: Data from: Interactive Stratospheric Aerosol moels resonse to different amount and altitude of SO2 injections

during the 1991 Pinatubo eruption, https://doi.org/10.7298/mm1s-ae98, 2022. This maintains the source of the postprocessed data for reproducibility. We make sure to properly cite the instrument publications in section 3.

Table:

- Table 1: you can get rid of the third column and include the text in the table caption

  We have removed the nudging column and added this information to the table caption: "Table 1: Model details for the simulations used within this study. All simulations are run for 5 years (1989-1993) with 1989 discarded for aerosol spinup. All E3SMv2 simlations are run with U + V winds nudged to MERRA2 reanalysis data; CESM2-WACCM has U + V winds and temperature nudged to MERRA2 reanalysis."

Literature

- Barkstrom, B. R.: The Earth Radiation Budget Experiment (ERBE), Am. Meteorol. Soc., 65, 1170–1185, 1984.
- Barkstrom, B. R. and Smith, G. L.: The Earth Radiation Budget Experiment: Science and implementation, Rev. Geophys., 24, doi:10.1029/RG024i002p00379, 1986.
- Brodowsky, C., Analysis of the global atmospheric background sulfur budget in a multi-model framework, EGUsphere [preprint], https://doi.org/10.5194/egusphere-2023-1655, 2023.
- Clyne, M., et al..: Model physics and chemistry causing intermodel disagreement within the VolMIP-Tambora Interactive Stratospheric Aerosol ensemble, Atmos. Chem. Phys., 21, 3317–3343, https://doi.org/10.5194/acp-21-3317-2021, 2021.
- Free, M., and J. K. Angell, 2002: Effect of volcanoes on the vertical temperature profile in radiosonde data. J. Geophys. Res., 107,4101, doi:10.1029/2001JD001128
- Free, M., and J. Lanzante, 2009: Effect of Volcanic Eruptions on the Vertical Temperature Profile in Radiosonde Data and Climate Models. *Climate*, **22**, 2925–2939, https://doi.org/10.1175/2008JCLI2562.1.
- Kremser, S., et al.: Stratospheric aerosol – Observations, processes, and impact on climate, Rev. Geophys., 54, 1–58,https://doi.org/10.1002/2015RG000511, 2016.
- Marshall, L., et al..: Multi-model comparison of the volcanic sulfate deposition from the 1815 eruption of Mt. Tambora, Atmos. Chem. Phys., 18, 2307–2328, https://doi.org/10.5194/acp-18-2307-2018, 2018.
- Quaglia, I.et al.: Interactive stratospheric aerosol models' response to different amounts and altitudes of SO2 injection during the 1991 Pinatubo eruption, Atmos. Chem. Phys., 23, 921–948, https://doi.org/10.5194/acp-23-921-2023, 2023.
- Tilmes, S.et al.: Description and performance of a sectional aerosol microphysical model in the Community Earth System Model (CESM2), Geosci. Model Dev., 16, 6087–6125, https://doi.org/10.5194/gmd-16-6087-2023, 2023.
- Timmreck, C. et al.: The Interactive Stratospheric Aerosol Model Intercomparison Project (ISA-MIP): Motivation and experimental design, Geosci. Model Dev., 11, 2581-2608, doi.org/10.5194/gmd-11-2581-2018, 2018.

- Toohey, M., Krüger, K., Niemeier, U., and Timmreck, C.: The influence of eruption season on the global aerosol evolution and radiative impact of tropical volcanic eruptions, Atmos. Chem. Phys.,11, 12351–12367, doi:10.5194/acp-11-12351-2011, 2011.
- Weisenstein, D. K., Visioni, D., Franke, H., Niemeier, U., Vattioni, S., Chiodo, G., Peter, T., and Keith, D. W.: An interactive stratospheric aerosol model intercomparison of solar geoengineering by stratospheric injection of SO2 or accumulation-mode sulfuric acid aerosols,Atmos. Chem. Phys., 22, 2955–2973, https://doi.org/10.5194/acp-22-2955-2022, 2022.

---

## Author Comment (AC4)

We appreciate the reviewer's time and thank them for their thorough and thoughtful feedback. We have addressed your comments below and have added our own comments in RED:

Brown et al present a detailed and well-written documentation of their modifications and validation of E3SM, which I believe following some mostly modest textual changes can be ready for publication. Replicating the 1991 Mt. Pinatubo eruption is an important test case for climate models having interactive stratospheric microphysics. These models exist partly to understand the aerosol development and evolution during observed eruptions, and more commonly as a means to evaluate cases that lack aerosol data that otherwise could be prescribed into simpler model versions (e.g. eruptions of the distant past, solar radiation modification). As Pinatubo is the most clearly observed case of stratospheric aerosols altering global climate, being able to reasonably replicate its aerosol layer is essential for establishing credence in further experiments. The presented E3SM version is not flawless here and the main code changes are reboots from a related model (Mills et al, 2016). But documenting and testing this version is important for the interpretation of future model uses, remaining biases are honestly presented, and this does replicate aerosol properties slightly above average compared to other models (cf ISA-MIP). I believe it's important that the authors add discussion of how the reported issues can affect future uses of the model, and better delineate what is and is not verified by this evaluation, which is complicated by reliance on offline radiation calculations instead of full reliance on E3SM, as well as the use of nudged winds.

We have included discussion of the future uses of the model and have test the model in a historical context. We also noted that there was confusion as to why we used the offline Mie calculations and we hope that the changes we made below have helped clear this up.

I also think the authors can better explain the study's purpose and clarify some methodological choices, but if the authors make a reasonable effort to address these comments (detailed below) I think this can be suitable for publication relatively quickly, and so have selected minor revisions.

Main comments

The study reports multiple reasons to expect this model version will overstate stratospheric aerosol impacts on climate but does not include a paragraph explaining the ramifications for future uses.

We have included more information on future use in the introduction as well as the discussion.

I expect this validation study was made largely to be cited by future studies on non-Pinatubo experiments as a reason to have confidence in the model version, or could be even if not the intention. Biases are hence important to put into context. The two issues of overly small aerosols and ammonium sulfate optical constants (to represent sulfuric acid, which absorbs more LW) will both exaggerate the surface cooling, as well as the precipitation response to a cooler lower

troposphere. For a case of roughly similar stratospheric mass as Pinatubo (potentially including SRM), the results suggest this model version would give cooling – which is not presented here – on the edge of what should be acceptable as 'scientifically ready' (I approximate a net forcing bias of ~25-50% based on these two biases).

As addressed in comment responses below, we want to clarify that the ammonium sulfate representation only applies to the aerosol molecular weight and density, and the optical properties used here are those of sulfuric acid. Also, while there is an identified bias in aerosol size, we have added a TOA net forcing comparison that shows relatively similar behavior in E3SMv2-SPA to ERBS data and CESM2-WACCM, albeit underestimated. Without removal of model internal variability, we hesitate to quantify the effect that size changes have on the net TOA flux.

If this model version were used for an eruption multiple times the mass of Pinatubo – and maybe there is no intent for this, but an external user could presumably do so on their own – the net forcing bias would balloon, due to the shortwave and longwave effects becoming both very large and more closely offsetting one another.

This is a good point. We have included comparisons to historical simulations of the volcanic record (1850-2014) and note that the AOD from larger magnitude eruptions are well represented compared to the historical prescribed volcanic forcing datasets.

There's a long tradition in the volcano and SRM communities of using models with strongly exaggerated aerosol forcings as the basis for arguments on dire consequences (or detectability) of stratospheric aerosols, so it's important for caveats to be laid out at this early stage. Could the authors please add a paragraph (~5-8 sentences) to the Discussion section to guide future users on what their results imply, putting into a useful context some of the issues mentioned here? Also please make clear what is and is not verified in this study, as E3SM's stratospheric winds and radiation scheme being sidelined in these experiments complicates the ability for studies on future experiments to point to this study as validation of model reliability. I feel the authors have been transparent on their specific results, which is commendable, but just need to tie things together for future users and readers of upcoming works that use this model version.

Thank you for pointing this out. We have spent more time in the updated paper relating our results to preexisting multi-model comparison studies and Pinatubo specific modeling work. We have also added a paragraph to the discussion regarding recommended applications of this model. We also mention biases in E3SMv2 circulations and stratosphere that may impact the results. Based on the comparisons to observations and generally good agreement between long term free-running historical simulations and prescribed historical simulations, we believe this model has applications in the present day and historical contexts.

I also think this manuscript would be easier to appreciate if the authors add 3-5 sentences explaining the purpose of this study and model version in the Introduction, as currently this is extremely brief (lines 64-66). First off, to please explain more clearly why the focus is on the Pinatubo test case and why interactive aerosol microphysics is useful to represent. This model version is surely not meant to just replicate Pinatubo's impacts, for which interactive aerosols do

not need to be simulated because there are satellite retrievals of extinction and size retrievals. I offer some reasoning on why interactive models and the Pinatubo test case are important in this review's first paragraphs – maybe the authors' reasons are different or they are unable to reveal specifics, but it would be good to see more explanation here.

We have added a paragraph detailing the future work for which this model was developed. We have also added more in the introduction and discussion regarding the necessity for prognostic aerosol in future studies.

Second, is this version only for very specific stratospheric aerosol experiments by a small group, or can (and should?) anyone familiar with E3SM easily run the "SPA" version in the GitHub or make the modifications themselves for diverse stratospheric aerosol cases?

More advanced versions of this implementation are set to be released in the next version of E3SM, and our implementation was a temporary solution given that version's unavailability. As a result, we don't recommend modifying over our version, but we encourage others to use the code accompanying this publication if it would fit the purpose in the meantime as we think the validation supports its use.

 Third, the authors made an effort by retuning to get the troposphere right, so I'd like to see some statement on whether the authors view this as satisfactory for a full experiment including troposphere and stratospheric responses (e.g. historical runs), or if this is unknown as more validation would be needed. Anything the authors can contribute to give the reader a better sense of this model version's reason for existence and its suitable uses.

We have included information on historical as referenced in comments below.

The manuscript would be more useful if it included the magnitudes of shortwave, longwave, and net forcings, as well as stratospheric warming, all of which should be standard E3SM outputs, attainable as eruption years minus pre-eruption period. So why not show these or any results that are a function of the model's radiation code beside diffuse and direct radiation? Is this because the optical constants for sulfuric acid aren't well represented, or E3SM's radiation scheme isn't yet set up to feed in interactive stratospheric composition? It's understandable if only the aerosol properties are being verified within the scope of the present study, and possibly there are pertinent issues with the E3SM radiation scheme that are difficult to resolve. But to get no explanation is frustrating for the reader wanting to know whether or not this model can reliably replicate stratospheric sulfate's climate impacts, and even more so for anyone trying to figure out if they want to use this model. As this is a GMD article on stratospheric aerosols in a climate model, can the authors at least be upfront in the manuscript (~2-3 sentences in Methods and/or Results) about why they don't show the most climate-relevant outputs?

We agree that this makes the study more difficult to place in context. We have included TOA flux comparisons to ERBS as well as atmospheric temperatures for the year 1992 (see comments below).

Specific comments

38. Can the authors please word the CESM2-WACCM part of this sentence a bit better, as it would seem obvious these would give similar results. CESM2-WACCM is an odd model choice for comparison, given it has much of the same code as E3SM so doesn't serve as much of a benchmark.

Our goal was to show that, even without the more complete atmospheric chemistry, E3SMv2-SPA performed well. We have changed the line to read: "E3SMv2-SPA reasonably reproduces stratospheric aerosol lifetime, burden, and aerosol optical depth when compared to remote sensing observations. E3SMv2-SPA also has close agreement with the interactive chemistry-climate model CESM2-WACCM - which has a more complete chemical treatment - and the observationaly-constrained, prescribed volcanic aerosol treatment in E3SMv2"

50. It seems odd to tout this E3SM model as a useful alternative to an older model that uses much of the same code, including the same or very similar aerosol scheme. I would word this more logically or just focus on the E3SM-to-observation comparison here.

We do see how this statement is a bit odd. However, given that CESM2-WACCM has been used extensively in stratospheric aerosol studies and has a more advanced sulfur chemistry, we argue that this is a reasonable comparison as it indicates a similar applicability of E3SMv2-SPA. We have reworded this to read: "The overall agreement of E3SMv2-SPA compared to observations and its similar performance to the well-validated CESM2-WACCM makes E3SMv2-SPA a viable alternative to simulating climate impacts from stratospheric sulfate aerosols."

60. "net primary productivity of plants" or similar, as "productivity" alone is vague.

This sentence now reads: "…net primary productivity of plants"

68. I would say you're only "validating" against the observations, and separately that you're comparing against CESM, as CESM is just a model whereas the observations are – despite their own flaws and uncertainties – the standard approximation of truth. Can the authors also please briefly explain their choice of CESM2-WACCM here? Showing the ISA-MIP Pinatubo models (Quaglia et al, 2023, already cited in the manuscript) would have given a better impression of this model's performance against its peers. I understand that replicating an already verified model having many of the same features serves as a sanity check (and maybe some of the unique aspects of E3SM lead to improvement?), but this is worth a 1-line explanation in the text.

Thank you for the suggestion. We have separated this sentence into observational and model sentences which now read: ". As a first step towards this goal, this paper presents a validation of a prognostic volcanic aerosol implementation within the Department of Energy (DOE) Earth Energy Exascale Model version 2 (E3SMv2) (Golaz et al., 2022; Wang et al., 2020) against observational data from the Pinatubo eruption. Furthermore, E3SMv2 is compared with version 2 of the Community Earth System Model (CESM2) (Danabasoglu et al., 2020) with the Whole Atmosphere Community Climate Model version 6 (WACCM6; Gettelman et al., 2019), which shares many similarities between E3SM in its aerosol microphysical

parameterizations but has more advanced atmospheric chemistry. This is to help identify any performance issues associated with a simpler chemical treatment in E3SMv2 and to serve as further validation of our implementation."

70. Saying "most climate models" use GloSSAC isn't accurate. For some historical eruptions it's an option, but then there's the CMIP dataset cited elsewhere in this study (SAGE-3λ). And for eruptions in the distant past or hypothetical eruption cases, simplified forcing generators like Easy Volcanic Aerosol (Toohey et al., 2016) are now the standard option. I would just amend this into a more general statement.

*Toohey, M., Stevens, B., Schmidt, H., & Timmreck, C. (2016). Easy Volcanic Aerosol (EVA v1. 0): an idealized forcing generator for climate simulations. Geoscientific Model Development, 9(11), 4049-4070.*

Thank you for pointing this out. We have changed this section's introduction to be more general and it now reads: "When simulating large-magnitude explosive volcanic eruptions, some climate models use prescribed volcanic forcing datasets as a way reduce computational demand and to avoid uncertainties in prognostic aerosol formation. These datasets can estimate forcing based on satellite data, ground based retrievals, ice core records, and other other volcanic evidence (Toohey et al., 2016). One such dataset is the Global Space-based Stratospheric Aerosol Climatology (GloSSAC), which prescribes aerosol properties from a compilation of satellite, airborne, and ground based observations (Kovilakam et al., 2020; Thomason et al., 2018). While GLoSSAC and other prescribed datasets…"

82. "more complete approach", maybe. But mostly these interactive models are used for cases where we lack suitable observations, or are isolating a particular microphysical effect (since we observe properties, but not processes). I think the need for good interactive aerosol models can be described better, which would also help tout why the rigorous technical work here is useful (see main comments).

We follow your recommendation and added a line to communicate the purpose of these interactive models: "Prognostically modeling the formation and evolution of sulfate aerosol from sulfur dioxide ($SO_2$) injected into the stratosphere is an alternative, more complete approach for simulating volcanic eruptions, with a variety of methods for representing sulfate aerosol mass, size, and number**. This approach can serve to recreate conditions where observations are lacking as well as help elucidate microphysical processes that contribute to aerosol properties. Aerosol forcing is also more dynamic in prognostic simulations given that it is not tied to the spatial pattern of the prescribed forcing. This allows the for simulation of evolving aerosol forcings and feedbacks in fully-coupled model simulations or ensemble sets.** The simplest…"

107. Please summarize around this line the model version's purpose (see main comments). Can anyone use this scheme? For what purposes is it suitable and for what experiments does this validation apply?

We include a copy of this code in the data availability section and include runscripts and datafiles to run it. We encourage others to run the code with this publication as a reference and to reach out with any questions. We have added the following paragraph to the introduction to address model purpose: "Here we present a new stratospheric prognostic aerosol capability within E3SMv2 that modifies the microphysical treatment of stratospheric aerosol in the 4-mode Modal Aerosol Module (MAM4; Liu et al. (2012, 2016)) to enable simulation of the evolution of volcanic stratospheric aerosols and their properties. Similar to Mills et al. (2016), we add a stratospheric specific sulfate treatment to compliment the preexisting MAM4 chemistry and physics (default MAM4 includes the oxidation of SO2 to form sulfate aerosol, their further growth through condensation and coagulation into larger aerosol size modes, sedimentation of these aerosols, and removal via wet and dry deposition). This model parallels work by Hu et al. (In Prep.) on a 5-mode Modal Aerosol Module (MAM5) that incorporates more complete sulfate chemistry and an additional volcanic sulfate mode in E3SMv2. The validation of our implementation presented here will support forthcoming detection and attribution studies of societally relevant climatic impacts from stratospheric aerosols in free-running coupled climate simulations with varying volcanic source characteristics. By enabling dynamical consistency between transport, aerosol distribution, microphysical properties, and eruption characteristics (e.g., impact magnitude, timing and location), this modeling capability facilitates the development of multivariate and multi-step attribution studies sensitive to spatio-temporal evolution (Hegerl et al., 2010). As future studies with this model capability will be free-running, they also enable better differentiation of the role of the climatic state on the detected and attributed impact."

110-111. I'm not convinced that this study pays "more attention to […] global and regional climate impacts" than the cited Mills et al 2017 study. That study actually shows radiative forcings and surface temperature responses, while this one does not. I think there are novel aspects to this study – the radiation sections are for instance quite different from the two Mills et al studies – and that the authors should more accurately represent their uniqueness here.

Thank you for this comment. We have changed it to read: "Here we detail the aerosol evolution and examine how model representations of the aerosol size distributions are related to global and regional radiative impacts at the surface and in the stratosphere."

158. It looks from Table S1 like the mode size cutoffs are exactly the same as in the Mills et al study, and the only difference is the dust and sea salt tuning. Can the authors please make this clear in the manuscript text?

We have added the following line in the first paragraph of section 2.1.2: "The major modifications to MAM4 include (1) the transfer of aerosol mass and number from the accumulation to coarse mode to increase aerosol size and represent the rapid aerosol growth following the Pinatubo eruption and (2) adjustment of the coarse mode and accumulation mode $\sigma_g$ and minimum/maximum geometric mean diameters to increase aerosol lifetime. **We note that these changes make the E3SMv2-SPA modal widths and size cutoffs identical to those in CESM2-WACCM6**."

180. It's a bit odd the authors don't show any maps of tropospheric aerosol to back these statements, though I'm willing to accept this is outside the range of this study. Can the authors at least make a statement if the model as modified here is ready for experiments where both the troposphere and stratosphere are important, or if further validation effort is needed then please say so. Is there any reason *not* to use this over the current E3SM, beside maybe the unavailability of long control runs? As is I feel anyone who reads this thinking they may want to use the model would be pretty lost.

We don't show the plots of our tuning, but we do mention the global average results of the tuning on dust (0.0281) and total AOD (0.1617) compared to observations (0.02-0.02 and ~0.17, respectively). I have included the pre-tuned and post-tuned plots below:

**Before Dust/Seasalt tuning,** E3SMv2-SPA for 5-yr present day simulation (perpetual 2010 forcing)

[Figure]

**After Dust/Seasalt tuning,** E3SMv2-SPA for 5-yr present day simulation (perpetual 2010 forcing)

[Figure]

As to the model performance in both troposphere and stratosphere, we have run two multiple 175-year historical simulations that show reasonable performance in total AOD and 2m surface temperatures when compared to 5-ensemble member runs of E3SMv2 with prescribed volcanic aerosol. We have added a paragraph to reference these simulations and results: "The tuning of coarse mode aerosol does not appear to significantly affect global measures of the simulated tropospheric climate. Two fully-coupled, 164-year historical simulations (1850-2014) were run with E3SMv2-SPA, initialized from years 50 and 100 of a 100-year pre-industrial spin-up simulation. These simulations show total AOD (Fig. S2), 2m surface temperatures (T2m ; Fig. S3), and global radiative balance (Fig. S4) that track the five-member E3SMv2 historical simulations with prescribed volcanic forcing from Phase 6 of the Coupled Model Intercomparison Project (CMIP6; (Golaz et al., 2022)). Differences in atmospheric modes of variability (e.g., El Niño Southern Oscillation (ENSO; (Trenberth, 1997))) due to internal variability affect T2m during the Pinatubo period (Fig. S2, S4) but interval variability would average out if a mean were taken over more ensemble members."

204. We're only seeing results with nudged winds, so have no idea whether to trust E3SM's stratospheric dynamics. I get that this is outside the scope of this study, but have these been verified? If known, could the authors provide a line on how E3SM's stratospheric winds perform and maybe a citation on this? Stratospheric dynamics plot from historicals?

We added the following note at the end of this paragraph: "Note that both E3SM and CESM2-WACCM6 have a variety of stratospheric dynamics biases (e.g., Gettelman et al., 2019)) that are avoided here through atmospheric nudging. An upcoming publication on E3SM stratospheric processes details a variety of biases in E3SM that may impact free-running volcanic eruption modeling, including a weak-amplitude tropical Quasi-Biennial Oscillation which oscillates too frequently and a weak Brewer-Dobson circulation (Christiane Jablonowski, personal communication)."

214. What about Cerro Hudson and the other non-Pinatubo eruptions mentioned later in the text? Please say at least that these are included based on the same SO2 dataset, if so.

We added the following: "The VolcanEE3SMv3.11 dataset contains estimates of $SO_2$ from volcanic eruptions on a 1.9x2.5-degree latitude by longitude grid, with 1 km altitude spacing from the surface to 30 km. **In our period of interest (1991-1993) this includes the Pinatubo, Hudson, Spurr, and Lascar eruptions**."

219. I feel "E3SMv2-presc" would be more suitably given a fuller (1-2 line) mention in the text, as it's odd to leave all but a brief mention of its existence to a table.

We added the following 2 sentences before the referenced paragraph and Table 1: "A prescribed volcanic forcing simulation (E3SMv2-presc) is run in addition to the prognostic volcanic aerosol simulations. This simulation uses the default prescribed forcing dataset in E3SMv2 (GLoSSAC V1) and allows for an additional validation of prognostic aerosol model performance where observational data are lacking."

244. Please specify that it's 75% H2SO4 + 25% H2O "by mass", as by volume would be different.

We added "by mass" after H2O.

278. As above, 75% H2SO4 "by mass"

We added the following to address your concern: "…and a sulfate refractive index corresponding to 75% $H_2SO_4$ aerosol **mass** composition"

290. Optional, but it may or may not be worth mentioning the instrument saturation issue that occurred during Pinatubo. This was for instance mentioned in the already cited Quaglia et al, 2023 study.

We added a sentence to this effect in the section 4.1: "The E3SMv2-SPA tends to overestimate aerosol burden compared to HIRS and SAGE-3$\lambda$ in the 6 months after Pinatubo

but agrees well with the slow decay reported in observations during 1992. **In the four months following Pinatubo, models agree best with HIRS, likely due to saturation issues identified in SAGE-II limb-occulation data (Russell et al., 1996; Sukhodolov et al., 2018; Quaglia et al., 2023)**. From 1992 onward, stratospheric mass burden in E3SMv2-SPA agrees the best with SAGE-3$\lambda$, which reports higher burdens in 1993 than HIRS."

312. The ammonium sulfate assumption deserves more description. The validation here barely uses/tests E3SM's actual radiation code (mostly relying on external Mie calculations instead), but this would be an issue for future uses of the model that do, so I think deserves more mention. First off, the imaginary refractive indices are higher for sulfuric acid than ammonium sulfate (see for example a comparison in Gosse et al, 1997). This would bias low the longwave effect, driving the model to cause too much surface cooling. In our own evaluations (not published), we found switching Pinatubo aerosol from ammonium sulfate to sulfuric acid optics increased the longwave forcing by ~50% and reduced the net forcing by 10-15%. Second I wonder if the ammonium sulfate assumption increases density and fallout of the aerosols, which would affect the aerosol properties shown here? Can the authors please comment on this and add a line or two to the text to guide anyone interested in using this model?

Gosse, S. F., Wang, M., Labrie, D., & Chylek, P. (1997). Imaginary part of the refractive index of sulfates and nitrates in the 0.7–2.6-µm spectral region. Applied optics, 36(16), 3622-3634.

We hope the confusion related to the model optics calculations was resolved by our responses to your comment on line 516 (further in this document). But to briefly clarify, the model radiation code is used to calculate radiative flux, longwave heating, temperature change, and diffuse/direct radiation. The offline mie calculations are mainly used to enable direct comparison between observations and models.

While the models assume sulfate density and mw to be ammonium bisulfate, they also assume the refractive index is that of H2SO4. Therefore, the model prescribed refractive index is not thought to introduce the bias you mention above.

E3SM and CESM use the assumption of ammonium bisulfate density and mw throughout the atmosphere, and this assumption will impact removal as you point out. How much, I am not sure. But it would likely decrease the aerosol lifetime due to the higher density of ammonium bisulfate than H2SO4, increasing AOD, effective radius, and impacts on radiative flux. The higher density may also influence aerosol size for the same mass of sulfate, leading to smaller aerosol and potentially decreasing scattering (depending on effective radius) and decreasing longwave absorption.

We have added a line in the discussion to address density effects: "There are also inherent model assumptions that may also be contributing to the aerosol small bias. One is the treatment of sulfate aerosol density as that of ammonium bisulfate (1.7 kg m$^{-3}$) as opposed to sulfuric acid (~1.6 kg m$^{-3}$ (Seinfeld and Pandis, 2006)), which may be impacting aerosol R$_{eff}$ through more efficient removal of coarse mode aerosol and the formation of denser, smaller aerosol."

We have also added a few sentences to the end of the Aerosol Size distributions section: "…We present a couple of possibilities for this size bias. One is that the aerosol density is too large due to the model assumption of ammonium bisulfate density of 1.7 kg m$^{-3}$ instead of the sulfuric acid density of ~1.6 kg m$^{-3}$ (Seinfeld and Pandis, 2006). This may bias the aerosol small while also contributing to more rapid removal of coarse mode aerosol. Another is the lack of van der Waals forces in both E3SMv2 and CESM2-WACCM. This intermolecular attraction has been shown to aid in the coagulation of smaller particles, enhancing R$_{eff}$ (English et al., 2013)."

323. Please add a line explaining the improvement from E3SM-PA to E3SM-SPA for sulfur burden.

We were using aerosol lifetime to refer to burden and attempt to make this a little more clear with the following modifications: "The improved aerosol **burden – and thus, aerosol lifetime –** in the stratosphere is mainly due to our modifications to the coarse mode $\sigma_g$ in E3SMv2-SPA. **While E3SMv2-PA reaches a similar peak in sulfate burden**, the underestimated aerosol **burden following Pinatubo** in E3SMv2-PA is mainly caused by too wide an aerosol number distribution, causing fast sedimentation of the larger coarse mode particles in the upper tail of the distribution."

388. Please clarify here whether Cerro Hudson and the other small eruptions are included in all simulations.

We added the following to the first paragraph of section 4.3: "Aerosol growth continues until approximately mid 1992 when R$_{eff}$ peaks, lagging peak values in other metrics such as mass burden and AOD. **The smaller magnitude eruptions of Cerro Hudson, Spurr, and Lascar also contribute to an increased R$_{eff}$ over this period, with more of an impact in near-source regions**."

392. An important question is *why* the aerosol size is persistently too small in all models used here. It could be that none of the models include enhancement of coagulation by Van der Waals forces. This was reported to drive a sizable increase in aerosol size in a paper by English et al (2013) that is already cited, and may be worth mentioning in this paragraph (and checking that it isn't in CESM2-WACCM as used here). For reference, the equations needed to add this are presented in more detail in a study by Sekiya et al (2016). It's very possible there are other factors, and for one I wonder if the mass of ammonium sulfate (35% higher H2SO4's true mass) is connected to gravitational settling in a way that would make the coarse particles fall out faster than they should. I'm not sure this needs to be discussed within the manuscript (though it could be helpful to someone wanting to improve the model further), but I hope the authors can share a bit of thinking on this.

*Sekiya, T., Sudo, K., & Nagai, T. (2016). Evolution of stratospheric sulfate aerosol from the 1991 Pinatubo eruption: Roles of aerosol microphysical processes. Journal of Geophysical Research: Atmospheres, 121(6), 2911-2938.*

This is a good point and we add a few lines regarding aerosol size in the size distribution section:

"… We present a couple of possibilities for this size bias. One is that the aerosol density is too large due to the model assumption of ammonium bisulfate density of 1.7 kg m$^{-3}$ instead of the sulfuric acid density of ~1.6 kg m$^{-3}$ (Seinfeld and Pandis, 2006). This may bias the aerosol small while also contributing to more rapid removal of coarse mode aerosol. Another is the lack of van der Waals forces in both E3SMv2 and CESM2-WACCM. This intermolecular attraction has been shown to aid in the coagulation of smaller particles, enhancing $R_{eff}$ (English et al., 2013). …"

And the discussion of size:

"… While choice of bulk, sectional, or modal aerosol representation will impact $R_{eff}$ (English et al., 2013; Laakso et al., 2022; Tilmes et al., 2023), the presence of this $R_{eff}$ underestimation is in modal as well as sectional models with similar injection parameters (e.g., ECHAM6-SALSA, SOCOL-AERv, UM-UKCA (Quaglia et al., 2023; Dhomse et al., 2020)) indicates that it isn't attributed solely to the choice of aerosol microphysical approach. We present a variety of potential reasons and solutions for why the model simulates aerosols that are too small, though testing these is beyond the scope of our study. As with AOD, higher injection levels and plume lofting could increase $R_{eff}$ by increasing tropical confinement (Clyne et al., 2021). Improvement of the stratospheric dry bias in E3SMv2 may improve this disagreement by increasing aerosol size due to water uptake. There are also inherent model assumptions that may be contributing to the aerosol small bias: 1) the treatment of sulfate aerosol density as that of ammonium bisulfate (1.7 kg m$^{-3}$) as opposed to sulfuric acid (~1.6 kg m$^{-3}$ (Seinfeld and Pandis, 2006)), which may be impacting aerosol $R_{eff}$ through more efficient removal of coarse mode aerosol and the formation of denser, smaller aerosol; 2) the lack of interparticle van der Waals forces in both E3SMv2 and CESM-WACCM (though recent iterations of WACCM coupled with the CARMA sectional aerosol model do include these forces (Tilmes et al., 2023)), which have been shown to drive aerosol nucleation and lead increased peak $R_{eff}$ following Pinatubo-sized eruptions (English et al., 2013; Sekiya et al., 2016); 3) the nucleation scheme used in these models may be overestimating nucleation rates by 3-4 orders of magnitude (Yu et al., 2023), leading to more numerous smaller particles. …"

395. I wonder if neglecting volcanic ash also has a size influence through lofting, as Stenchikov et al showed in a more recent paper that including ash is the only way to get the aerosol plume to form at an appropriately high level of the stratosphere. This could conceivably slow coagulation by spreading the aerosol out vertically, which would keep aerosol smaller (though I haven't seen this tested). Maybe not worth mentioning in the text, but if the authors do further tests it could be worth considering.

*Stenchikov, G., Ukhov, A., Osipov, S., Ahmadov, R., Grell, G., Cady-Pereira, K., ... & Iacono, M. (2021). How does a Pinatubo-size volcanic cloud reach the middle stratosphere?. Journal of Geophysical Research: Atmospheres, 126(10), e2020JD033829.*

This is an interesting prospect that we will keep in mind. I can't think of a reasonable way to validate this with the current simulations we have given it does not have volcanic ash. I do think the aerosol lofting due to ash as shown by Stenchikov et al. is an important concept to address for AOD comparisons with AVHRR, and we have added a statement to the discussion on AOD.

511. Clarify that the curves are the size modes, please: "dN/dlog D size modes (curves)" or similar

We added your recommendation to the figure description for this figure.

516. Please explain in the first paragraph why you chose to present output from offline Mie code instead of standard model output involving E3SM's radiation scheme. Are the radiative fundamentals worth an in depth dive here? Is this something novel compared to other model validations? I think this is acceptable, and the benefits of this work are worth advertising better. However, there's certainly a drawback that we aren't given enough information from the actual model to have confidence in its ability to produce reliable shortwave scattering (or other radiative effects), which is really what I'd expect in an interactive aerosol model validation study. So a brief explanation is expected.

We added the following sentence to explain our use of offline mie code:

In section 4.7: "The $R_{eff}$ can be used to characterize the evolving aerosol size distribution, which conveniently can be used in the offline calculation of single particle optical properties using Mie theory (see Appendix B). **We choose to use offline Mie code for both model and observations to allow for a direct comparison between optical properties derived from modelled and observed $R_{eff}$.**"

In section 4.7.1: "**We diverge from the idealized $Q_s$ calculations in section 4.7 to analyze modeled radiation diagnostics in the atmosphere, calculated via the online model Mie code and a** two-stream approximation for calculating multiple scattering in the atmosphere (Iacono et al., 2008; Neale et al., 2012). **The latter** calculates diffuse and direct radiation at the surface from…"

In section 4.8: "…Figure 13 shows the $Q_a$ calculated **offline** from **model and in-situ** $R_{eff}$, with a dark red line indicating the wavelength of peak…"

In section 4.8.1: "The absorption of outgoing longwave radiation in the stratosphere results in local heating of the aerosol layer.  Figure 14 shows the longwave heating rate (LWH; K day$^{-1}$) due only to Pinatubo aerosols and normalized to 1990 monthly means over the same regions as Fig. 11 and 12. **As for Fig. 11 and 12, Fig. 14 is calculated from the online model radiation code**. Higher initial global (Fig. 14a)"

We also add the word "Modeled" to sections 4.7.1 and 4.8.1 to enforce that these diagnostics are from the working model radiation code.

516. Please also remind the reader that this output is from an offline Mie scattering routine and perhaps link them to Appendix B.

We added a reference to Appendix B in the first sentence: "The $R_{eff}$ can be used to characterize the evolving aerosol size distribution, which conveniently can be used in the offline calculation of single particle optical properties using Mie theory (see Appendix B)."

555. What are dotted vs solid curves in Fig. 7? Different modes?

Sorry for the confusion. We added a comment in the figure caption: "Figure 10: Monthly stratosphere scattering efficiency ($Q_s$) using effective size parameter ($x_{eff} = 2\pi R_{eff}/\lambda$) and sulfate refractive index at 0% relative humidity (Hess et al., 1998). **Solid lines are scattering efficiencies over 41°N 105°W and dotted lines are scattering efficiencies over 20°S–20°N.**"

We also added some clarification to the main text: "Figure 10 shows monthly $Q_s$ generated from averaged stratospheric $R_{eff}$ at a midlatitude site (**solid line**) (Fig. 6c) and the tropics (**dotted line**) (Fig. 6d)."

559. Please add a line here to tell the reader why we should care that the model can replicate diffuse and direct radiation breakdown. The relevance for plants is listed extremely briefly in the Introduction, but should be here in slightly more detail (mentioning the influence of radiation type on shadow experienced by plants and photosynthesis, for instance).

We added a couple of sentences to add more context for the diffuse/direct radiation comparison: "The differences in $R_{eff}$ between the different model and observational datasets ultimately affect diffuse and direct radiation at the surface by changing AOD and other aerosol optical properties. In scattering incoming shortwave, a small fraction of light is scattered back to space reducing the amount of energy incident to Earth. The forward scattered SW radiation increases the diffusivity of incident radiation. **Increased diffuse radiation can differentially impact certain crop yields by decreasing the direct radiation on sunlit leaves and increasing radiation exposure for shaded leaves. While having an overall negative influence on crop yield, this may influence some crop types more negatively (e.g., maize) than others (e.g., rice, soybean, and wheat) (Proctor et al., 2018).**"

564. So unlike all other radiation output in this study, here it actually uses E3SM output and is not just fitting aerosol properties into a radiative transfer model? Why not just show the actual shortwave, longwave, and net forcings, which are the main indicator of stratospheric aerosol impacts on climate? Wouldn't this be worth being shown in this GMD study, even if there are some remaining issues?

We have added an additional TOA flux figure and section (4.3. TOA radiative flux) to address your concern. We also include clear-sky and aerosol-only model diagnostics to identify biases arising from our subtraction of year 1990 conditions:

**3.3 Top-of-atmosphere radiative flux**

**3.3.1 ERBS**

[revised manuscript text omitted]

560. The transition between sentences feels like an incomplete comparison. I would amend it to "More substantially, the forward scattered SW […]" or similar.

Thank you for the recommendation. We have added "More substantially…" to the beginning of this sentence.

581. Certainly these quantities are linked, but I think AOD being a "good indicator" of diffuse radiation is unrealistic given the curves have different shapes and can peak months apart.

This is a good point. We have rephrased the sentence to read: "Diffuse radiation is not only attributed to aerosol size and material properties – as is $Q_s$ in Fig. 7 – but is also related to the aerosol number concentration. This is also true of AOD, which is a cloud-free, aerosol-specific input into the multiple scattering radiation calculations in the model. AOD, which accounts for both number and $Q_s$, gives a cleaner signal of the aerosol influence. Figure 9 shows AOD over the same spatial regions as Fig. 8. While peaks in AOD and diffuse radiation differ, the intermodal relationships are very similar between the two comparisons. …"

613. Could the authors please add how the weaker "wavelength dependence of Qs" relates to there being a weaker longwave absorption Reff sensitivity than for shortwave reflection? This is best seen with an x-axis of Reff for fixed wavelengths, as in Fig. 1a of Lacis 2015, but is directly related to Figs. 7 & 10 here via the size parameter. I think the authors' method of going directly from aerosol properties to the fundamentals of radiative effects is informative, but as this is a GMD climate model validation I think they could better connect this to radiative forcing and climate response. I recommend they take a look at this short Lacis paper that very succinctly puts Qa and Qs into context.

*Lacis, A. (2015). Volcanic aerosol radiative properties. Past Global Changes Magazine, 23, 51-51.*

Thanks for this comment and reference. We added the following to this line: "The wavelength dependence of $Q_a$ is weaker than $Q_s$ (proportional to $x_{eff}$ (i.e, $\lambda^{-1}$)) **due to a weaker longwave absorption sensitivity to $R_{eff}$ (Lacis, 2015), and $Q_a$** is strongly tied to the imaginary part of sulfate refractive index at 0% relative humidity ($n_{Hess}$; Appendix B) (Hess et al., 1998)."

613. It's worth pointing out in the text that Qs and Qa apply to different frequencies and this should be noted when comparing Figs. 7 & 10 (citing the vertical lines). And it would be nice to get a 1-line explanation of the most clear difference between these figures: in Fig. 7 (Qs) differences are right-left, while in Fig. 10 (Qa) they are up-down.

We have added a couple of sentences to this paragraph to address your comment: "While the Pinatubo eruption resulted in a net climate cooling effect due to increased scattering, it also contributed to stratospheric warming through the absorption of outgoing longwave radiation within the aerosol layer (Kinne et al., 1992). Fig. 13 shows the $Q_a$ calculated offline from model and in-situ $R_{eff}$, with a dark red line indicating the wavelength of peak terrestrial black-body irradiance (10 μm; $\lambda_{earth}$). **While sulfate scatters strongly at visible wavelengths ($\lambda_{solar}$ in Fig. 10 and 9),** it is not an effective absorber at visible wavelengths ($Q_a = 0$ at $\lambda_{solar}$) and the determining factor in stratospheric heating is the magnitude of $Q_a$ at

$\lambda_{earth}$. The wavelength dependence of $Q_a$ is weaker than $Q_s$, (proportional to $x_{eff}$ (i.e, $\lambda^{-1}$)) and is strongly tied to the imaginary part of sulfate refractive index at 0% relative humidity ($n_{Hess}$; Appendix B) (Hess et al., 1998). The proportionality to $x_{eff}$ is denoted in Fig. 13 by the linear increase in $Q_a(\lambda)$ with increasing $R_{eff}$; the dependence on the imaginary part of $n_{Hess}$ is reflected in the unchanging pattern in absorption magnitudes for all datasets and times. **Generally speaking, $Q_s$ is characterized by changes in the horizontal and is more sensitive to aerosol size fluctuation (Fig. 10 and 9), while $Q_a$ shows changes in the vertical and is less sensitive to aerosol size fluctuation."**

626. Modeling studies can add H2Ov during an eruption to simulate (poorly constrained) direct volcanic emission of water vapor, which would hydrate the stratosphere. May or may not be worth mentioning here that this might have alleviated the issue.

We added the following paragraph to address injection of volcanic water vapor and its possible impacts: "It is unclear whether injecting volcanic $H_2O$ would help offset the stratospheric dry bias in E3SM. Abdelkader et al. (2023) showed that the injection of $H_2O$ for a 20 km Pinatubo-like injection increases sulfate mass and stratospheric AOD by ~5%, but at the low temperatures of the lower stratosphere almost all of this water vapor freezes and sediments out. Retention of water vapor in the stratosphere depends on injections at higher altitudes where temperatures are warmer. Given that E3SMv2-SPA already tends to produce higher sulfate burdens compared to other simulations (Fig. 1), it is possible that the additional of sulfate aerosol mass attributed to the water vapor injection would bias the aerosol sulfate content high compared to observations, even as it would increase the aerosol size. Furthermore, the lower injection height of our simualtions (20 km) and a cold-point tropopause that is too cold in E3SMv2 (Christiane Jablonowski, personal communication, 2024) would aid in the rapid removal of most of the injected water, reducing the effectiveness of the injection on aerosol properties."

626. (but really the Supplement) The 3 panels in Fig. S11 appear too close together, partly covering the panel titles ("global", etc). I appreciate that the authors show this data.

Fixed.

637. As in the Fig. 7 caption, the Fig. 10 caption doesn't say what the solid vs dotted curves are.

We added the following to the now Fig. 13 caption: "Figure 13: Monthly stratosphere absorption efficiency ($Q_a$) using effective size parameter ($x_{eff} = 2pR_{eff}/\lambda$) and sulfate refractive index at 0% relative humidity (Hess et al., 1998). **Solid lines are scattering efficiencies over 41°N 105°W and dotted lines are scattering efficiencies over 20°S–20°N.** "

641. What is "normalized" here? Maybe this is the same normalization as earlier in the study, but please define it here or at least cite that it is as previously stated.

We have added a comment clarifying this statement: "Figure 13 shows the longwave heating rate (LWH; K day$^{-1}$) due only to Pinatubo aerosols and normalized to 1990 monthly means over the same regions as Fig. 11 and 12."

647. Since – as stated – E3SMv2 has no LWH, please remove it from the legend of Fig. 11. I found myself looking for it but it simply isn't there.

We have removed E3SMv2-presc from the legend.

641. Where do these longwave heating rates come from? The expectation would be that these are from E3SM itself but this does not appear so. Is it a simple equation involving the Qa's from the previous section? It could be worth showing this, but more definitely there should at least be a small description.

These do come from the model, and we had used the offline calculations to enable comparison with in-situ data. We have added some clarifications as to the difference between offline absorption efficiency calculation and online radiation diagnostics (see previous comment).

649. Radiative heating rates cannot be directly observed, though there are observations of stratospheric temperature increases. The already cited Mills et al., 2016 study includes a comparison between modeled and radiosonde post-Pinatubo stratospheric temperatures. It may or may not make sense to cite this component of the Mills study here.

Given the implications for our findings on temperature, we have added additional temperature analyses and a new section (4.4) that shows annual average atmospheric temperature profiles and level means for the year 1992. The following are the additions to the paper including observational datasets, figure + table, and figure results:

[revised manuscript text omitted]

667. I wonder if the mass burden differences involve the type of sulfate in each model. Do both E3SMvs-SPA and CESM2-WACCM use ammonium sulfate? Could this bias the aerosols heavy?

Both E3SM and CESM set sulfate density and MW as that of ammonium bisulfate. WACCM includes $H_2SO_4$ equilibrium pressures and water update parameterizations in the stratosphere, but this is separate from how the actual aerosol mass is characterized. Therefore, I don't think this will affect agreement between models.

689. I'd prefer if the wording were "this suggests that the models will overestimate" instead of "indicates" they "may" do so, as the results show pretty clearly to expect a bias in this direction.

We have included your recommended change.

695. Please add a paragraph on to what extent direct use of this model – or methods based on the aerosol properties the model simulates – could result in biased evaluations of climate responses, along with other statements that can aid interpretation of results from future uses of this scheme (see main comments). The authors could make a recommendation only to use E3SMv2-SPA for very similar experiments as performed here (Pinatubo-sized eruptions, nudged winds, little reliance on E3SM's radiation scheme), as certainly the more dissimilar the experiment the less the validation applies. But I expect there could be interest in further uses, so the authors would be well suited to preemptively give guidance (e.g. what uses are suitable, what biases are pertinent, are climate responses trustworthy).

We have added multiple statements to address this. One is in the introduction regarding applicability and future use, another is in section 2.1.2 where I touch on the good performance of the model in a historical context, and the last is in the discussion.

703. Which refractive indices are "assumed across observations and models"? Are these ammonium sulfate or sulfuric acid? Is there a reference to cite?

These are sulfuric acid refractive indices. We have included a reference to Hess et al., 1998 and Appendix B to clarify this statement: "Here, the same refractive index is assumed across observations and models to make them more comparable (Hess et al., 1998; Appendix B)."

765. Reff is area mean radius. Please reconcile this.

Thank you for pointing this out. This was a typo we found after submission and have rectified. This section and equation now read:

"The $m_{sulf}$ and $m_{wat}$ are summed across the three aerosol modes. The soluble aerosol volume from sulfate and water across modes ($V_s$) is calculated as:

$$V_s = \frac{m_{sulf}}{\rho_{sulf}} + \frac{m_{wat}}{\rho_{wat}} \tag{5}"$$

780. The GitHub for E3SMv2-SPA doesn't show any indication of being particularly for this stratosphere-optimized version. Is there a particular git branch that should be used?

This is the GitHub for the project under which this model code was developed. The master branch of this code contains the prognostic aerosol version. We have changed our data availability statement to reference a copy of this code on Zenodo: "The model code base used to generate E3SMv2-SPA and E3SMv2-PA – along with information for how to access the publicly available CESM2-WACCM code base – can be found on Zenodo at https://doi.org/10.5281/zenodo.10602682."

Technical comments

38. The comma that precedes "CESM2-WACCM" seems unnecessary/odd.

Removed comma.

40. "too small of accumulation […] mode" to "overly small accumulation […] mode".

Thank you for the suggestion. We changed this line in the text.

120. "Simlations" to "Simulations"

Fixed.

138. Totally optional, but maybe spelling out the experiment names could help the reader remember what's what? I assume "PA" is "prognostic aerosol"?

We have changed the first sentence to read: "In the default, prognostic volcanic aerosol simulations (E3SMv2-PA) , prescribed volcanic extinction is removed and the sulfate aerosol precursor, $SO_2$, is emitted in the stratosphere."

154. Also optional, but as above, is SPA "stratosphere-optimized prognostic aerosol" experiment or something similar? Could be better to spell this out than letting the reader wonder.

It is not present in the first sentence, but in the second sentence of this paragraph we do spell out the definition of this model acronym.

178. "Dg,low" has an obvious meaning given "Dg,hi" is defined above, but this really should also be spelled out before use.

We have defined these variables at their first occurance, and have change Dg,hi to Dg,high to make its meaning more obvious in later uses.

201. The line has some grammatical issues. I would switch "2022) where" to "2022, with", and then in the following line switch "use" to "using".

We have made these changes.

250. (and also 253, 315, etc) Please just ensure to fix all "SAGE-3l" mentions to "SAGE-3λ" by the time this is published.

Fixed.

301. "from the global to the microphysical" sounds like the authors are starting with global and ending with microphysical, where really everything's jumbled together. Not critical, but maybe can be reworded to avoid this confusion ("across scales global to microphysical" or similar).

We appreciate your revision and have included this in the paper.

304. Reff should be defined within the text before being used here.

We have defined '$R_{eff}$' in the introduction and redefine in the section 2.4.

304. "small bias" to "bias toward small size" or similar.

We have changed this line to read: ". Regional comparisons to remote and in situ observations of stratospheric $R_{eff}$ (Section 4.3) identify a model **bias toward smaller sizes**,"

381. "mid 1992" to "mid-1992"

Fixed.

390. "Identical" to "identical"

Fixed.

392. "the models" to "these models"

Fixed.

402. "Theselarger" to "These larger"

Fixed.

408. "1993-02" looks a bit awkward in the text (like 1993-2002). "February of 1993" looks nicer, though I'm fine either way as "1993-02" is what's stated on the figure for brevity.

Agreed. Changed to "February of 1993"

520. Please rectify that "xeff" is not defined in the text before its use here. It is defined in a figure caption later in the paper.

This sentence now reads: "$Q_s$ is proportional to the effective size parameter ($x_{eff} = 2\pi R_{eff}/\lambda$) to the fourth power ($x_{eff}^4$) (i.e., $\lambda^{-4}$) (Petty, 2006)."

558. Maybe switch "different" to "presented" or similar, as you already start the sentence with "the differences".

We have changed 'different' to 'presented'.

614. n_Hess isn't defined in the text, only in the Appendix. Please define it before use.

This sentence now reads: "The wavelength dependence of $Q_a$ is weaker than $Q_s$, (proportional to $x_{eff}$ (i.e, $\lambda^{-1}$)) and is strongly tied to the imaginary part of sulfate refractive index at 0% relative humidity ($n_{Hess}$; Appendix B) (Hess et al., 1998)."

638. No space in "long wave" for consistency

Fixed.

676. Can the word "also" just be added to make clear that this is a different comparison than the instrument validation: "E3SMv2-SPA also has slightly smaller Reff […]"

Added 'also' to the sentence.

1004. "teh" to "the"

Fixed

1005. Excess space in "SO 2"

Fixed